# Explaining with trees: interpreting CNNs using hierarchies

**Caroline Mazini Rodrigues**[*]
*Laboratoire de Recherche de l'EPITA – LRE – Le Kremlin-Bicêtre, 94270, France*
*& Univ Gustave Eiffel, LIGM – Marne-la-Vallée, 77454, France*
*caroline.mazinirodrigues@esiee.fr*

**Nicolas Boutry**
*Laboratoire de Recherche de l'EPITA – LRE – Le Kremlin-Bicêtre, 94270, France*
*nicolas.boutry@lrde.epita.fr*

**Laurent Najman**
*College of Computing and Mathematical Sciences, Dept of Mathematics, Khalifa University, Abu Dhabi, UAE*
*& Univ Gustave Eiffel, CNRS, LIGM – Marne-la-Vallée, 77454, France*
*laurent.najman@esiee.fr*

**Reviewed on OpenReview:** *https://openreview.net/forum?id=zjyWZh5IiI*

## Abstract

Challenges remain in providing interpretable explanations for neural network decision-making in explainable AI (xAI). Existing methods like Integrated Gradients produce noisy maps, and LIME, while intuitive, may deviate from the model's internal logic. We introduce a framework that uses hierarchical segmentation techniques for faithful and interpretable explanations of Convolutional Neural Networks (CNNs). Our method constructs model-based hierarchical segmentations that maintain fidelity to the model's decision-making process and allow both human-centric and model-centric segmentation. This approach can be combined with various xAI methods and provides multiscale explanations that help identify biases and improve understanding of neural network predictive behavior. Experiments show that our framework, *xAiTrees*, delivers highly interpretable and faithful model explanations, not only surpassing traditional xAI methods but shedding new light on a novel approach to enhancing xAI interpretability.

## 1 Introduction

In modern deep learning applications, especially in healthcare and finance, there is a growing need for transparency and explanation. Understanding a model's rationale is crucial before relying on its predictions. This need arises from biases present at various stages of model development and deployment. While some biases help in learning data distribution (Goyal & Bengio, 2022), others may indicate data imbalance, incorrect correlations, or prejudices in data collection.

Explainable Artificial Intelligence (xAI) provides methods that clarify models' decision-making processes with different levels of interpretability, which can be described as the measure of how easy it is to understand an explanation (Gilpin et al., 2018). Providing interpretable explanations is especially important in the previously mentioned contexts (e.g., health), where humans need to understand models' decisions.

For this purpose, some xAI methods use object-structure-based visualizations to enhance human interpretation. They decompose images in ways that mimic human perception, grouping objects by attributes such as color, texture, and edges (Hubel & Wiesel, 1959). Techniques such as LIME (Ribeiro et al., 2016) and

---

[*]https://carolmazini.github.io/

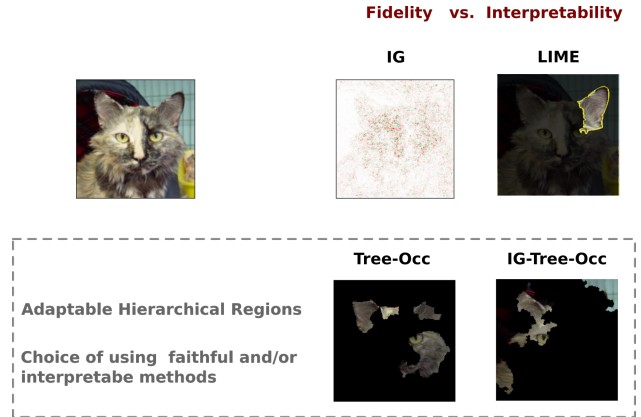

Figure 1: One challenge in xAI is achieving a good trade-off between fidelity and interpretability. We propose using region-based segmentation combined with hierarchies to adapt region size, while providing the flexibility to use high-fidelity methods for constructing the segmented regions.

KernelSHAP (Lundberg & Lee, 2017) have used this approach effectively, segmenting images into meaningful parts to improve interpretability. However, the size of the segmented regions affects the information extracted: small regions can be difficult to interpret, while large regions may overlook fine details. Additionally, using a segmentation framework may introduce a form of human bias, encouraging the model to attribute importance to structures that a human might consider relevant. This can aid comprehension but may reduce fidelity to the model's actual behavior (Miró-Nicolau et al., 2024), which does not necessarily align with human reasoning.

On the other hand, methods like Deconvolution (Zeiler & Fergus, 2014), Integrated Gradients (IG)(Sundararajan et al., 2017), and LRP(Bach et al., 2015), which attribute importance to features (pixels), aid in understanding deep learning models across various applications (Borys et al., 2023; Dharshini et al., 2023; Chaddad et al., 2023), serving as better approximations of model behavior (Borys et al., 2023), and locally explaining decisions. However, they often lack interpretability due to their pixel-level explanations, which can be difficult for humans to understand (Kim et al., 2018). Different techniques tend to prioritize either faithfulness to the model's behavior or human interpretability, making it challenging to balance the two.

We explore the trade-off between model fidelity and human interpretability in explaining Convolutional Neural Networks (CNNs) (Figure 1). We introduce *xAiTrees*, a framework that combines hierarchical segmentation with region-based explanation methods to produce human-friendly, multiscale visualizations inspired by Multiscale Interpretable Visualization (Ms-IV) (Rodrigues et al., 2024). Unlike conventional region-based XAI methods that rely on a fixed segmentation scale, *xAiTrees* leverages hierarchical segmentation to adapt region sizes across different levels of abstraction, mitigating the limitations of overly small or excessively coarse regions.

In addition, we propose a model-based segmentation strategy that uses pixel-wise attribution methods to approximate the model's "visual perspective," enabling the transformation of pixel-level explanations into coherent region-based interpretations.

To assess whether the proposed hierarchical and model-adaptive design effectively balances model fidelity and human interpretability, we conduct a series of quantitative evaluations comparing our approach with six representative explanation techniques, including perturbation-based, heatmap-based, concept-based, and attribution methods. The evaluation relies on three complementary metrics: occlusion, which measures the impact of removing regions identified as important; inclusion, which evaluates whether these regions are sufficient to support correct classification; and a novel metric, Pixel Impact Rate (PIR), designed to evaluate the specificity of the explanations by penalizing large regions being considered important. Across these metrics, our framework achieves competitive performance with strong baselines such as XRAI.

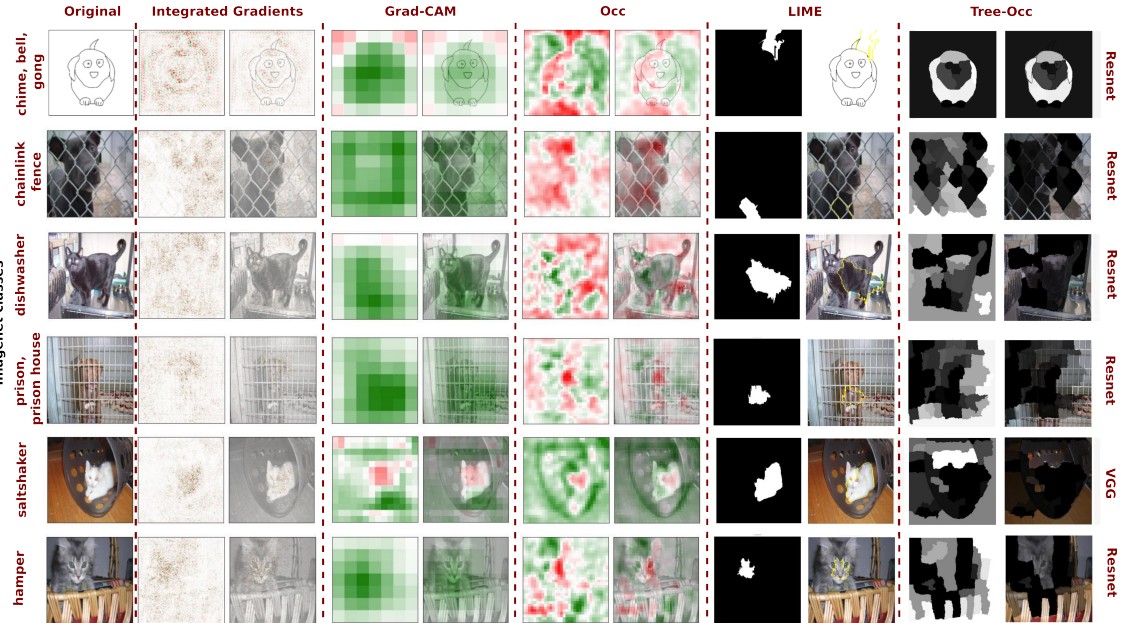

Figure 2: Explanations of six image classes misclassified by VGG-16 or Resnet18 models trained on ImageNet. We compare well-known xAI method explanations with one configuration of *xAiTrees*: Tree-Occ. Methods such as Integrated Gradients are noisy and difficult to interpret. Shapes such as the grades and the fence seem to be better highlighted by Tree-Occ, which is helpful for interpretation. When compared to highly interpretable methods like LIME, Tree-Occ avoids the mistake of highlighting the cat when the models predict classes such as dishwasher, saltshaker, and hamper.

Most importantly, we conduct a qualitative user study in which participants are presented with visual explanations generated by different explanation methods. The study evaluates, under three distinct bias scenarios, which methods best support (i) the detection of bias (i.e., determining whether the model is biased) and (ii) the identification of bias (i.e., understanding the nature of the bias). Across nearly all scenarios and evaluation questions, our proposed framework achieves the better performance.

The key contributions of this paper include:

1. We propose *xAiTrees*, a hierarchical segmentation explanation framework that aggregates importance across multiple spatial scales, enabling humans to better understand and diagnose model bias;
2. We evaluate *xAiTrees* using several complementary metrics — occlusion, inclusion, and the newly proposed Pixel Impact Rate (PIR) — to assess both the sufficiency and specificity of explanations;
3. We conduct a human-subject study showing that our explanations more effectively help users detect and identify biases under multiple bias scenarios.

We organize the paper as follows: in Section 2, we present some prior research on xAI. Section 3 outlines the preliminary concepts used in our framework, while in Section 4 we provide a detailed explanation of our methodology. In Section 5 we present and discuss our experimental results. Finally, we conclude and discuss possible future research directions

## 2 Related work

**Classification problems and xAI:** One fundamental task in machine learning is classification. The basic concept involves working with a training dataset, denoted as $\mathcal{DS} = (\mathcal{I}_i, GT_i)_{i \in [1, NbIm]}$, which consists of pairs of images $\mathcal{I}_i$ and their associated labels $GT_i$. Each label belongs to one of a set of classes represented by $c \in [1, NbClasses]$. The goal is to train a model, denoted as $\Xi$, to effectively distinguish between different classes within the dataset.

In this configuration, we express $\Xi$ as $\Xi = \Xi^{classif} \circ \Xi^{enc}$, the combination of two elements: an $\Xi^{enc}$, responsible for converting each input image $\mathcal{I}_i$ into a feature vector, and a $\Xi^{classif}$, which analyzes these features to classify the images. The outcome of this process, referred to as the "logit" for image $\mathcal{I}_i$, is a vector $\mathbf{out}_i \in \mathbb{R}^{NbClasses}$ that signifies the activation levels across various classes. Typically, we apply a *Softmax* layer to $\mathbf{out}_i$ to determine the class with the highest activation, ideally aligning with the ground truth label $GT_i$ for perfect classification.

**Pixel-wise explanations:** In neural networks, optimizing the model $\Xi$ involves the backpropagation process. Exploiting this process, certain explainable Artificial Intelligence (xAI) methods like Integrated Gradients (Sundararajan et al., 2017), Guided-Backpropagation (Springenberg et al., 2015), and Deconvolution (Zeiler & Fergus, 2014) utilize it to identify input features that enhance the response of a specific class, aiming to maximize the value of a particular position in the output vector $\mathbf{out}_i$. Consequently, attribution maps are generated, illustrating pixel-level explanations, as depicted in Figure 2 for Integrated Gradients (IG).

**Region-based explanations:** Additional techniques like Sensitivity Analysis (Zeiler & Fergus, 2014), LIME (Ribeiro et al., 2016), and SHAP (Lundberg & Lee, 2017) utilize occlusions of image regions to assess the network's sensitivity to each region within an image. These methods provide explanations at a region level rather than a pixel level, as illustrated in Figure 2 for LIME. More recently, a region-based technique, XRAI (Kapishnikov et al., 2019), proposed to combine Integrated Gradients (Sundararajan et al., 2017) and perturbation-based approaches to generate saliency maps as explanations.

**Concept-based explanations:** However, many of these techniques focus on explaining individual samples separately, which limits our understanding of how the model behaves globally across various scenarios. That is why methods like TCAV (Kim et al., 2018), ACE (Ghorbani et al., 2019), Explanatory graphs (Zhang et al., 2018), LGNN (Tan et al., 2022), and Ms-IV (Rodrigues et al., 2024) aim to comprehend the overall behavior of the model. In particular, Ms-IV also considers the impact of occlusions, not on individual predictions, but on the model's output space.

**Neural-symbolic explanations:** Techniques such as DCR (Barbiero et al., 2023), X-NeSyL (Díaz-Rodríguez et al., 2022), and the approach proposed by (Ngan et al., 2023) adopt neural-symbolic strategies to explain the decisions of neural networks within specific contexts. These methods enhance transparency by translating low-level neural activations into high-level concepts, logical rules, or hierarchical structures, making explanations more aligned with human reasoning. However, this often comes at the cost of increased complexity, as such approaches may require manually crafted symbolic rules or domain-specific annotations to generate meaningful explanations.

## 3 Preliminaries

To ensure a thorough understanding of the sequel, we provide in this section the general techniques and metrics employed during this work. In subsection **A**, we provide a brief overview of the selected hierarchical segmentation techniques, highlighting their significance. In subsection **B**, we shortly present the occlusion-based metrics used in the construction of our methodology.

**A. Segmentation techniques:** As an important step for our framework, we employ image segmentation algorithms that decompose images into more interpretable structures, enabling better human understanding and interpretation. We specifically employ hierarchical segmentation techniques due to their capability to decompose images into multiple levels of detail, from fine to coarse, mirroring how humans naturally perceive objects: initially observing the overall structure before delving into the finer details. A hierarchical segmentation algorithm produces a merging tree, that indicates how two given regions merge. In this paper, we use the tree structures available in the Higra package (Perret et al., 2019; 2018): Binary Partition Tree (BPT) and Hierarchical watershed. See details in A.1.

**B. Occlusion-based metrics:** In this work, we use two metrics to generate our segmentation based on the model explainability: (i) *Occlusion*, which is the impact of occluding an image region on its classification output, and (ii) CaOC, which is the intra-class impact of occluding an image region that employs a sliding metric that ranks images based on the highest activations for a given class. This sliding metric measures

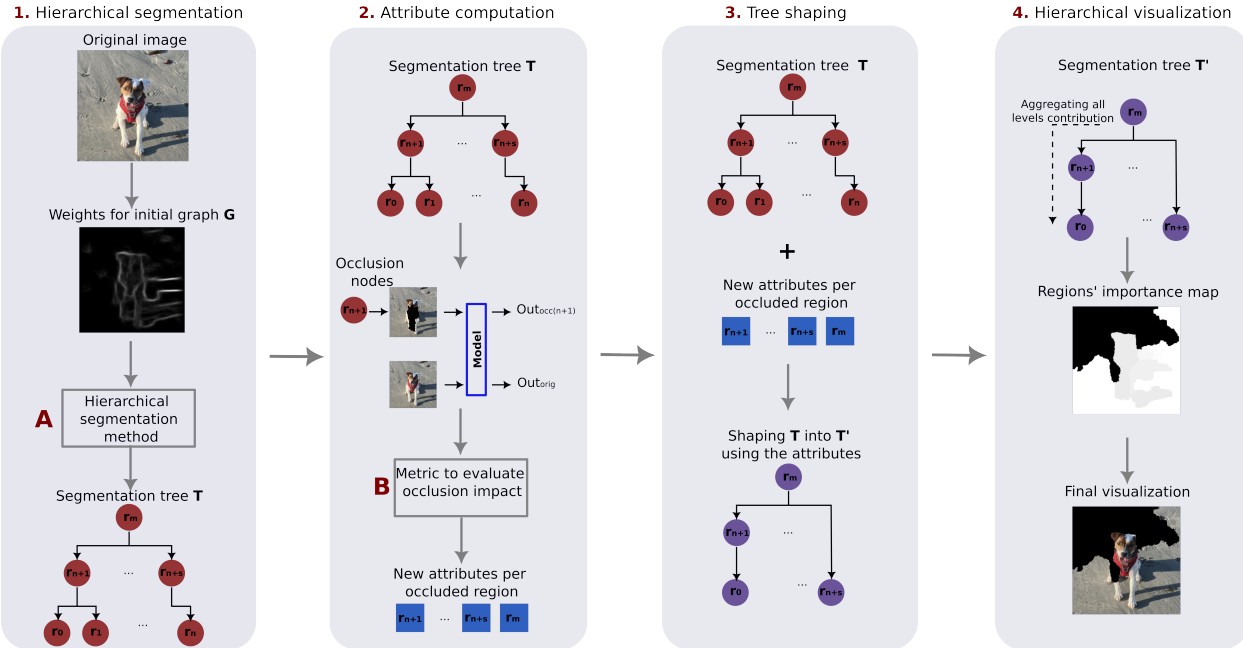

Figure 3: Our framework *xAiTrees* operates through four key steps: **1.** Generate a segmentation hierarchy using either the image's edge map for human-based segmentation or pixel-wise importance based on xAI techniques for model-based segmentation. **2.** Systematically occlude each region of the segmentation to evaluate its impact on the model's decision, obtaining an occlusion attribute for each region. **3.** Assess the persistence of the occlusion attribute using a shaping approach (Xu et al., 2015; 2016). **4.** Aggregate the contributions of each region from the highest to the lowest level of the tree to create a comprehensive multiscale visualization.

the movement in this ranking after occluding a region of the image (detailed in A.2). Although the main experimentation uses these methods, we present in A.7 a framework's variation using LIME to show *xAiTrees* versatility with other types of xAI techniques. Explanations example in Figure 9.

# 4   Methodology

In this section, we outline our four-step methodology (Figure 3): (1) hierarchical segmentation, (2) attribute computation, (3) tree shaping, and (4) hierarchical visualization. In step 1, we convert the data into a hierarchical representation, creating various regions at different scales in the image. In step 2, we evaluate some xAI-based attributes (**B**) on the regions. In step 3, we assess the importance of the region attributes. Finally, in step 4, we explain how to generate a visualization map from the importance of the attributes.

**1. Hierarchical segmentation:**

Intuitively, any hierarchical segmentation algorithm works by iteratively merging first the pixels, then the regions, according to a similarity criterion. In this paper, we test two ways for measuring the similarity: human-based and model-based.

- The human-based approach relies on the Structured Edge Detection (SED) algorithm (Dollár & Zitnick, 2014), which captures complex edge patterns and produces precise edge maps, in accordance with human intuition.
- The model-based approach uses a visual representation of the image's pixels most influential in a model's decision. Although less intuitive for humans, this approach helps to understand how the model reasons. We test pixel-wise explainable AI methods: Integrated Gradients (IG) (Sundararajan et al., 2017), Guided-Backpropagation (Springenberg et al., 2015), Input x Gradient (Shrikumar et al., 2017), and Saliency (Si-

**Eyes outside hierarchy**

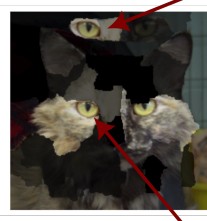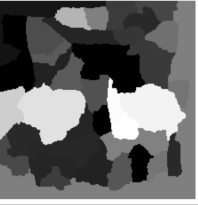

**Eyes inside hierarchy**

Figure 4: The hierarchy aids to discriminate similar important regions. Example of the method's behavior with the same structure inside and outside a hierarchy. The cat's eyes were replicated outside the cat's face. However, the importance of each region is the combination of the importance of each hierarchy part. Therefore, the cat's eyes inside the face (an important hierarchy region) score higher, as evidenced by the lighter regions in the right image.

monyan & Zisserman, 2015) (all from Captum framework). The methods were chosen because of their availability (in frameworks like Captum) and for their higher fidelity as pixel-wise importance attribution based on gradients (Miró-Nicolau et al., 2024).

Using such a similarity criterion, we obtain a hierarchical segmentation, which can be represented as a tree **T**, completing the first step of our pipeline. See Figure 3, first column.

**2. Attribute computation:** The segmentation tree generated in the previous step provides many segments. We assess the model's response on each segmented region in the tree, for all regions large enough. We apply a metric to evaluate the occlusion impact caused by each region. These occlusion scores reveal the influence of each segmented regions on the model's output. The metric employed to assess the impact of regions can be any occlusion-based metric. See Figure 3, second column.

**3. Tree shaping:** To assess the importance of the nodes' attributes, it is not enough to simply take the regions with the highest attributes: there are different levels of the hierarchy to be analyzed. Instead, we rely on a process called *shaping* (Xu et al., 2015; 2016). The main idea is to look at the undirected, vertices-weighted graph $G$, whose vertices are the node of **T**, whose edges are formed by the parent-children relationship in **T**, and whose weights are the attributes of the nodes. We now look at the level-sets of $G$. A vertex of $G$ (a node of **T**) is important according to its persistence in the level sets of $G$. More precisely, a connected component is born when a local maximum of the attribute appear; when two connected components merge, one of the two maxima disappear, and the time of life of this maximum is its persistence. We can compute such persistence by building a new tree **T'** on $G$, **T'** is the tree of all the connected component of the upper-level sets of $G$. The persistence of a node of **T** is easily computed on **T'** by computing the length of the branch it belongs to. We refer to Xu et al. (2015; 2016) for more details. See Figure 3, third column.

**4. Construction of the hierarchical visualization:** With **T'** from the previous step, we now produce a visualization of the important regions. Using the persistence of a node directly for visualization can yield conflicting results for interpretation. Consider an example where we want to generate explanations for a model that classifies images of dogs. The persistence might indicate that eyes are the primary features for correct classification. If the image under scrutiny shows a dog with its owner, the persistence might erroneously highlight the eyes of both the human and the dog as relevant, which is misleading since only the dog's eyes should matter (in an ideal, unbiased model). To avoid such effect, we recursively sum the persistence of each node from the root to the leaves of **T'**. This ensures that smaller segments inherit the importance of their parent nodes. In our example, if the parent segment of the eyes is the entire face, the dog's face carries importance for the model's decision, while the human face does not. By adding the dog's facial region information to the eye segments, we ensure the dog's eyes are prioritized over the human eyes and, therefore, become more prominent in the explanation. This process aggregates the importance of various scales of the image into the pixels, resulting in a hierarchical, multi-scale, visualization. We show an example in Figure 4. We use this aggregated persistence as the final score for each region of the hierarchical segmentation. We

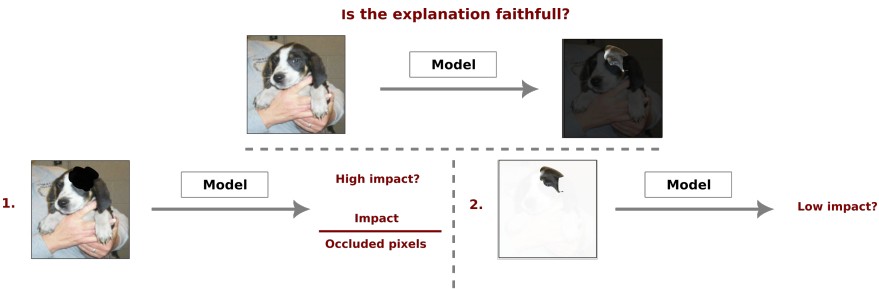

Figure 5: Our quantitative experiments aim to answer the question: Is the explanation faithful? In (1), we occlude the selected important regions and evaluate the impact using the Pixel Impact Rate (PIR) based on the model's response. In (2), we evaluate the inclusion of only the important regions by analyzing their Softmax and Accuracy Information Curves.

select a minimum importance score, and retain the regions accordingly. We superpose the retained region on the original image to generate the **Final visualization** (Figure 3, fourth column).

## 5    Experiments and results

We evaluated the methods using two architectures, VGG-16 (Simonyan & Zisserman, 2015) and ResNet18 (He et al., 2016), trained on three datasets: Cat vs. Dog (Cukierski, 2016) (RGB images size 224x224), CIFAR-10 (Krizhevsky, 2009; Krizhevsky et al., 2009) (RGB images size 32x32), and ImageNet (Deng et al., 2009) (RGB images). Explanations were generated for 512 images from the Cat vs. Dog dataset, 10,000 images from the CIFAR-10 dataset, and 100,000 images from the ImageNet test set. A detailed description of the methods' parameters and datasets is provided in A.3, A.4 and A.5. We organize our experiments and results into two categories: quantitative and qualitative analysis. In the quantitative analysis, we conduct a series of experiments utilizing the metrics discussed in Section 3 to assess the impact of image region occlusion of various explainable frameworks. During our qualitative analysis, we do a more subjective examination, evaluating the human interpretability of the explanations generated by the models. The experiments were conducted on GPU (NVIDIA Quadro RTX 8000 48GB). We discuss some limitations in **??**.

### 5.1    Quantitative evaluations

We selected state-of-the-art region-based methods as baseline (**B**) to be compared: Occlusion, Grad-CAM, LIME, Ms-IV, and XRAI (configuration of each method in A.4). Although ACE presents good concept-based explanations, we only use it in the human evaluation experiments because, as a global explanation method, it is not directly comparable to the local ones in these quantitative experiments (more details in A.6). We compare the baseline methods with different configurations of our proposed methodology, showing its adaptability (all the configurations in A.3 and A.5). This is done to explore variations, including different sizes of minimal regions in the visualizations, pixel weights for graph construction in segmentation, and methods for generating hierarchical segmentation. We show eight of them here compared to the baseline methods. When we refer to **Tree-CaOC** or **TreeW-CaOC**, we mean the human-based segmentation (edges' map) using Watershed area and CaOC as occlusion metric. When we refer to **IG-Tree-Occ** or **IG-TreeW-Occ**, we mean the model-based segmentation (using Integrated Gradients (IG) attributions) using Watershed area and Occ (simple occlusion – Equation (1)) as occlusion metric. When we refer to **BP-TreeB-Occ**, we mean the model-based segmentation (using Guided Backpropagation (BP) attributions) using BPT and Occ as occlusion metric. In the results, we refer to configurations using watershed as group **C1** and using BPT as group **C2**.

In **C1**, we present results for minimal regions of 500 pixels (Cat vs. Dog and ImageNet) and 64 pixels (CIFAR-10), utilizing edges and Integrated Gradients (IG) as pixel weights, and the watershed-by-area hierarchical segmentation. This configuration was selected for its stability across different minimal region sizes in the datasets, and its visualizations were used for human evaluation. While **C2** presents results

Table 1: Percentage of images with the original class changed after the **exclusion** of selected explanation regions. We test two configurations of our methodology (**C1** and **C2** – other configurations in Supplementary Materials) against four region-based baseline methods, Occlusion, Grad-CAM, LIME, Ms-IV, and XRAI, in two architectures, VGG-16 and ResNet18, and datasets, Cat vs. Dog, CIFAR10, and ImageNet. We expect higher percentage of class change (**Ch.**) when the region is excluded. **Same** column shows images maintaining the original class when the output was reduced, and **Total** is the sum of class change (**Ch.**) and class reduction (**Same**). We compare the each method to the best configuration (**BP-TreeB-Occ**) showing the p-score in brackets (Mcnemar test).

| % of images | | Cat vs. Dog | | | | | | Cifar10 | | | | | | Imagenet | | | | | |
|---|---|---|---|---|---|---|---|---|---|---|---|---|---|---|---|---|---|---|---|
| | | VGG | | | ResNet | | | VGG | | | ResNet | | | VGG | | | ResNet | | |
| | | Ch. | Same | Total | Ch. | Same | Total | Ch. | Same | Total | Ch. | Same | Total | Ch. | Same | Total | Ch. | Same | Total |
| **B** | Occlusion | 0.05(0.0) | 0.93 (0.0) | 0.98 | 0.06 (0.0) | 0.89 (0.0) | 0.95 | 0.26 (0.0) | 0.60 (0.0) | 0.86 | 0.30 (0.0) | 0.62 (0.0) | 0.92 | **0.39 (0.0)** | 0.60 (0.0) | **0.99** | 0.39 (0.0) | 0.59 (0.0) | **0.98** |
| | Grad-CAM | 0.07(0.0) | 0.82 (0.0) | 0.89 | 0.13 (0.0) | 0.83 (0.0) | 0.96 | 0.14 (0.0) | 0.42 (0.0) | 0.56 | 0.90 (0.0) | 0.09(0.0) | 0.99 | 0.25 (0.0) | 0.65 (0.0) | 0.90 | **0.68 (0.0)** | 0.30 (0.0) | **0.98** |
| | LIME | 0.07 (0.0) | 0.83 (0.0) | 0.90 | 0.07 (0.0) | 0.76 (0.0) | 0.83 | **0.84 (0.0)** | 0.14 (0.0) | **0.98** | **0.82 (2.4-5)** | 0.16 (0.17) | **0.98** | 0.34 (0.0) | 0.61 (0.0) | 0.95 | 0.38 (0.0) | 0.55 (0.0) | 0.93 |
| | Ms-IV | 0.06(0.0) | 0.76 (0.0) | 0.82 | 0.07 (0.0) | 0.66 (0.0) | 0.73 | 0.30 (0.0) | 0.42 (0.0) | 0.72 | 0.33 (0.0) | 0.47 (0.0) | 0.80 | **0.44 (0.0)** | 0.49 (0.0) | 0.93 | **0.48 (0.0)** | 0.43 (0.0) | 0.91 |
| | XRAI | 0.04 (0.0) | 0.85 (0.0) | 0.89 | 0.06 (0.0) | 0.79 (0.0) | 0.85 | **0.52 (0.0)** | 0.35 (0.0) | **0.87** | **0.50 (0.0)** | 0.40 (0.0) | **0.90** | **0.41 (0.0)** | 0.52 (0.0) | 0.93 | **0.45 (0.0)** | 0.46 (0.0) | 0.91 |
| **C1** | TreeW-CaOC | 0.16 (0.0) | 0.48 (2.0-5) | 0.64 | 0.22 (0.0) | 0.41 (0.15) | 0.63 | 0.12 (0.0) | 0.16 (0.0) | 0.28 | 0.15 (0.0) | 0.19 (2.8-9) | 0.34 | 0.23 (0.0) | 0.51 (0.0) | 0.74 | 0.26 (0.0) | 0.46 (0.0) | 0.72 |
| | TreeW-Occ | **0.31 (0.0)** | 0.63 (3.2-4 | **0.94** | **0.35 (1.3-11)** | 0.60 (0.01) | **0.95** | 0.60 (0.0) | 0.15 (0.0) | 0.75 | 0.60 (0.0) | 0.15 (0.0) | 0.65 | **0.40 (0.0)** | 0.56 (0.0) | **0.96** | **0.44 (0.0)** | 0.51 (0.0) | **0.95** |
| | IG-TreeW-CaOC | 0.29 (0.0) | 0.46 (0.0) | 0.75 | 0.21(0.0) | 0.45 (7.61-13) | 0.66 | 0.20 (0.0) | 0.29 (0.0) | 0.49 | 0.23 (0.0) | 0.33 (0.38) | 0.56 | 0.26 (0.0) | 0.52 (0.0) | 0.78 | 0.30 (0.0) | 0.46 (0.0) | 0.86 |
| | IG-TreeW-Occ | **0.43 (1.7-11)** | 0.54 (5.3-10) | **0.97** | **0.32 (1.14-13)** | 0.61 (2.3-14) | **0.93** | **0.73 (0.0)** | 0.21 (0.0) | **0.94** | **0.73 (0.0)** | 0.22 (0.0) | **0.95** | **0.43 (0.0)** | 0.53 (0.0) | **0.96** | **0.48 (0.0)** | 0.48 (0.0) | **0.96** |
| **C2** | TreeB-CaOC | **0.35 (0.0)** | 0.39 (0.1) | 0.74 | 0.27 (0.0) | 0.42 (0.09) | 0.69 | 0.15 (0.0) | 0.26 (0.0) | 0.41 | 0.17 (0.0) | 0.31(0.0) | 0.48 | 0.23 (0.0) | 0.49 (0.0) | 0.72 | 0.25 (0.0) | 0.44 (0.0) | 0.69 |
| | TreeB-Occ | **0.51 (3.9-5)** | 0.44 (0.06) | **0.95** | **0.41 (2.0-6)** | 0.52 (0.28) | **0.93** | **0.81 (0.0)** | 0.12 (0.0) | **0.93** | **0.76 (0.0)** | 0.17 (0.0) | **0.93** | **0.57 (0.0)** | 0.38 (0.0) | **0.95** | **0.60 (0.0)** | 0.35 (0.0) | **0.95** |
| | BP-TreeB-CaOC | **0.56 (5.7-6)** | 0.32 (0.001) | 0.88 | **0.39 (2.2-16)** | 0.39 (4.5-7) | 0.78 | 0.09 (0.0) | 0.34 (1.7-9) | 0.43 | 0.11 (0.0) | 0.39 (0.03) | 0.50 | 0.11 (0.0) | 0.36 (0.0) | 0.47 | 0.10 (0.0) | 0.31 (0.0) | 0.41 |
| | BP-TreeB-Occ | **0.63** | 0.35 | **0.98** | **0.55** | 0.37 | **0.92** | **0.88** | 0.10 | **0.98** | **0.80 (0.0)** | 0.16 | **0.96** | **0.71 (0.0)** | 0.16 (0.0) | **0.87** | **0.74 (0.0)** | 0.13 (0.0) | **0.87** |

for minimal regions of 200 pixels (Cat vs. Dog and ImageNet) and 4 pixels (CIFAR-10), using edges and Guided Backpropagation (BP) as pixel weights, and the BPT tree. This configuration yielded the highest performance. In A.5, we provide comprehensive results for other configurations. Here, we propose four main quantitative evaluations, (i and ii) inspired by feature removal (Covert et al., 2021) and (iii and iv) inspired by Performance Information Curves (PICs) (Kapishnikov et al., 2019): (i) Exclusion of important regions; (ii) Inclusion of important regions; (iii) Softmax Information Curve (SIC); and (iv) Accuracy Information Curve (AIC). For (i) and (ii), we used the McNemar test (McNemar, 1947) to compare each method with the best configuration and determine whether there were statistically significant differences between the results. We show the main idea of this evaluation in Figure 5, and the details are discussed below:

**Exclusion of important regions:** Given that each region-based explainable AI (xAI) method identifies important regions that explain the prediction of a model, we performed occlusion of these regions, in order to measure the impact of each selection. For methods that assign scores to regions, we masked the 25% highest scores (this excludes LIME, which inherently provides information to directly mask each region – a detailed explanation is included in A.6).

The first idea for the metric was to calculate the impact on the *logits* after occlusion. However, any kind of perturbation can affect the *logits* and not necessarily the classification. In this particular case, since we are dealing with a classification problem, we consider the class change as the main evidence that an important image region has been occluded. Therefore, the values from Table 1 **Ch.** is the percentage of class changing images, **Same** is the percentage of images with same class prediction but with reduced *logits* (reduced classification certainty), and **Total** is the percentage of all images with the class negatively impacted by the removal of important regions (sum of **Ch.** and **Same**).

In Table 1, we present results (Ch., Same, Total) for each explainable technique (**B** – Baseline, **C1**, and **C2** – our proposition) applied to a network (VGG or ResNet) classifying images from a dataset (Cat vs. Dogs, Cifar10, or ImageNet). Higher **Ch.** values indicate that the identified regions are more class-representative. High **Same** values complement **Ch.**, suggesting that the best results are shown by higher **Ch.** and **Same** values. Thus, while **Total** sums **Ch.** and **Same**, the optimal result is reflected by initially higher **Ch.** and then higher **Same** values.

Table 1 shows that baseline methods like **Occlusion** achieved **Total** values above 80%, but our **C2** configurations delivered the best results, particularly for class changes (**Ch.**). Our methods had over 60% class changes for Cat vs. Dog images, exceeded 80% for CIFAR-10, and 70% for ImageNet. Among baselines, LIME and XRAI performed best. The results highlight the superiority of our **C1** and **C2** configurations in identifying impactful regions, achieving robust performance across datasets, and providing more accurate insights into deep neural network interpretability.

Table 2 presents results from a second experiment addressing reduced precision in explanations where methods highlight the entire image as important, potentially leading to class changes upon occlusion. To address this, we propose the **Pixel Impact Rate (PIR)**, a metric that quantifies the per-pixel impact on class activation during occlusion. Unlike the percentage of class change, PIR distinguishes whether changes are caused from full or near-full image occlusion(details in A.5 Equation 2). Higher PIR values indicate significant average impact per pixel, while lower PIR suggests occlusion of larger portions or the entire image (imprecise explanations). The table summarizes the average (**avg**) and standard deviation (**std**) of PIR across networks, techniques, and datasets.

Table 2: Pixel Impact Rate (PIR) of the chosen regions. The metric is the rate of the impact under occlusion (difference between the original class output and the output under occlusion) by the number of pixels of the occlusion mask. We test two configurations of our methodology (**C1** and **C2** – other configurations in Supplementary Materials) against four region-based baseline methods, Occlusion, Grad-CAM, LIME, Ms-IV, and XRAI, in two architectures, VGG-16 and ResNet18, and datasets, Cat vs. Dog, CIFAR10, and ImageNet. We expect higher values, on average, for PIR, meaning each occluded pixel has a high impact.

| | PIR | Cat vs. Dog | | | | Cifar10 | | | | Imagenet | | | |
|---|---|---|---|---|---|---|---|---|---|---|---|---|---|
| | | VGG | | ResNet | | VGG | | ResNet | | VGG | | ResNet | |
| | | avg | std | avg | std | avg | std | avg | std | avg | std | avg | std |
| **B** | Occlusion | **4.60e-03** | 4.05e-03 | **1.50e-03** | 1.28e-03 | 8.95e-02 | 1.48e-01 | **9.67e-02** | 1.35e-01 | **1.16e-02** | 1.13e-02 | **8.02e-03** | 7.03e-03 |
| | Grad-CAM | **1.12e-03** | 1.02e-03 | 2.76e-04 | 2.07e-04 | 6.39e-03 | 1.34e-02 | 5.38e-03 | 1.46e-03 | 3.05e-03 | 3.16e-03 | 1.11e-03 | 8.02e-04 |
| | LIME | **9.03e-04** | 1.10e-03 | 3.47e-04 | 3.89e-04 | 6.75e-03 | 3.27e-03 | 6.41e-03 | 3.25e-03 | 2.11e-03 | 2.50e-03 | 1.58e-03 | 1.75e-03 |
| | Ms-IV | 4.30e-04 | 4.74e-04 | 1.83e-04 | 2.45e-04 | **1.16e-02** | 1.44e-02 | **1.18e-02** | 1.33e-02 | 9.73e-04 | 1.10e-02 | 7.21e-04 | 7.59e-04 |
| | XRAI | **1.16e-03** | 9.92e-04 | 4.44e-04 | 6.32e-04 | **2.09e-02** | 1.77e-02 | **1.92e-02** | 2.02e-02 | 3.05e-03 | 1.09e-02 | 2.04e-03 | 5.90e-03 |
| **C1** | Tree-CaOC | 3.61e-04 | 4.70e-04 | 1.92e-04 | 2.86e-04 | 4.73e-03 | 1.02e-02 | 5.56e-03 | 1.05e-02 | 1.16e-03 | 1.60e-03 | 1.10e-03 | 1.47e-03 |
| | Tree-Occ | **3.66e-04** | 5.30e-04 | **1.69e-04** | 2.52e-04 | 9.20e-03 | 1.32e-02 | 9.88e-03 | 1.21e-02 | 1.10e-03 | 1.50e-03 | 1.05e-03 | 1.39e-03 |
| | IG-Tree-CaOC | 3.04e-04 | 3.48e-04 | 2.26e-04 | 3.09e-04 | 9.55e-03 | 1.30e-02 | 9.51e-03 | 1.27e-02 | 1.54e-03 | 1.83e-03 | 1.46e-03 | 1.66e-03 |
| | IG-Tree-Occ | **3.10e-04** | 3.61e-04 | **2.11e-04** | 3.05e-04 | **1.69e-02** | 1.56e-02 | **1.63e-02** | 1.40e-02 | 1.52e-03 | 1.72e-03 | 1.46e-03 | 1.58e-03 |
| **C2** | TreeB-CaOC | 2.16e-04 | 2.91e-04 | 1.26e-04 | 2.26e-04 | 8.92e-03 | 1.90e-02 | 1.12e-02 | 2.09e-02 | 7.20e-04 | 1.08e-03 | 6.76e-04 | 9.63e-04 |
| | TreeB-Occ | **2.26e-04** | 3.32e-04 | 1.03e-04 | 1.81e-04 | **1.14e-02** | 2.06e-02 | **1.15e-02** | 2.00e-02 | 5.83e-04 | 8.19e-04 | 5.13e-04 | 7.27e-04 |
| | BP-TreeB-CaOC | **5.23e-03** | 3.59e-02 | **2.58e-03** | 1.81e-02 | **1.94e-01** | 3.87e-01 | **1.94e-01** | 3.52e-01 | **1.43e-02** | 7.37e-02 | **1.19e-02** | 5.15e-02 |
| | BP-TreeB-Occ | **8.64e-04** | 1.60e-02 | **1.18e-03** | **8.90e-03** | **1.10e-01** | 4.34e-01 | **1.18e-01** | 4.14e-01 | **4.51e-03** | 3.43e-02 | **3.25e-03** | 2.35e-02 |

In the PIR experiments (Table 2), **C2**, particularly **BP-TreeB-CaOC**, achieved the highest average PIR values, with **Occlusion** and **XRAI** also performing well. These methods demonstrated strong region specificity, enhancing the impact of occluded pixels. However, it is also important to consider the method's stability across different images, indicated by a smaller PIR standard deviation (**std**). While **C2** presented higher PIR, **C1** showed greater consistency (smaller **std**).

**Inclusion of important regions:** Additional experimentation was conducted to demonstrate a method's capability to identify an image region with sufficient information for the original class. The goal of this experiment is to determine whether the selected important region, when the only one left unoccluded in the image, can maintain the classification in its expected class. This experiment elucidates the critical role of these identified regions, providing strong evidence that they indeed contain essential information for accurate classification. We occluded **all regions** in the images except for the one selected by each method. We then calculated the percentage of images that changed class. The results are presented in Table 3. Lower percentages indicate better performance, as they mean that a smaller percentage of images changed class, demonstrating that the chosen regions were sufficient to preserve the class for most of the images.

Table 3 demonstrates that both LIME and our configurations (**C1**, **C2**) effectively identify regions that describe a class. However, **BP-TreeB-Occ** outperformed LIME, with fewer images changing class, indicating it provides more essential information for class attribution. Additional insights include: **Occlusion** combined with our methodology yields superior local explanations, and "model"-based segmentation enhances explanation fidelity. Overall, our method outperformed traditional xAI baselines, including LIME, which still delivered consistently good results.

**SIC/AIC for hierarchy evaluation:** As explained in Section 4, the hierarchy of our explanation is combined by summing up importance regions values. Therefore, to select different hierarchies it suffices to filter by different scores. In this line, we evaluate our methodology by imposing different thresholds for the explanations. Inspired by the metrics Softmax Information Curve (SIC) and Accuracy Information Curve (AIC) proposed by Kapishnikov et al. (2019) we calculated the Softmax and Accuracy curves by including only selected image regions as model input. To preserve the original data distribution, we integrated these

Table 3: Percentage of images with the original class changed after the **inclusion** (exclusively) of this same regions. We test two configurations of our methodology (**C1** and **C2** – other configurations in Supplementary Materials) against five region-based baseline methods, Occlusion, Grad-CAM, LIME, Ms-IV, and XRAI, in two architectures, VGG-16 and ResNet18, and datasets, Cat vs. Dog, CIFAR10, and ImageNet. We expect lower when the region in included. We compare each method to the best configuration (**BP-TreeB-Occ**) showing the p-score in brackets (Mcnemar test).

| % of images | | Cat vs. Dog | | Cifar10 | | Imagenet | |
|---|---|---|---|---|---|---|---|
| | | VGG | ResNet | VGG | ResNet | VGG | ResNet |
| **B** | Occlusion | 0.47 (0.0) | 0.50 (0.0) | 0.89 (0.0) | 0.89 (0.0) | 0.99 (0.0) | 0.99 (0.0) |
| | Grad-CAM | 0.51 (0.0) | 0.30 (2.35-5) | 0.86 (0.0) | 0.0 (0.0) | 0.99 (0.0) | **0.87 (0.0)** |
| | LIME | **0.16 (2.02-10)** | 0.30 (6.44-5) | **0.44 (0.14)** | **0.49 (2.62-9)** | 0.93 (0.0) | 0.95 (0.0) |
| | Ms-IV | 0.20 (3.11-14) | 0.54 (0.0) | 0.78 (0.0) | 0.81 (0.0) | **0.88 (0.0)** | **0.92 (0.0)** |
| | XRAI | 0.32 (0.0) | 0.41 (2.03-14) | 0.70 (0.0) | 0.74 (0.0) | 0.96 (0.0) | 0.97 (0.0) |
| **C1** | Tree-CaOC | 0.26 (0.0) | 0.32 (3.54-7) | 0.80 (0.0) | 0.85 (0.0) | 0.96 (0.0) | 0.97 (0.0) |
| | Tree-Occ | **0.17 (0.0)** | **0.23 (0.0)** | **0.54 (0.0)** | **0.61 (0.0)** | **0.90 (0.0)** | **0.91 (0.0)** |
| | IG-Tree-CaOC | 0.41 (8.12-11) | 0.43 (0.05) | 0.84 (0.0) | 0.83 (0.0) | 0.98 (0.0) | 0.99 (0.0) |
| | IG-Tree-Occ | 0.19 (1.37-12) | 0.42 (0.05) | **0.68 (0.0)** | **0.69 (0.0)** | **0.92 (0.0)** | **0.93 (0.0)** |
| **C2** | TreeB-CaOC | **0.16 (1.8-9)** | 0.22 (0.15) | 0.79 (0.0) | 0.81 (0.0) | 0.95 (0.0) | 0.96 (0.0) |
| | TreeB-Occ | **0.11 (0.0)** | **0.19 (8.44-7)** | **0.44 (0.0)** | **0.51 (0.0)** | **0.72 (0.0)** | **0.74 (0.0)** |
| | BP-TreeB-CaOC | 0.38 (3.7-5) | **0.28 (0.79)** | 0.87 (0.02) | 0.86 (0.01) | 0.97 (0.0) | 0.98 (0.0) |
| | **BP-TreeB-Occ** | **0.04** | 0.18 | **0.45** | **0.53** | **0.51 (0.0)** | **0.50 (0.0)** |

important regions back into a blurred version of the original image (details in A.5). The regions were selected based on thresholds of 0.5%, 1%, 2%, 3%, 4%, 5%, 7%, 10%, 13%, 21%, 34%, 50%, and 75% percent, representing the most significant region values according to each evaluated xAI method. These thresholds, represented on the x-axis, indicate the percentage of important regions required to affect accuracy and class activations. Figure 6 shows the results for 1,000 randomly selected images for AIC (due to time consumption restrictions – time analysis in A.5) from the ImageNet dataset and VGG16 model, with additional results for ResNet18 and the Cat vs. Dog dataset provided in A.5.

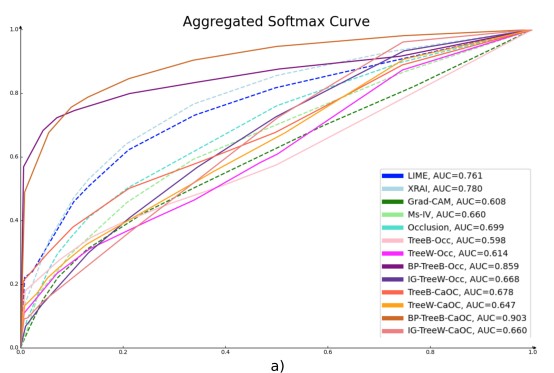
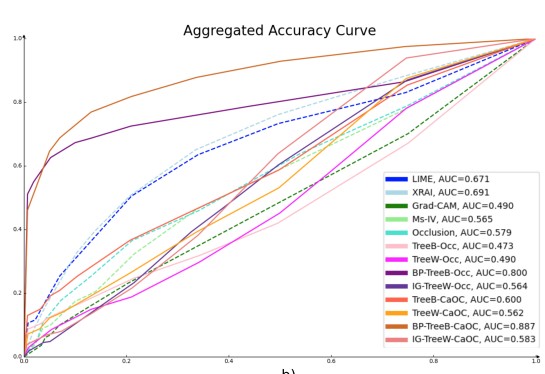

Figure 6: Softmax (a) and Accuracy (b) when including regions filtered by different percentage thresholds of most important scores. We evaluate each threshold as a hierarchy level in eight configurations of *xAiTrees* (**C1** and **C2**), in a bottom-up approach (from smaller highly important regions to the bigger structures). We compare these configurations to the baselines: LIME, XRAI, Grad-CAM, Ms-IV, and Occlusion, by filtering the maps using the same threshold. The curves are averaged across 1,000 randomly selected images from Imagenet dataset. AUC values are included in the graphs. BP-TreeB-Occ and BP-TreeB-CaOC considerably surpassed the other curves. However, we notice a good early behavior of the methods except for Grad-CAM, Ms-IV and Occlusion.

Based on Figure 6, the methods BP-TreeB-CaOC, BP-TreeB-Occ, XRAI, and LIME achieved the highest AUC scores, in that order. When considering more restrictive levels of the hierarchy (using 0.5% and 1.0% thresholds), most methods—except for Grad-CAM, Ms-IV, and Occlusion—performed well. *xAiTrees* configurations, along with LIME and XRAI, effectively identified the most important regions, with **C2** slightly outperforming **C1**. However, **C2**'s greater variation in PIR results (Table 2) suggested that it

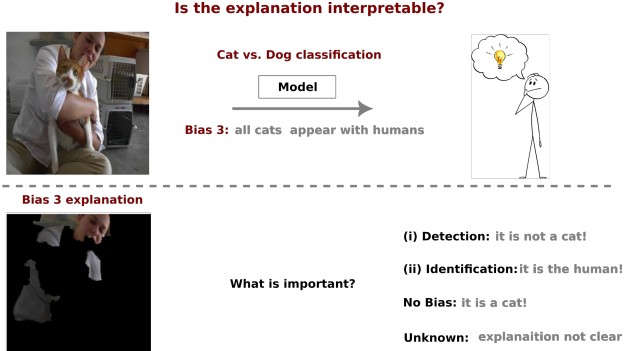

Figure 7: The objective of this qualitative experiment is to evaluate the interpretability of the methods under biased conditions. We ask participants to identify what appears to be the important part in each explanation, and we categorize their responses as follows: (i) the participant **detected** the presence of a bias, (ii) the participant **identified** the bias, (iii) the bias was **not detected**, or (iv) the explanation was unclear (**unknown**).

often highlighted larger regions, diluting per-pixel impact. To support qualitative experiments and simplify human analysis, we chose the **C1** group to minimize region size. For human analysis, we did not filter the explanations, but we indicated the hierarchy levels by region brightness, making distinctions clear.

## 5.2 Qualitative analysis

As qualitative experiments, we want to visually evaluate the explanations for different interpretability tasks. In this section, we perform experiments to (i) identify reasons for misclassification of images, and (ii) evaluate explanations through the human interpretation of biased-trained networks.

**Comparison of misclassified images:** We searched for examples that were misclassified by models (VGG-16 and Resnet18) trained on ImageNet (Deng et al., 2009). Figure 2 shows the explanations generated by Integrated Gradients, Grad-CAM, Occlusion, LIME, and Tree-Occ (500 pixels minimal region) of six images incorrectly classified. In Figure 2, the first column displays classes (such as chime, fence, dishwasher, among others) alongside examples of misclassified images. These images should have been classified as cat or dog. We then apply methods used in previous quantitative comparisons to generate visual explanations for why these images were misclassified. The figure illustrates that methods like Integrated Gradients, Grad-CAM, and Occlusion (Occ) may cause confusion in precisely identifying what caused the misclassification and may lead to poor human interpretation (we properly evaluate this in next experiment **Human evaluation in bias analysis**). Although LIME and our proposed Tree-Occ method can pinpoint interesting regions, the Tree-Occ method better illustrates the motivation behind misclassified results, as evident in the last column. For instance, in the fence example, it highlights the diamond pattern found on fences, while in the dishwasher example, it focuses solely on the sink region, disregarding the cat. Considering the hierarchical characteristic of our methodology, we can perform a deeper analysis of the explanations by selecting regions by the percentage of importance to be visualized. Examples in A.6.

**Human evaluation in bias analysis:** As previously mentioned, we used the configuration **C1** for human-interpretation evaluation (Figure 7). We trained three Resnet18 models subjected to data bias: (a) **Bias 1** – a model trained with dogs and only cats on cushions; (b) **Bias 2** – a model with cats and only dogs with grids; (c) **Bias 3** – and a model with dogs and only cats with humans (details of validation accuracy and visualizations in A.6). We presented the same five image visualizations (from corrected classified images by the biased class) for the baseline methods and the methods from **C1**. We intended to verify if: (i) humans can detect the wrong focus given based on a class prediction (**Detection**); and (ii) humans can recognize which was the cause of the bias (**Identification**).

To test (i) and (ii), for each **Bias** (a,b, or c) type we produce for each of the xAI methods an explanation image. By presenting five image explanations (the same images) for each of the xAI methodologies, we asked

volunteers, based on the explanations provided, what they think the highlighted regions referred to (generated explanations and extra experiments in A.6).

Table 4: Human evaluation assessed bias detection and identification using five image explanations from biased datasets: (1) dogs with only cats on cushions, (2) cats with only dogs on grids, and (3) dogs with only cats and humans. Metrics included bias detection (**Detection (i)**), bias identification (**Identification (ii)**), misunderstanding of explanations (**Not identified**), and focus on animals (**Animal**). Higher detection and identification rates were desired, with lower misunderstanding rates. Concept-aware methods like Ms-IV, ACE, and Tree-CaOC performed better, with Tree-CaOC using human-based segmentation achieving the best results across all biased datasets.

| | | IG | Grad-CAM | Occ | LIME | Ms-IV | ACE | Tree-Occ | Tree-CaOC | IG-Tree-Occ | IG-Tree-CaOC |
|---|---|---|---|---|---|---|---|---|---|---|---|
| **Bias 1** | **Detection (i)** | 24.4 | 0.0 | **31.7** | 14.6 | **31.7** | 0.0 | 19.5 | **47.4** | 12.5 | 0.0 |
| | **Identification (ii)** | 12.2 | 0.0 | 9.7 | 7.3 | **14.6** | 0.0 | 2.4 | **21.1** | 0.0 | 0.0 |
| | **No bias** | 41.5 | 97.6 | 31.7 | 48.8 | 41.4 | 70.0 | 56.1 | 26.3 | 52.5 | 87.5 |
| | **Unknown** | 34.1 | 2.4 | 36.6 | 36.6 | 26.8 | 30.0 | 24.4 | 26.3 | 35.0 | 12.5 |
| **Bias 2** | **Detection (i)** | 7.3 | 0.0 | 0.0 | 7.3 | 24.4 | **58.5** | 56.4 | **57.9** | 42.1 | 29.2 |
| | **Identification (ii)** | 2.4 | 0.0 | 0.0 | 7.3 | 19.5 | **43.9** | 41.0 | **47.4** | 34.2 | 26.8 |
| | **No bias** | 81.1 | 100.0 | 86.6 | 61.0 | 48.8 | 2.4 | 12.8 | 10.5 | 23.7 | 34.1 |
| | **Unknown** | 12.2 | 0.0 | 17.1 | 31.7 | 26.8 | 39.0 | 30.8 | 31.6 | 34.2 | 36.6 |
| **Bias 3** | **Detection (i)** | 35.0 | 17.1 | **43.9** | **51.2** | 29.3 | 9.8 | 41.1 | **56.4** | 50.0 | **55.0** |
| | **Identification (ii)** | 19.5 | 12.2 | **32.4** | **36.6** | 17.1 | 4.9 | 10.3 | **33.3** | 22.5 | **37.5** |
| | **No bias** | 29.3 | 70.8 | 29.3 | 4.9 | 24.4 | 26.9 | 7.7 | 5.2 | 2.5 | 2.5 |
| | **Unknown** | 36.6 | 12.2 | 26.8 | 43.9 | 46.3 | 63.4 | 51.3 | 38.5 | 47.5 | 42.5 |

Table 4 presents the results of evaluating 41 individuals from diverse continents (South America, Europe, and Asia), fields (Human, Biological, and Exact sciences), and levels of AI expertise (ranging from no knowledge to expert, with over half being non-experts). We show some participants' statistics in A.6. The experiment aims to identify effective methods for revealing trained-with biases. For each xAI method (IG, Grad-CAM, Occ, LIME, Ms-IV, ACE, Tree-Occ, Tree-CaOC, IG-Tree-Occ, IG-Tree-CaOC) used to explain biases (1, 2, and 3), we show the percentage of participants who detected, identified, or did not identify the bias in the explanation. **Detection** indicates perceiving the xAI explanation as either background or reflecting the bias, while **Identification** denotes successful interpretation of the explanation as the induced bias. **Not Identification** refers to being unable to interpret the explanation. Higher percentages in the Identification row are desirable. If not, we prioritize high values in the Detection row. Lower values in the Not Identification or **Animal** rows indicate clearer human interpretation of our trained-with bias.

Table 4 shows that IG and Grad-CAM struggled with interpretability, leading to many **Not identified** or **Animal** responses, indicating unclear explanations for imposed biases. Methods leveraging contextual or global explanations, such as Ms-IV, ACE, Tree-CaOC, and IG-Tree-CaOC, achieved better detection and identification results due to their global approach to model knowledge. However, this was not always enough for humans to provide a complete interpretation of the model's knowledge. Tree-CaOC, combining global metrics (CaOC) and human-based segmentation, consistently achieved the best results for all three **Bias** categories, highlighting its superiority in enhancing human interpretability.

# 6 Limitations

The computational time is a limitation when using time-expensive methods to attribute region scores, such as LIME. We show the time comparison including the baseline methods in A.5 Table 15. That is why we limited our analysis to Tree-Occ and Tree-CaOC.

The analysis was limited to a specific task and dataset: classification of cats and dogs. Applying the method to other tasks, such as representation learning, or to different modalities, such as text, requires adaptations, which are discussed in Section A.7.

The proposed version of *xAiTrees* framework is dependent on the base methods used. Therefore, by using an edge-based segmentation method, we will not obtain a semantic-based explanation, *i.e.*, the final technique will inherit the limitations of the base methods. Future works will be focused on semantic segmentation.

## 7   Conclusion

In this paper, we present a framework, *xAiTrees*, aimed at integrating multiscale region importance in model predictions, providing more faithful and interpretable explanations. Our approach outperforms traditional xAI methods like LIME, especially in identifying impactful and precise regions, in datasets such as Cat vs. Dog, CIFAR-10, and ImageNet. Qualitative analysis demonstrates that our Tree-Occ method better elucidates misclassification motivations and provides clearer, hierarchical interpretations of model predictions. Techniques like Tree-CaOC, merging global-aware metrics with human-based segmentation, excel in detection and identification tasks, achieving superior results in human interpretability. In summary, our framework delivers highly interpretable and faithful model explanations, significantly aiding in bias detection and identification, and demonstrating its effectiveness in the field of explainable AI. Therefore, potentially aiding to reduce the societal negative impact that could be generated by deep learning models in high-risk decision-making process.

### Reproducibility Statement

We release the code at [https://github.com/CarolMazini/reasoning_with_trees/](https://github.com/CarolMazini/reasoning_with_trees/) and detail our experimental setup and disclose all hyperparameters in the Appendix.

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

# A    Appendix

## A.1    A. Segmentation techniques

As an important step for our framework, we employ segmentation techniques so we can decompose images, based on specific attributes, into more interpretable structures, enabling better human understanding and interpretation. We specifically employ hierarchical segmentation techniques due to their capability to decompose images into multiple levels of detail, mirroring how humans naturally perceive objects: initially observing the overall structure before delving into the finer details.

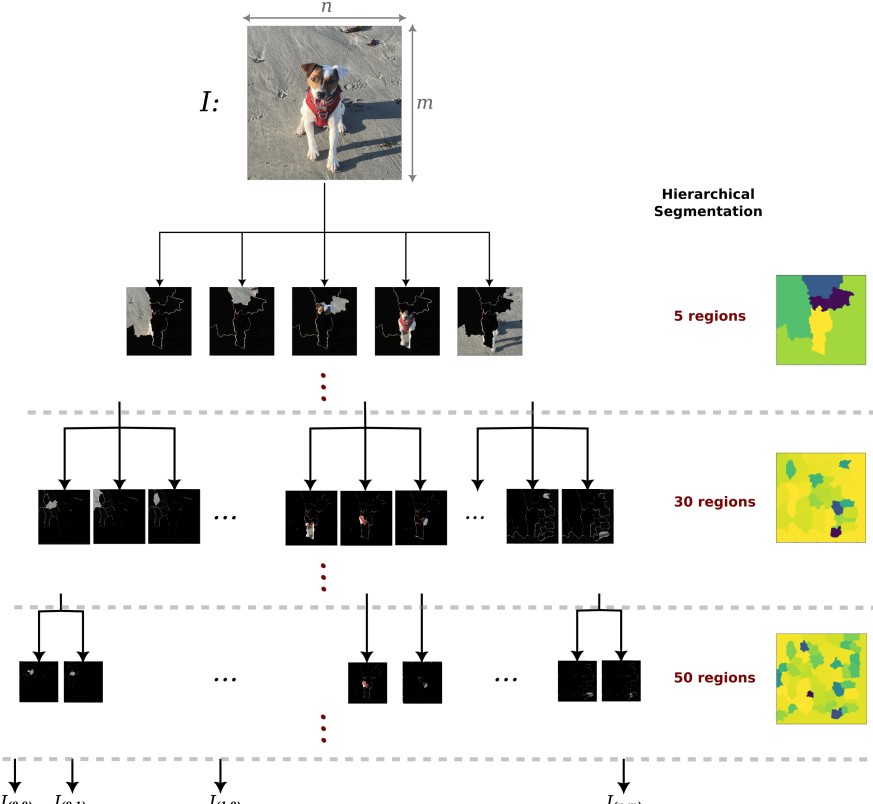

Figure 8: An example of a hierarchical segmentation tree, where the root represents the entire image and each subsequent level recursively divides it into an increasing number of subregions, reaching the leaf nodes that correspond to individual pixels.

**Trees:** A tree is an acyclic graph, consisting of nodes that connect to zero or more other nodes. It starts with a "root" node that branches out to other nodes, ending in "leaves" with no children. In image representation, the root node represents the entire image, and each leaf represents a pixel, resulting in as many leaves as pixels. The structure between the root and leaves groups pixels into clusters at each level based on similarity metrics, with each level abstracting the one below. Using a segmentation tree, we can make *cuts* at various levels to obtain different numbers and sizes of segmented regions. We present an example in Figure 8.

**Binary Partition Tree (BPT):** A Binary Partition Tree (BPT) is a data structure in which each node represents a region of the image. Similarly, the tree starts with a root node representing the entire image and branches out through a series of binary splits until reaching the leaf nodes, representing the individual pixels. Different from the tree, in which a node could have multiple splits, in the BPT each split, divides a region into two smaller sub-regions based on a criterion.

**Watershed:** This algorithm (Cousty et al., 2008) constructs a hierarchical segmentation tree based on a minimum-spanning forest rooted in the local minima of an edge-weighted graph. In this context, local minima are points in the graph where the surrounding edge weights are higher, representing the lowest values in their neighborhood. These minima serve as starting points for the segmentation. The algorithm iteratively merges regions beginning from these local minima, guided by the edge weights that indicate dissimilarity between adjacent pixels. By progressively combining these regions, the algorithm builds the segmentation tree, effectively capturing the hierarchical structure of the image.

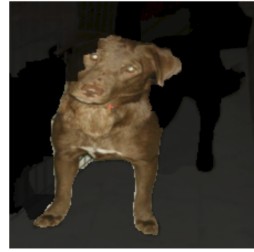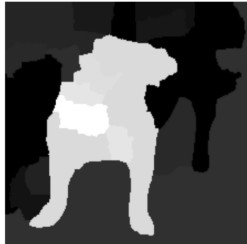

Figure 9: Explanation obtained using Tree-LIME – *xAiTrees* combined to LIME instead of occlusion to score regions. We show here the adaptability of the framework to different xAI techniques. Due to the high time consumption of Tree-LIME (A.5 Table 15), we present some preliminary results in A.7.

## A.2  B. Occlusion-based metrics:

Here, we discuss the metrics used to generate our segmentation based on the model explainability (block **B** in Figure 3). We present two metrics: (i) *Occlusion*, which is the impact of occluding an image region on its classification output, and (ii) CaOC which is the intra-class impact of occluding an image region. For (i), we assess how the output of a model changes when an image region is occluded. For (ii), we employ a sliding metric that ranks images based on the highest activations for a given class. We then measure the movement in this ranking after occluding a region of the image, determining the intra-class impact of the occlusion.

***Occlusion:*** Let us say we have a model $\Xi$ producing an output $\mathbf{out}_i$ for an image $\mathcal{I}_i$. By concealing portions of this image, creating a new image $\mathcal{I}_i^{\blacksquare}$, we obtain a different model output $\mathbf{out}_i^{\blacksquare}$. The significance of the occluded area concerning a particular class $c$ is assessed by comparing the outputs:

$$\left| \mathbf{out}_{i,c} - \mathbf{out}_{i,c}^{\blacksquare} \right|. \tag{1}$$

If there is a significant difference, it indicates that the model strongly relies on this region for class activation, meaning that these regions have a high *impact* on the model's decision.

***CAOC:*** In the Ms-IV method, introduced by Rodrigues *et al.* (Rodrigues et al., 2024), CaOC employs rankings to assess how occlusions affect the model's output space. A *ranking* is a sequence of objects ordered according to a specific criterion, from the object most aligned with it to the least aligned. Suppose the criterion is to maximize class $c$. In that case, the first index $i$ in this sequence represents the object (in our case, the image $\mathcal{I}_i$) with the highest activation for class $c$ in the output $\mathbf{out}_i$. If we define a function *argsort* to obtain the indices of an ordered sequence of objects, we can derive the sequence of image indices that maximize class $c$: $Seq_c = argsort\left(\mathbf{out}_{.,c}, decreasing\right)$, with $\mathbf{out}_{.,c}$ the vector of outputs for class $c$ of a set of input images $(\mathcal{I}_i)_{i \in [1, NbIm]}$.

CaOC computes an initial ranking $Seq_c$ for a subset of images $\mathcal{DS}' \subset \mathcal{DS}$, and then a subsequent ranking $Seq_c'$ after occluding one region of image $\mathcal{I}_i \in \mathcal{DS}'$. The significance of this occluded image region for the model is determined by the difference in the positions of this image in the rankings given by $\left| position\left(Seq_{i,c}, \mathcal{I}_i\right) - position\left(Seq_{i,c}', \mathcal{I}_i\right) \right|$.

This metric aims to assess the impact of occluding image regions not only against the original output $\mathbf{out}_i$ but also against the outputs of a range of images. Incorporating the model's output space into the analysis ensures that explanations consider the broader context (global model's behavior). Hence, we can characterize it as globally aware, even when explaining a single sample.

Although the main experimentation uses these methods, we present in A.7 a framework's variation using LIME to show *xAiTrees* versatility with other types of xAI techniques. Explanations example in Figure 9.

### A.3 Tested framework's configuration

We tested four different sizes of minimal region for filtering the initial segmentation. For Cat vs. Dog and ImageNet datasets: 200, 300, 400, and 500 pixels. For CIFAR-10: 4, 16, 32 and 64 pixels.

For the model-based segmentation and datasets Cat vs. Dog and CIFAR10, we tested four xAI techniques to generation the initial graph **G** on Figure 3: Integrated Gradients (IG), Guided-Backpropagation (BP), Input X Gradient (I X G), and Saliency (S). Given the number of images (time of computation), for ImageNet dataset we tested only Integrated Gradients (IG), Guided-Backpropagation (BP).

We tested three algorithms to construct the hierarchical segmentation: Binary Partition Tree (BPT), Watershed with Area, and Watershed with Volume.

We tested two different occlusion based metrics to obtain the impact of regions used to shape the hierarchical tree: CaOC and OCC.

When we refer to **Tree-CaOC** or **TreeW-CaOC**, we mean the human-based segmentation (edges' map) using Watershed area and CaOC as occlusion metric. When we refer to **IG-Tree-Occ** or **IG-TreeW-Occ**, we mean the model-based segmentation (using Integrated Gradients (IG) attributions) using Watershed area and Occ (simple occlusion – Equation (1)) as occlusion metric. When we refer to **BP-TreeB-Occ**, we mean the model-based segmentation (using Guided Backpropagation (BP) attributions) using BPT and Occ as occlusion metric.

### A.4 Parameters of the baseline methods

For Grad-CAM method, we used the last convolutional layer of each architecture with layer Grad-CAM from captum framework. For Occlusion (from Captum framework) we used, for Cat vs. Dog and ImageNet, step of 3x7x7 and sliding window of 3x14x14. Since CIFAR10 is much smaller, the step was 3x2x2 and sliding window of 3x4x4. For LIME, we used the standard configuration for Cat vs. Dog and ImageNet (Quickshift kernel size of 4) and, Quickshift kernel size of 2 for CIFAR10. All the other methods followed the standard configuration. For Ms-IV, we used the original configuration from the paper (Rodrigues et al., 2024). For XRAI, we used the original implementation (Kapishnikov et al., 2019) of the fast version.

### A.5 Quantitative evaluations

**Models' description:** Table 5 shows the number of images in train and validation sets for Cat vs. Dog and CIFAR10 datasets. We also include the train and validation accuracies for the models ResNet18 and VGG-16 used in the quantitative evaluations.

Cat vs. Dog models were trained with initial weights from ImageNet, learning rate $1e - 7$, cross-entropy loss, the Adam optimizer, and early stop in 20 epochs of non-improving validation loss.

CIFAR10 models were adapted to receive 32x32 input images, and they were trained with initial weights from ImageNet, learning rate $1e - 2$, cross-entropy loss and the stochastic gradient descent optimizer (code from Phan (2021)).

The ImageNet models used the pre-trained weights from the PyTorch implementation.

Table 5: Number of images and accuracy on train and validation sets for ResNet18 and VGG-16 models. We train the models with two different dataset: Cat vs. Dog and CIFAR10.

| | | Train | | Val. | |
|---|---|---|---|---|---|
| | | Num. images | Acc. (%) | Num. images | Acc. (%) |
| **Cat vs. Dog** | **ResNet18** | 19,891 | 98.21 | 5,109 | 97.86 |
| | **VGG-16** | | 99.04 | | 98.61 |
| **CIFAR10** | **ResNet18** | 50,000 | 99.56 | 10,000 | 92.53 |
| | **VGG-16** | | 99.84 | | 93.54 |

Table 6: Percentage of images with the original class changed after the **exclusion** of selected explanation regions for Cat vs. Dog dataset. Highlighted in blue are the configurations presented in the main paper. We tested hierarchies constructed by filtering out smaller regions than 200, 300, 400 and 500 pixels, segmentation based on **Edges**, Integrated Gradients (**IG**), Guided-Backpropagation (**BP**), Input X Gradients (**I X G**) and **Saliency**. We tested three different strategies to for the first hierarchical segmentation: BPT, watershed with area attribute, and watershed with volume attribute. **Same** column shows images maintaining the original class when the output was reduced, and **Total** is the sum of class change (**Ch.**) and class reduction (**Same**).

| % of images | | | VGG Edges Ch. | VGG Edges Same | VGG IG Ch. | VGG IG Same | VGG BP Ch. | VGG BP Same | VGG I X G Ch. | VGG I X G Same | VGG Saliency Ch. | VGG Saliency Same | ResNet Edges Ch. | ResNet Edges Same | ResNet IG Ch. | ResNet IG Same | ResNet BP Ch. | ResNet BP Same | ResNet I X G Ch. | ResNet I X G Same | ResNet Saliency Ch. | ResNet Saliency Same |
|---|---|---|---|---|---|---|---|---|---|---|---|---|---|---|---|---|---|---|---|---|---|---|
| 200 | BPT | CaOC | 0.35 | 0.39 | 0.32 | 0.15 | 0.56 | 0.32 | 0.26 | 0.12 | 0.46 | 0.38 | 0.27 | 0.42 | 0.12 | 0.29 | 0.39 | 0.39 | 0.10 | 0.21 | 0.28 | 0.43 |
| 200 | BPT | Occ | 0.51 | 0.44 | 0.33 | 0.23 | 0.63 | 0.35 | 0.27 | 0.19 | 0.64 | 0.35 | 0.41 | 0.52 | 0.17 | 0.34 | 0.55 | 0.37 | 0.13 | 0.30 | 0.39 | 0.55 |
| 200 | Watershed area | CaOC | 0.15 | 0.48 | 0.23 | 0.45 | 0.21 | 0.46 | 0.22 | 0.46 | 0.15 | 0.49 | 0.20 | 0.44 | 0.18 | 0.45 | 0.16 | 0.50 | 0.17 | 0.46 | 0.17 | 0.46 |
| 200 | Watershed area | Occ | 0.30 | 0.66 | 0.42 | 0.55 | 0.41 | 0.56 | 0.42 | 0.55 | 0.31 | 0.62 | 0.34 | 0.61 | 0.31 | 0.63 | 0.32 | 0.64 | 0.30 | 0.63 | 0.31 | 0.63 |
| 200 | Watershed volume | CaOC | 0.18 | 0.48 | 0.16 | 0.49 | 0.12 | 0.46 | 0.18 | 0.47 | 0.13 | 0.50 | 0.18 | 0.44 | 0.17 | 0.46 | 0.12 | 0.49 | 0.17 | 0.45 | 0.18 | 0.45 |
| 200 | Watershed volume | Occ | 0.33 | 0.64 | 0.33 | 0.61 | 0.31 | 0.64 | 0.37 | 0.57 | 0.32 | 0.63 | 0.33 | 0.62 | 0.29 | 0.65 | 0.25 | 0.69 | 0.30 | 0.65 | 0.32 | 0.64 |
| 300 | BPT | CaOC | 0.36 | 0.38 | 0.22 | 0.07 | 0.56 | 0.30 | 0.17 | 0.07 | 0.46 | 0.38 | 0.28 | 0.41 | 0.07 | 0.19 | 0.39 | 0.37 | 0.06 | 0.13 | 0.30 | 0.41 |
| 300 | BPT | Occ | 0.50 | 0.46 | 0.23 | 0.13 | 0.61 | 0.34 | 0.17 | 0.11 | 0.58 | 0.41 | 0.41 | 0.53 | 0.09 | 0.27 | 0.48 | 0.39 | 0.07 | 0.20 | 0.38 | 0.53 |
| 300 | Watershed area | CaOC | 0.16 | 0.49 | 0.25 | 0.43 | 0.22 | 0.45 | 0.24 | 0.46 | 0.15 | 0.50 | 0.20 | 0.42 | 0.20 | 0.44 | 0.17 | 0.47 | 0.18 | 0.47 | 0.19 | 0.45 |
| 300 | Watershed area | Occ | 0.30 | 0.65 | 0.43 | 0.53 | 0.42 | 0.55 | 0.42 | 0.55 | 0.31 | 0.63 | 0.35 | 0.59 | 0.30 | 0.63 | 0.32 | 0.62 | 0.30 | 0.63 | 0.31 | 0.62 |
| 300 | Watershed volume | CaOC | 0.17 | 0.51 | 0.18 | 0.49 | 0.15 | 0.43 | 0.19 | 0.48 | 0.14 | 0.51 | 0.20 | 0.41 | 0.18 | 0.44 | 0.12 | 0.48 | 0.18 | 0.45 | 0.18 | 0.44 |
| 300 | Watershed volume | Occ | 0.32 | 0.63 | 0.34 | 0.60 | 0.30 | 0.64 | 0.36 | 0.57 | 0.32 | 0.61 | 0.34 | 0.61 | 0.29 | 0.64 | 0.26 | 0.69 | 0.30 | 0.64 | 0.32 | 0.63 |
| 400 | BPT | CaOC | 0.36 | 0.39 | 0.15 | 0.05 | 0.51 | 0.29 | 0.10 | 0.05 | 0.43 | 0.40 | 0.29 | 0.42 | 0.04 | 0.16 | 0.35 | 0.37 | 0.04 | 0.11 | 0.29 | 0.41 |
| 400 | BPT | Occ | 0.47 | 0.48 | 0.15 | 0.09 | 0.54 | 0.37 | 0.10 | 0.08 | 0.51 | 0.47 | 0.41 | 0.52 | 0.06 | 0.23 | 0.42 | 0.39 | 0.05 | 0.14 | 0.36 | 0.55 |
| 400 | Watershed area | CaOC | 0.16 | 0.49 | 0.26 | 0.46 | 0.25 | 0.43 | 0.24 | 0.46 | 0.17 | 0.50 | 0.21 | 0.41 | 0.20 | 0.44 | 0.18 | 0.46 | 0.20 | 0.46 | 0.21 | 0.42 |
| 400 | Watershed area | Occ | 0.30 | 0.64 | 0.43 | 0.53 | 0.42 | 0.54 | 0.42 | 0.55 | 0.31 | 0.62 | 0.34 | 0.61 | 0.32 | 0.61 | 0.32 | 0.61 | 0.30 | 0.63 | 0.31 | 0.63 |
| 400 | Watershed volume | CaOC | 0.18 | 0.49 | 0.20 | 0.47 | 0.16 | 0.45 | 0.22 | 0.45 | 0.16 | 0.50 | 0.21 | 0.40 | 0.19 | 0.45 | 0.13 | 0.47 | 0.19 | 0.42 | 0.20 | 0.44 |
| 400 | Watershed volume | Occ | 0.33 | 0.63 | 0.35 | 0.60 | 0.30 | 0.64 | 0.38 | 0.57 | 0.31 | 0.62 | 0.34 | 0.61 | 0.30 | 0.64 | 0.25 | 0.69 | 0.29 | 0.63 | 0.33 | 0.63 |
| 500 | BPT | CaOC | 0.35 | 0.40 | 0.11 | 0.04 | 0.45 | 0.29 | 0.07 | 0.03 | 0.40 | 0.42 | 0.30 | 0.41 | 0.04 | 0.13 | 0.31 | 0.34 | 0.04 | 0.11 | 0.29 | 0.42 |
| 500 | BPT | Occ | 0.45 | 0.49 | 0.12 | 0.06 | 0.49 | 0.36 | 0.07 | 0.04 | 0.47 | 0.49 | 0.41 | 0.51 | 0.04 | 0.17 | 0.38 | 0.38 | 0.04 | 0.11 | 0.37 | 0.54 |
| 500 | Watershed area | CaOC | 0.16 | 0.48 | 0.29 | 0.46 | 0.26 | 0.42 | 0.25 | 0.47 | 0.18 | 0.48 | 0.22 | 0.41 | 0.21 | 0.45 | 0.20 | 0.46 | 0.20 | 0.46 | 0.21 | 0.42 |
| 500 | Watershed area | Occ | 0.31 | 0.63 | 0.43 | 0.54 | 0.41 | 0.55 | 0.41 | 0.54 | 0.31 | 0.63 | 0.35 | 0.60 | 0.32 | 0.61 | 0.34 | 0.61 | 0.30 | 0.62 | 0.30 | 0.66 |
| 500 | Watershed volume | CaOC | 0.19 | 0.48 | 0.20 | 0.47 | 0.16 | 0.44 | 0.22 | 0.45 | 0.18 | 0.51 | 0.22 | 0.39 | 0.20 | 0.46 | 0.14 | 0.47 | 0.18 | 0.43 | 0.22 | 0.43 |
| 500 | Watershed volume | Occ | 0.33 | 0.63 | 0.34 | 0.61 | 0.29 | 0.66 | 0.38 | 0.56 | 0.32 | 0.60 | 0.33 | 0.61 | 0.30 | 0.64 | 0.25 | 0.68 | 0.29 | 0.64 | 0.32 | 0.63 |

**Exclusion of important regions:** Given that each region-based explainable AI (xAI) method identifies important regions in an image to explain the prediction of a model, we performed occlusion of these regions to measure the impact of each selection and evaluate the methods.

Except for LIME, which proposes a ranking of the most important image segments, the methods we used as a baseline provide values for measuring the importance of pixels, which in visualization is similar to regions or segmentation for humans. However, if we select all the positive importance values provided by these methods, we are likely to cover a large part of the image. By selecting regions with only the top 25% higher values, we reduce the size of the mask. In fact, different datasets and models will show different visualization behaviors, so we chose to define a high threshold that is fixed (not specific to a single dataset) and common to all methods in order to have a fair comparison. Therefore, for methods that assign scores to regions, we masked the 25% highest scores.

We present, in Tables 6, 7, and 8, the complete experiments of different configurations of our framework for the datasets Cat vs. Dog, CIFAR10, and ImageNet respectively.

**PIR values:** To address the issue of unhelpful explanations resulting from methods selecting the entire image as important, potentially leading to class changes upon occlusion, we introduce a novel metric termed **Pixel Impact Rate (PIR)**. The idea of PIR is to evaluate the impact per pixel/per image:

$$PIR(exp_i) = \frac{\left| \mathbf{out}_{i,class\_orig} - \mathbf{out}^{\blacksquare}_{i,class\_orig} \right|}{num\_pixels\_exp} \quad (2)$$

where $exp_i$ is the explanation or image $i$, $\mathbf{out}_{i,class\_orig}$ is the original *logit* corresponding to the analyzed class, $\mathbf{out}^{\blacksquare}_{i,class\_orig}$ is the *logit* after the perturbation, and $num\_pixels\_exp$ is the number of occluded pixels.

This metric quantifies the impact on class activation per occluded pixel. Complementing the percentage of class change, PIR distinguishes whether changes are primarily caused by complete or near-complete occlusion of the image. Higher PIR values indicate that each occluded pixel has a significant average impact, suggesting

Table 7: Percentage of images with the original class changed after the **exclusion** of selected explanation regions for CIFAR10 dataset. Highlighted in blue are the configurations presented in the main paper. We tested hierarchies constructed by filtering out smaller regions than 4, 16, 32 and 64 pixels, segmentation based on **Edges**, Integrated Gradients (**IG**), Guided-Backpropagation (**BP**), Input X Gradients (**I X G**) and **Saliency**. We tested three different strategies to for the first hierarchical segmentation: BPT, watershed with area attribute, and watershed with volume attribute. **Same** column shows images maintaining the original class when the output was reduced, and **Total** is the sum of class change (**Ch.**) and class reduction (**Same**).

| % of images | | | VGG Edges Ch. | VGG Edges Same | VGG IG Ch. | VGG IG Same | VGG BP Ch. | VGG BP Same | VGG I X G Ch. | VGG I X G Same | VGG Saliency Ch. | VGG Saliency Same | ResNet Edges Ch. | ResNet Edges Same | ResNet IG Ch. | ResNet IG Same | ResNet BP Ch. | ResNet BP Same | ResNet I X G Ch. | ResNet I X G Same | ResNet Saliency Ch. | ResNet Saliency Same |
|---|---|---|---|---|---|---|---|---|---|---|---|---|---|---|---|---|---|---|---|---|---|---|
| 4 | BPT | CaOC | 0.15 | 0.26 | 0.08 | 0.34 | 0.09 | 0.34 | 0.06 | 0.34 | 0.18 | 0.37 | 0.17 | 0.31 | 0.09 | 0.40 | 0.11 | 0.39 | 0.08 | 0.40 | 0.19 | 0.42 |
| | | Occ | 0.81 | 0.12 | 0.91 | 0.08 | 0.88 | 0.10 | 0.91 | 0.07 | 0.85 | 0.14 | 0.76 | 0.17 | 0.83 | 0.13 | 0.80 | 0.16 | 0.82 | 0.14 | 0.81 | 0.18 |
| | Watershed area | CaOC | 0.16 | 0.34 | 0.27 | 0.32 | 0.27 | 0.33 | 0.27 | 0.33 | 0.26 | 0.33 | 0.18 | 0.40 | 0.24 | 0.37 | 0.24 | 0.37 | 0.25 | 0.37 | 0.25 | 0.36 |
| | | Occ | 0.72 | 0.22 | 0.74 | 0.23 | 0.75 | 0.22 | 0.76 | 0.22 | 0.73 | 0.24 | 0.70 | 0.25 | 0.74 | 0.24 | 0.73 | 0.25 | 0.76 | 0.23 | 0.74 | 0.25 |
| | Watershed volume | CaOC | 0.16 | 0.34 | 0.26 | 0.33 | 0.27 | 0.33 | 0.26 | 0.34 | 0.26 | 0.34 | 0.18 | 0.40 | 0.24 | 0.37 | 0.23 | 0.38 | 0.25 | 0.37 | 0.24 | 0.37 |
| | | Occ | 0.73 | 0.21 | 0.73 | 0.24 | 0.72 | 0.25 | 0.75 | 0.23 | 0.72 | 0.25 | 0.71 | 0.24 | 0.74 | 0.25 | 0.72 | 0.26 | 0.75 | 0.23 | 0.74 | 0.25 |
| 16 | BPT | CaOC | 0.08 | 0.11 | 0.07 | 0.11 | 0.09 | 0.17 | 0.05 | 0.09 | 0.18 | 0.31 | 0.12 | 0.12 | 0.09 | 0.14 | 0.11 | 0.20 | 0.07 | 0.11 | 0.21 | 0.38 |
| | | Occ | 0.55 | 0.09 | 0.70 | 0.07 | 0.78 | 0.10 | 0.63 | 0.06 | 0.79 | 0.17 | 0.55 | 0.09 | 0.64 | 0.09 | 0.69 | 0.13 | 0.57 | 0.09 | 0.76 | 0.21 |
| | Watershed area | CaOC | 0.16 | 0.33 | 0.22 | 0.35 | 0.22 | 0.35 | 0.22 | 0.35 | 0.22 | 0.35 | 0.18 | 0.39 | 0.25 | 0.38 | 0.23 | 0.39 | 0.25 | 0.37 | 0.25 | 0.37 |
| | | Occ | 0.71 | 0.21 | 0.75 | 0.22 | 0.75 | 0.22 | 0.76 | 0.21 | 0.72 | 0.24 | 0.70 | 0.24 | 0.75 | 0.24 | 0.74 | 0.25 | 0.76 | 0.23 | 0.74 | 0.24 |
| | Watershed volume | CaOC | 0.16 | 0.33 | 0.22 | 0.35 | 0.21 | 0.36 | 0.23 | 0.34 | 0.22 | 0.35 | 0.19 | 0.39 | 0.24 | 0.38 | 0.22 | 0.40 | 0.26 | 0.37 | 0.25 | 0.37 |
| | | Occ | 0.72 | 0.21 | 0.73 | 0.24 | 0.72 | 0.24 | 0.75 | 0.22 | 0.72 | 0.24 | 0.71 | 0.24 | 0.74 | 0.24 | 0.71 | 0.27 | 0.75 | 0.23 | 0.74 | 0.24 |
| 32 | BPT | CaOC | 0.03 | 0.03 | 0.04 | 0.04 | 0.07 | 0.09 | 0.03 | 0.03 | 0.16 | 0.17 | 0.07 | 0.05 | 0.07 | 0.05 | 0.10 | 0.11 | 0.06 | 0.04 | 0.18 | 0.22 |
| | | Occ | 0.39 | 0.03 | 0.44 | 0.05 | 0.62 | 0.09 | 0.40 | 0.03 | 0.70 | 0.15 | 0.40 | 0.04 | 0.44 | 0.05 | 0.56 | 0.09 | 0.38 | 0.05 | 0.70 | 0.18 |
| | Watershed area | CaOC | 0.15 | 0.28 | 0.21 | 0.36 | 0.19 | 0.37 | 0.20 | 0.36 | 0.21 | 0.35 | 0.18 | 0.34 | 0.23 | 0.40 | 0.21 | 0.41 | 0.23 | 0.40 | 0.23 | 0.40 |
| | | Occ | 0.69 | 0.20 | 0.75 | 0.22 | 0.74 | 0.22 | 0.75 | 0.21 | 0.72 | 0.24 | 0.68 | 0.22 | 0.74 | 0.23 | 0.73 | 0.25 | 0.75 | 0.23 | 0.73 | 0.25 |
| | Watershed volume | CaOC | 0.16 | 0.29 | 0.20 | 0.36 | 0.19 | 0.37 | 0.20 | 0.36 | 0.21 | 0.35 | 0.18 | 0.34 | 0.23 | 0.40 | 0.21 | 0.41 | 0.23 | 0.40 | 0.23 | 0.40 |
| | | Occ | 0.70 | 0.20 | 0.72 | 0.24 | 0.70 | 0.25 | 0.74 | 0.22 | 0.72 | 0.24 | 0.69 | 0.22 | 0.73 | 0.25 | 0.69 | 0.28 | 0.75 | 0.23 | 0.73 | 0.25 |
| 64 | BPT | CaOC | 0.01 | 0.00 | 0.02 | 0.01 | 0.04 | 0.04 | 0.01 | 0.01 | 0.07 | 0.05 | 0.04 | 0.02 | 0.05 | 0.02 | 0.08 | 0.05 | 0.04 | 0.02 | 0.11 | 0.08 |
| | | Occ | 0.25 | 0.00 | 0.23 | 0.02 | 0.42 | 0.05 | 0.19 | 0.01 | 0.50 | 0.06 | 0.27 | 0.02 | 0.25 | 0.02 | 0.38 | 0.05 | 0.21 | 0.02 | 0.53 | 0.07 |
| | Watershed area | CaOC | 0.12 | 0.16 | 0.20 | 0.29 | 0.18 | 0.27 | 0.20 | 0.28 | 0.22 | 0.30 | 0.15 | 0.19 | 0.23 | 0.33 | 0.21 | 0.34 | 0.22 | 0.34 | 0.22 | 0.36 |
| | | Occ | 0.60 | 0.15 | 0.73 | 0.21 | 0.73 | 0.21 | 0.75 | 0.20 | 0.70 | 0.23 | 0.60 | 0.15 | 0.73 | 0.22 | 0.72 | 0.23 | 0.74 | 0.22 | 0.70 | 0.25 |
| | Watershed volume | CaOC | 0.12 | 0.16 | 0.19 | 0.31 | 0.18 | 0.30 | 0.20 | 0.29 | 0.21 | 0.28 | 0.15 | 0.18 | 0.22 | 0.36 | 0.21 | 0.37 | 0.22 | 0.35 | 0.22 | 0.34 |
| | | Occ | 0.60 | 0.14 | 0.71 | 0.22 | 0.68 | 0.23 | 0.73 | 0.21 | 0.70 | 0.22 | 0.60 | 0.15 | 0.71 | 0.24 | 0.67 | 0.27 | 0.73 | 0.23 | 0.71 | 0.24 |

that concealing larger portions or the entire image leads to lower PIR, indicating less precision in the concealed area.

Tables 9, 10, and 11 display for each network, and tested configurations of our framework, the average (**avg**) and standard deviation (**std**) of PIR, for the datasets Cat vs. Dog, CIFAR10, and ImageNet respectively.

**Inclusion of important regions:** Additional experimentation was conducted to demonstrate a method's capability to identify an image region with sufficient information for the original class. The goal of this experiment is to determine whether the selected important region, when the only one left unoccluded in the image, can maintain the classification in its expected class. This experiment elucidates the critical role of these identified regions, providing strong evidence that they indeed contain essential information for accurate classification. We occluded **all regions** in the images except for the one selected by each method. We then calculated the percentage of images that changed class. We present the results from the three datasets, Cat vs. Dog, CIFAR10, and ImageNet, and all the tested framework configurations in Tables 12, 13, and 14, respectively. Lower percentages indicate better performance, meaning that a smaller percentage of images changed class, demonstrating that the chosen regions were sufficient to preserve the class for most of the images.

**SIC and AIC for hierarchy evaluation:** Inspired by the metrics Softmax Information Curve (SIC) and Accuracy Information Curve (AIC) proposed by Kapishnikov et al. (2019) we calculated the Softmax and Accuracy curves by including only selected image regions as model input. We used the parameters from the original paper: maintaining 10% of the original pixels and using linear interpolation to generate the blur. We used the thresholds of $0.5\%, 1\%, 2\%, 3\%, 4\%, 5\%, 7\%, 10\%, 13\%, 21\%, 34\%, 50\%$, and $75\%$ percent, representing the most significant region values according to each evaluated xAI method. Instead of using the image entropy values as the x-axis we used the thresholds. Figure 10 presents the curves for the mean of 1,000 randomly selected ImageNet images, and Figure 11 presents the results for 512 analyzed images from the Cat vs. Dog dataset.

Table 8: Percentage of images with the original class changed after the **exclusion** of selected explanation regions for Imagenet dataset. Highlighted in blue are the configurations presented in the main paper. We tested hierarchies constructed by filtering out smaller regions than 200, 300, 400 and 500 pixels, segmentation based on **Edges**, Integrated Gradients (**IG**), and Guided-Backpropagation (**BP**). We tested three different strategies to for the first hierarchical segmentation: BPT, watershed with area attribute, and watershed with volume attribute. **Same** column shows images maintaining the original class when the output was reduced, and **Total** is the sum of class change (**Ch.**) and class reduction (**Same**).

| % of images | | | Imagenet VGG Edges Ch. | Same | IG Ch. | Same | BP Ch. | Same | ResNet Edges Ch. | Same | IG Ch. | Same | BP Ch. | Same |
|---|---|---|---|---|---|---|---|---|---|---|---|---|---|---|
| **200** | BPT | **CaOC** | 0.23 | 0.49 | 0.00 | 0.02 | 0.11 | 0.36 | 0.25 | 0.44 | 0.01 | 0.04 | 0.10 | 0.31 |
| | | **Occ** | 0.57 | 0.38 | 0.35 | 0.01 | 0.71 | 0.16 | 0.60 | 0.35 | 0.50 | 0.02 | 0.74 | 0.13 |
| | Watershed area | **CaOC** | 0.26 | 0.51 | 0.27 | 0.51 | 0.27 | 0.51 | 0.27 | 0.45 | 0.31 | 0.45 | 0.30 | 0.45 |
| | | **Occ** | 0.42 | 0.55 | 0.44 | 0.54 | 0.45 | 0.52 | 0.48 | 0.50 | 0.50 | 0.47 | 0.52 | 0.46 |
| | Watershed volume | **CaOC** | 0.26 | 0.51 | 0.26 | 0.51 | 0.25 | 0.52 | 0.28 | 0.46 | 0.31 | 0.44 | 0.28 | 0.46 |
| | | **Occ** | 0.44 | 0.54 | 0.39 | 0.59 | 0.38 | 0.60 | 0.49 | 0.49 | 0.47 | 0.51 | 0.45 | 0.53 |
| **300** | BPT | **CaOC** | 0.22 | 0.46 | 0.00 | 0.01 | 0.10 | 0.30 | 0.24 | 0.42 | 0.01 | 0.02 | 0.09 | 0.25 |
| | | **Occ** | 0.55 | 0.38 | 0.20 | 0.01 | 0.64 | 0.17 | 0.58 | 0.35 | 0.32 | 0.01 | 0.65 | 0.13 |
| | Watershed area | **CaOC** | 0.24 | 0.51 | 0.27 | 0.52 | 0.26 | 0.51 | 0.27 | 0.45 | 0.31 | 0.45 | 0.29 | 0.45 |
| | | **Occ** | 0.41 | 0.56 | 0.43 | 0.54 | 0.44 | 0.53 | 0.46 | 0.51 | 0.50 | 0.48 | 0.51 | 0.46 |
| | Watershed volume | **CaOC** | 0.24 | 0.51 | 0.26 | 0.52 | 0.25 | 0.52 | 0.27 | 0.46 | 0.30 | 0.45 | 0.28 | 0.46 |
| | | **Occ** | 0.42 | 0.54 | 0.38 | 0.59 | 0.37 | 0.61 | 0.47 | 0.49 | 0.46 | 0.51 | 0.43 | 0.54 |
| **400** | BPT | **CaOC** | 0.21 | 0.42 | 0.00 | 0.01 | 0.09 | 0.25 | 0.23 | 0.38 | 0.00 | 0.02 | 0.08 | 0.21 |
| | | **Occ** | 0.54 | 0.37 | 0.13 | 0.00 | 0.58 | 0.16 | 0.57 | 0.34 | 0.22 | 0.01 | 0.59 | 0.13 |
| | Watershed area | **CaOC** | 0.24 | 0.51 | 0.26 | 0.52 | 0.25 | 0.52 | 0.26 | 0.46 | 0.31 | 0.46 | 0.29 | 0.46 |
| | | **Occ** | 0.40 | 0.56 | 0.43 | 0.54 | 0.44 | 0.53 | 0.45 | 0.51 | 0.49 | 0.48 | 0.50 | 0.47 |
| | Watershed volume | **CaOC** | 0.24 | 0.51 | 0.25 | 0.52 | 0.24 | 0.52 | 0.26 | 0.46 | 0.30 | 0.46 | 0.27 | 0.46 |
| | | **Occ** | 0.42 | 0.54 | 0.38 | 0.59 | 0.36 | 0.61 | 0.46 | 0.50 | 0.45 | 0.52 | 0.42 | 0.55 |
| **500** | BPT | **CaOC** | 0.21 | 0.38 | 0.09 | 0.00 | 0.08 | 0.22 | 0.22 | 0.35 | 0.16 | 0.01 | 0.07 | 0.18 |
| | | **Occ** | 0.52 | 0.35 | 0.00 | 0.01 | 0.53 | 0.16 | 0.55 | 0.33 | 0.00 | 0.01 | 0.53 | 0.12 |
| | Watershed area | **CaOC** | 0.23 | 0.51 | 0.26 | 0.52 | 0.25 | 0.52 | 0.26 | 0.46 | 0.30 | 0.46 | 0.29 | 0.46 |
| | | **Occ** | 0.40 | 0.56 | 0.43 | 0.53 | 0.44 | 0.53 | 0.44 | 0.51 | 0.48 | 0.48 | 0.50 | 0.47 |
| | Watershed volume | **CaOC** | 0.23 | 0.51 | 0.25 | 0.52 | 0.24 | 0.53 | 0.26 | 0.46 | 0.29 | 0.46 | 0.27 | 0.47 |
| | | **Occ** | 0.41 | 0.54 | 0.37 | 0.59 | 0.36 | 0.60 | 0.46 | 0.50 | 0.44 | 0.52 | 0.42 | 0.55 |

Table 9: Pixel Impact Rate (PIR) of selected explanation regions for Cat vs. Dog dataset. Highlighted in blue are the configurations presented in the main paper. The metric is the rate of the impact under occlusion (difference between the original class output and the output under occlusion) by the number of pixels of the occlusion mask. We tested hierarchies constructed by filtering out smaller regions than 200, 300, 400 and 500 pixels, segmentation based on **Edges**, Integrated Gradients (**IG**), Guided-Backpropagation (**BP**), Input X Gradients (**I X G**) and **Saliency**. We tested three different strategies to for the first hierarchical segmentation: BPT, watershed with area attribute, and watershed with volume attribute. We expect higher values, on average (**avg**), for PIR, meaning each occluded pixel has a high impact.

**Cat vs. Dog**

| | | | VGG | | | | | | | | | | ResNet | | | | | | | | | |
| | | | Edges | | IG | | BP | | I X G | | Saliency | | Edges | | IG | | BP | | I X G | | Saliency | |
| PIR | | | avg | std | avg | std | avg | std | avg | std | avg | std | avg | std | avg | std | avg | std | avg | std | avg | std |
|---|---|---|---|---|---|---|---|---|---|---|---|---|---|---|---|---|---|---|---|---|---|---|
| 200 | BPT | CaOC | 2.16e-04 | 2.91e-04 | 1.73e-02 | 7.79e-02 | 5.23e-03 | 3.59e-02 | 1.75e-02 | 8.72e-02 | 1.62e-04 | 2.15e-04 | 1.26e-04 | 2.26e-04 | 3.60e-03 | 2.25e-02 | 2.58e-03 | 1.81e-02 | 3.47e-03 | 2.17e-02 | 1.30e-04 | 2.63e-04 |
| | | Occ | 2.26e-04 | 3.32e-04 | 2.93e-03 | 3.24e-02 | 8.64e-04 | 1.60e-02 | 3.71e-04 | 4.11e-02 | 1.65e-04 | 2.25e-04 | 1.03e-04 | 1.81e-04 | 1.49e-03 | 1.39e-02 | 1.18e-03 | 8.90e-03 | 1.79e-03 | 1.79e-02 | 1.04e-04 | 2.38e-04 |
| | Watershed area | CaOC | 6.46e-04 | 1.05e-03 | 5.43e-04 | 7.83e-04 | 6.15e-04 | 9.65e-04 | 5.48e-04 | 7.56e-04 | 5.69e-04 | 8.33e-04 | 3.02e-04 | 5.17e-04 | 3.34e-04 | 5.25e-04 | 3.62e-04 | 6.36e-04 | 3.57e-04 | 5.56e-04 | 3.13e-04 | 4.91e-04 |
| | | Occ | 6.33e-04 | 1.11e-03 | 4.16e-04 | 5.01e-04 | 4.69e-04 | 7.79e-04 | 4.34e-04 | 5.49e-04 | 4.62e-04 | 6.98e-04 | 2.63e-04 | 4.58e-04 | 3.35e-04 | 5.32e-04 | 2.90e-04 | 4.50e-04 | 3.30e-04 | 4.80e-04 | 2.96e-04 | 4.62e-04 |
| | Watershed volume | CaOC | 6.17e-04 | 1.09e-03 | 5.33e-04 | 8.06e-04 | 5.41e-04 | 9.04e-04 | 5.55e-04 | 8.54e-04 | 4.17e-04 | 5.97e-04 | 2.94e-04 | 5.11e-04 | 3.36e-04 | 5.69e-04 | 3.61e-04 | 6.90e-04 | 3.82e-04 | 5.79e-04 | 2.83e-04 | 5.08e-04 |
| | | Occ | 5.31e-04 | 1.02e-03 | 4.02e-04 | 4.88e-04 | 4.66e-04 | 6.96e-04 | 4.35e-04 | 7.50e-04 | 4.06e-04 | 6.97e-04 | 2.34e-04 | 3.96e-04 | 3.30e-04 | 5.75e-04 | 2.84e-04 | 5.67e-04 | 3.28e-04 | 5.03e-04 | 2.39e-04 | |
| 300 | BPT | CaOC | 2.01e-04 | 2.62e-04 | 1.43e-02 | 6.83e-02 | 1.93e-03 | 1.73e-02 | 1.49e-02 | 7.99e-02 | 1.62e-04 | 2.17e-04 | 1.07e-04 | 1.67e-04 | 2.08e-03 | 1.73e-02 | 1.38e-03 | 1.24e-02 | 3.27e-03 | 2.23e-02 | 9.35e-05 | 1.35e-04 |
| | | Occ | 2.22e-04 | 3.22e-04 | 2.82e-03 | 3.27e-02 | 3.42e-04 | 3.15e-03 | 2.07e-03 | 2.73e-02 | 1.75e-04 | 2.40e-04 | 9.79e-05 | 1.63e-04 | 3.04e-04 | 3.29e-03 | 1.23e-03 | 7.41e-03 | 1.72e-03 | 1.78e-02 | 9.74e-05 | 2.07e-04 |
| | Watershed area | CaOC | 4.60e-04 | 6.14e-04 | 4.08e-04 | 5.48e-04 | 4.57e-04 | 6.28e-04 | 4.60e-04 | 5.98e-04 | 4.43e-04 | 5.85e-04 | 2.72e-04 | 4.83e-04 | 2.89e-04 | 4.54e-04 | 2.87e-04 | 4.23e-04 | 2.75e-04 | 4.00e-04 | 2.57e-04 | 4.06e-04 |
| | | Occ | 4.67e-04 | 7.36e-04 | 3.62e-04 | 4.39e-04 | 4.11e-04 | 5.70e-04 | 4.00e-04 | 4.89e-04 | 4.08e-04 | 6.17e-04 | 2.05e-04 | 3.44e-04 | 2.61e-04 | 4.20e-04 | 2.33e-04 | 3.31e-04 | 2.58e-04 | 3.59e-04 | 2.19e-04 | 3.33e-04 |
| | Watershed volume | CaOC | 4.30e-04 | 6.09e-04 | 4.05e-04 | 5.25e-04 | 4.48e-04 | 7.33e-04 | 4.40e-04 | 6.06e-04 | 3.83e-04 | 5.51e-04 | 2.55e-04 | 4.51e-04 | 2.85e-04 | 4.80e-04 | 2.77e-04 | 5.02e-04 | 2.90e-04 | 3.86e-04 | 2.14e-04 | 3.11e-04 |
| | | Occ | 4.34e-04 | 7.04e-04 | 3.58e-04 | 4.39e-04 | 4.16e-04 | 6.69e-04 | 3.86e-04 | 6.18e-04 | 3.20e-04 | 4.23e-04 | 2.08e-04 | 3.71e-04 | 2.41e-04 | 4.30e-04 | 2.21e-04 | 4.21e-04 | 2.52e-04 | 3.91e-04 | 1.95e-04 | 3.24e-04 |
| 400 | BPT | CaOC | 1.90e-04 | 2.72e-04 | 1.07e-02 | 6.12e-02 | 1.92e-03 | 1.35e-02 | 1.32e-02 | 7.72e-02 | 1.60e-04 | 2.06e-04 | 9.56e-05 | 1.26e-04 | 1.46e-03 | 1.12e-02 | 1.58e-03 | 1.31e-02 | 2.47e-03 | 1.99e-02 | 9.27e-05 | 1.26e-04 |
| | | Occ | 2.23e-04 | 3.41e-04 | 2.63e-03 | 3.24e-02 | 3.78e-04 | 3.03e-03 | 1.22e-03 | 1.76e-02 | 1.82e-04 | 2.43e-04 | 9.51e-05 | 1.35e-04 | 4.84e-04 | 3.89e-03 | 1.39e-03 | 8.44e-03 | 1.43e-03 | 1.73e-02 | 9.03e-05 | 1.87e-04 |
| | Watershed area | CaOC | 4.07e-04 | 5.23e-04 | 3.59e-04 | 4.40e-04 | 3.87e-04 | 4.51e-04 | 3.87e-04 | 4.85e-04 | 3.74e-04 | 4.38e-04 | 2.24e-04 | 3.32e-04 | 2.58e-04 | 4.20e-04 | 2.55e-04 | 3.92e-04 | 2.57e-04 | 3.37e-04 | 2.40e-04 | 3.48e-04 |
| | | Occ | 4.06e-04 | 6.05e-04 | 3.44e-04 | 4.25e-04 | 3.70e-04 | 4.66e-04 | 3.56e-04 | 4.29e-04 | 3.34e-04 | 4.23e-04 | 1.87e-04 | 3.01e-04 | 2.33e-04 | 3.98e-04 | 2.10e-04 | 3.22e-04 | 2.27e-04 | 3.35e-04 | 2.02e-04 | 3.27e-04 |
| | Watershed volume | CaOC | 3.81e-04 | 5.50e-04 | 3.43e-04 | 4.40e-04 | 3.66e-04 | 4.59e-04 | 3.80e-04 | 4.43e-04 | 3.09e-04 | 3.67e-04 | 2.21e-04 | 3.30e-04 | 2.57e-04 | 4.36e-04 | 2.68e-04 | 4.70e-04 | 2.67e-04 | 3.62e-04 | 1.88e-04 | 2.58e-04 |
| | | Occ | 3.59e-04 | 4.74e-04 | 3.33e-04 | 4.00e-04 | 3.51e-04 | 4.57e-04 | 3.47e-04 | 5.85e-04 | 2.88e-04 | 3.36e-04 | 1.75e-04 | 2.76e-04 | 2.25e-04 | 4.40e-04 | 1.91e-04 | 3.58e-04 | 2.13e-04 | 3.38e-04 | 1.76e-04 | 3.08e-04 |
| 500 | BPT | CaOC | 1.86e-04 | 2.71e-04 | 9.72e-03 | 5.64e-02 | 1.30e-03 | 9.98e-03 | 1.16e-02 | 7.14e-02 | 1.45e-04 | 1.30e-04 | 9.20e-05 | 1.16e-04 | 1.30e-03 | 1.07e-02 | 1.47e-03 | 1.25e-02 | 1.91e-03 | 1.73e-02 | 8.59e-05 | 1.11e-04 |
| | | Occ | 2.12e-04 | 3.13e-04 | 2.64e-03 | 3.24e-02 | 3.78e-04 | 3.03e-03 | 2.29e-03 | 3.22e-02 | 1.65e-04 | 1.76e-04 | 8.66e-05 | 1.07e-04 | 2.64e-04 | 2.32e-03 | 8.62e-04 | 5.83e-03 | 9.64e-04 | 1.44e-02 | 7.79e-05 | 8.43e-05 |
| | Watershed area | CaOC | 3.61e-04 | 4.70e-04 | 3.04e-04 | 3.48e-04 | 3.67e-04 | 4.44e-04 | 3.30e-04 | 3.86e-04 | 3.34e-04 | 3.94e-04 | 1.92e-04 | 2.86e-04 | 2.26e-04 | 3.09e-04 | 2.11e-04 | 2.99e-04 | 2.35e-04 | 3.13e-04 | 2.03e-04 | 2.70e-04 |
| | | Occ | 3.66e-04 | 5.30e-04 | 3.10e-04 | 3.61e-04 | 3.45e-04 | 4.40e-04 | 3.29e-04 | 3.71e-04 | 3.18e-04 | 3.28e-04 | 1.69e-04 | 2.52e-04 | 2.11e-04 | 3.05e-04 | 1.80e-04 | 2.46e-04 | 2.13e-04 | 3.10e-04 | 1.68e-04 | 2.56e-04 |
| | Watershed volume | CaOC | 3.50e-04 | 5.09e-04 | 2.92e-04 | 3.11e-04 | 3.26e-04 | 4.08e-04 | 3.43e-04 | 3.94e-04 | 2.74e-04 | 3.40e-04 | 1.97e-04 | 2.99e-04 | 2.27e-04 | 3.27e-04 | 2.20e-04 | 3.36e-04 | 2.43e-04 | 3.25e-04 | 1.84e-04 | 2.47e-04 |
| | | Occ | 3.21e-04 | 4.17e-04 | 3.14e-04 | 3.75e-04 | 3.22e-04 | 3.98e-04 | 3.21e-04 | 3.97e-04 | 2.89e-04 | 3.43e-04 | 1.62e-04 | 2.60e-04 | 1.93e-04 | 3.08e-04 | 1.61e-04 | 2.53e-04 | 2.02e-04 | 3.17e-04 | 1.57e-04 | 2.71e-04 |

Table 10: Pixel Impact Rate (PIR) of selected explanation regions for CIFAR10 dataset. Highlighted in blue are the configurations presented in the main paper. The metric is the rate of the impact under occlusion (difference between the original class output and the output under occlusion) by the number of pixels of the occlusion mask. We tested hierarchies constructed by filtering out smaller regions than 200, 300, 400 and 500 pixels, segmentation based on **Edges**, Integrated Gradients (**IG**), Guided-Backpropagation (**BP**), Input X Gradients (**I X G**) and **Saliency**. We tested three different strategies to for the first hierarchical segmentation: BPT, watershed with area attribute, and watershed with volume attribute. We expect higher values, on average (**avg**), for PIR, meaning each occluded pixel has a high impact.

**CIFAR10**

| PIR | | | VGG Edges avg | Edges std | IG avg | IG std | BP avg | BP std | I X G avg | I X G std | Saliency avg | Saliency std | ResNet Edges avg | Edges std | IG avg | IG std | BP avg | BP std | I X G avg | I X G std | Saliency avg | Saliency std |
|---|---|---|---|---|---|---|---|---|---|---|---|---|---|---|---|---|---|---|---|---|---|---|
| 4 | BPT | CaOC | 8.92e-03 | 1.90e-02 | 2.04e-01 | 3.87e-01 | 1.94e-01 | 3.87e-01 | 1.92e-01 | 3.56e-01 | 3.06e-02 | 6.51e-02 | 1.12e-02 | 2.09e-02 | 2.38e-01 | 3.78e-01 | 1.94e-01 | 3.52e-01 | 2.26e-01 | 3.68e-01 | 3.11e-02 | 5.15e-02 |
| | | Occ | 1.14e-02 | 2.06e-02 | 1.33e-01 | 4.93e-01 | 1.10e-01 | 4.34e-01 | 1.35e-01 | 5.10e-01 | 2.01e-02 | 4.91e-02 | 1.15e-02 | 2.00e-02 | 1.64e-01 | 4.96e-01 | 1.18e-01 | 4.14e-01 | 1.62e-01 | 4.99e-01 | 2.03e-02 | 3.54e-02 |
| | Watershed area | CaOC | 1.64e-02 | 3.61e-02 | 7.94e-02 | 1.36e-01 | 8.56e-02 | 1.45e-01 | 7.86e-02 | 1.34e-01 | 7.42e-02 | 1.37e-01 | 1.94e-02 | 3.41e-02 | 6.37e-02 | 1.03e-01 | 6.37e-02 | 1.03e-01 | 6.33e-02 | 1.03e-01 | 6.01e-02 | 9.98e-02 |
| | | Occ | 1.85e-02 | 3.77e-02 | 9.70e-02 | 1.66e-01 | 9.65e-02 | 1.71e-01 | 9.22e-02 | 1.60e-01 | 8.97e-02 | 1.54e-01 | 2.00e-02 | 3.46e-02 | 6.47e-02 | 1.08e-01 | 6.29e-02 | 1.08e-01 | 6.47e-02 | 1.09e-01 | 6.37e-02 | 1.05e-01 |
| | Watershed volume | CaOC | 1.67e-02 | 3.69e-02 | 7.08e-02 | 1.20e-01 | 7.63e-02 | 1.28e-01 | 7.17e-02 | 1.23e-01 | 6.49e-02 | 1.20e-01 | 1.91e-02 | 3.39e-02 | 5.84e-02 | 9.73e-02 | 5.79e-02 | 9.37e-02 | 6.07e-02 | 9.97e-02 | 5.47e-02 | 9.33e-02 |
| | | Occ | 1.79e-02 | 3.72e-02 | 9.41e-02 | 1.67e-01 | 9.39e-02 | 1.72e-01 | 9.08e-02 | 1.58e-01 | 8.21e-02 | 1.46e-01 | 1.94e-02 | 3.45e-02 | 6.21e-02 | 1.03e-01 | 6.25e-02 | 1.05e-01 | 6.40e-02 | 1.06e-01 | 5.74e-02 | 9.44e-02 |
| 16 | BPT | CaOC | 2.74e-03 | 7.32e-03 | 1.06e-01 | 3.00e-01 | 1.01e-01 | 3.00e-01 | 1.07e-01 | 3.06e-01 | 9.35e-03 | 1.43e-02 | 3.49e-03 | 8.13e-03 | 1.36e-01 | 3.29e-01 | 1.02e-01 | 2.71e-01 | 1.26e-01 | 3.21e-01 | 1.14e-02 | 1.56e-02 |
| | | Occ | 6.67e-03 | 9.48e-03 | 9.94e-02 | 4.34e-01 | 8.11e-02 | 3.77e-01 | 9.90e-02 | 4.38e-01 | 1.51e-02 | 1.80e-02 | 6.62e-03 | 1.00e-02 | 9.70e-02 | 3.91e-01 | 6.95e-02 | 3.09e-01 | 1.02e-01 | 4.06e-01 | 1.55e-02 | 1.75e-02 |
| | Watershed area | CaOC | 1.44e-02 | 2.80e-02 | 2.92e-02 | 4.60e-02 | 3.05e-02 | 4.81e-02 | 2.97e-02 | 4.70e-02 | 2.92e-02 | 4.73e-02 | 1.74e-02 | 2.85e-02 | 3.05e-02 | 3.91e-02 | 2.97e-02 | 3.95e-02 | 3.06e-02 | 3.95e-02 | 3.03e-02 | 3.99e-02 |
| | | Occ | 1.73e-02 | 2.99e-02 | 4.02e-02 | 5.16e-02 | 3.86e-02 | 5.13e-02 | 4.01e-02 | 5.06e-02 | 4.14e-02 | 5.31e-02 | 1.88e-02 | 3.06e-02 | 3.66e-02 | 4.37e-02 | 3.49e-02 | 4.26e-02 | 3.68e-02 | 4.29e-02 | 3.73e-02 | 4.29e-02 |
| | Watershed volume | CaOC | 1.44e-02 | 2.78e-02 | 2.85e-02 | 4.62e-02 | 2.76e-02 | 4.47e-02 | 2.94e-02 | 4.58e-02 | 2.65e-02 | 4.38e-02 | 1.70e-02 | 2.84e-02 | 2.91e-02 | 3.90e-02 | 2.86e-02 | 3.84e-02 | 3.02e-02 | 3.92e-02 | 2.83e-02 | 3.75e-02 |
| | | Occ | 1.67e-02 | 2.94e-02 | 3.81e-02 | 5.04e-02 | 3.67e-02 | 5.03e-02 | 3.93e-02 | 5.05e-02 | 3.72e-02 | 4.97e-02 | 1.81e-02 | 2.94e-02 | 3.55e-02 | 4.24e-02 | 3.48e-02 | 4.32e-02 | 3.64e-02 | 4.23e-02 | 3.47e-02 | 4.05e-02 |
| 32 | BPT | CaOC | 9.45e-04 | 3.75e-03 | 6.20e-02 | 2.33e-01 | 5.91e-02 | 2.35e-01 | 6.72e-02 | 2.48e-01 | 4.69e-03 | 8.17e-03 | 1.34e-03 | 4.39e-03 | 8.33e-02 | 2.68e-01 | 6.43e-02 | 2.28e-01 | 8.73e-02 | 2.84e-01 | 5.45e-03 | 8.49e-03 |
| | | Occ | 4.00e-03 | 6.38e-03 | 6.25e-02 | 3.46e-01 | 5.66e-02 | 3.13e-01 | 6.23e-02 | 3.48e-01 | 1.11e-02 | 1.08e-02 | 4.06e-03 | 6.66e-03 | 5.69e-02 | 3.00e-01 | 4.76e-02 | 2.57e-01 | 6.58e-02 | 3.31e-01 | 1.09e-02 | 1.02e-02 |
| | Watershed area | CaOC | 1.02e-02 | 1.88e-02 | 1.66e-02 | 2.44e-02 | 1.62e-02 | 2.41e-02 | 1.64e-02 | 2.32e-02 | 1.65e-02 | 2.37e-02 | 1.23e-02 | 1.98e-02 | 1.85e-02 | 2.24e-02 | 1.75e-02 | 2.17e-02 | 1.88e-02 | 2.28e-02 | 1.89e-02 | 2.31e-02 |
| | | Occ | 1.41e-02 | 2.12e-02 | 2.57e-02 | 2.78e-02 | 2.47e-02 | 2.75e-02 | 2.55e-02 | 2.66e-02 | 2.71e-02 | 2.90e-02 | 1.53e-02 | 2.20e-02 | 2.50e-02 | 2.54e-02 | 2.33e-02 | 2.48e-02 | 2.54e-02 | 2.58e-02 | 2.63e-02 | 2.63e-02 |
| | Watershed volume | CaOC | 1.01e-02 | 1.87e-02 | 1.60e-02 | 2.36e-02 | 1.55e-02 | 2.36e-02 | 1.65e-02 | 2.37e-02 | 1.55e-02 | 2.32e-02 | 1.22e-02 | 1.98e-02 | 1.81e-02 | 2.25e-02 | 1.73e-02 | 2.20e-02 | 1.88e-02 | 2.31e-02 | 1.78e-02 | 2.24e-02 |
| | | Occ | 1.36e-02 | 2.03e-02 | 2.50e-02 | 2.76e-02 | 2.39e-02 | 2.75e-02 | 2.55e-02 | 2.68e-02 | 2.48e-02 | 2.75e-02 | 1.49e-02 | 2.15e-02 | 2.48e-02 | 2.57e-02 | 2.36e-02 | 2.54e-02 | 2.58e-02 | 2.62e-02 | 2.45e-02 | 2.50e-02 |
| 64 | BPT | CaOC | 2.74e-04 | 1.57e-03 | 3.06e-02 | 1.59e-01 | 2.57e-02 | 1.54e-01 | 3.14e-02 | 1.59e-01 | 1.66e-03 | 4.27e-03 | 5.26e-04 | 2.09e-03 | 4.34e-02 | 1.96e-01 | 3.30e-02 | 1.74e-01 | 4.71e-02 | 2.09e-01 | 2.01e-03 | 4.47e-03 |
| | | Occ | 2.33e-03 | 4.46e-03 | 2.80e-02 | 2.31e-01 | 2.93e-02 | 2.18e-01 | 2.81e-02 | 2.36e-01 | 6.56e-03 | 7.59e-03 | 2.45e-03 | 4.60e-03 | 2.88e-02 | 2.08e-01 | 2.66e-02 | 1.91e-01 | 3.51e-02 | 2.38e-01 | 6.63e-03 | 7.37e-03 |
| | Watershed area | CaOC | 4.73e-03 | 1.02e-02 | 9.20e-03 | 1.32e-02 | 8.43e-03 | 1.27e-02 | 9.47e-03 | 1.30e-02 | 1.02e-02 | 1.41e-02 | 5.56e-03 | 1.05e-02 | 9.88e-03 | 1.21e-02 | 9.23e-03 | 1.18e-02 | 1.03e-02 | 1.26e-02 | 1.06e-02 | 1.32e-02 |
| | | Occ | 9.55e-03 | 1.30e-02 | 1.69e-02 | 1.56e-02 | 1.62e-02 | 1.50e-02 | 1.75e-02 | 1.56e-02 | 1.82e-02 | 1.66e-02 | 9.51e-03 | 1.27e-02 | 1.63e-02 | 1.40e-02 | 1.53e-02 | 1.39e-02 | 1.65e-02 | 1.42e-02 | 1.72e-02 | 1.53e-02 |
| | Watershed volume | CaOC | 4.65e-03 | 1.01e-02 | 9.23e-03 | 1.30e-02 | 8.38e-03 | 1.23e-02 | 9.64e-03 | 1.33e-02 | 9.16e-03 | 1.31e-02 | 5.46e-03 | 1.04e-02 | 1.02e-02 | 1.24e-02 | 9.66e-03 | 1.22e-02 | 1.05e-02 | 1.28e-02 | 9.71e-03 | 1.25e-02 |
| | | Occ | 9.32e-03 | 1.26e-02 | 1.69e-02 | 1.56e-02 | 1.59e-02 | 1.54e-02 | 1.76e-02 | 1.58e-02 | 1.67e-02 | 1.57e-02 | 9.33e-03 | 1.25e-02 | 1.63e-02 | 1.42e-02 | 1.55e-02 | 1.43e-02 | 1.68e-02 | 1.44e-02 | 1.62e-02 | 1.47e-02 |

Table 11: Pixel Impact Rate (PIR) of selected explanation regions for ImageNet dataset. Highlighted in blue are the configurations presented in the main paper. The metric is the rate of the impact under occlusion (difference between the original class output and the output under occlusion) by the number of pixels of the occlusion mask. We tested hierarchies constructed by filtering out smaller regions than 200, 300, 400 and 500 pixels, segmentation based on **Edges**, Integrated Gradients (**IG**), and Guided-Backpropagation (**BP**). We tested three different strategies to for the first hierarchical segmentation: BPT, watershed with area attribute, and watershed with volume attribute. We expect higher values, on average (**avg**), for PIR, meaning each occluded pixel has a high impact.

| PIR | | | Imagenet | | | | | | | | | | | |
| | | | VGG | | | | | | ResNet | | | | | |
| | | | Edges | | IG | | BP | | Edges | | IG | | BP | |
| | | | avg | std | avg | std | avg | std | avg | std | avg | std | avg | std |
| 200 | BPT | CaOC | 7.20e-04 | 1.08e-03 | 1.12e-02 | 7.68e-02 | 1.43e-02 | 7.37e-02 | 6.76e-04 | 9.63e-04 | 9.66e-03 | 5.60e-02 | 1.19e-02 | 5.15e-02 |
| | | Occ | 5.83e-04 | 8.19e-04 | 1.85e-03 | 2.15e-02 | 4.51e-03 | 3.43e-02 | 5.13e-04 | 7.27e-04 | 1.39e-03 | 1.35e-02 | 3.25e-03 | 2.35e-02 |
| | Watershed area | CaOC | 2.15e-03 | 3.63e-03 | 3.07e-03 | 4.29e-03 | 2.76e-03 | 4.07e-03 | 1.78e-03 | 2.77e-03 | 2.48e-03 | 3.12e-03 | 2.11e-03 | 2.80e-03 |
| | | Occ | 1.97e-03 | 3.36e-03 | 3.04e-03 | 4.00e-03 | 2.67e-03 | 3.74e-03 | 1.67e-03 | 2.58e-03 | 2.46e-03 | 2.88e-03 | 2.04e-03 | 2.55e-03 |
| | Watershed volume | CaOC | 1.97e-03 | 3.40e-03 | 2.85e-03 | 4.16e-03 | 2.58e-03 | 3.92e-03 | 1.67e-03 | 2.65e-03 | 2.33e-03 | 3.06e-03 | 1.95e-03 | 2.74e-03 |
| | | Occ | 1.74e-03 | 3.09e-03 | 2.96e-03 | 3.90e-03 | 2.75e-03 | 3.69e-03 | 1.49e-03 | 2.40e-03 | 2.39e-03 | 2.79e-03 | 2.05e-03 | 2.52e-03 |
| 300 | BPT | CaOC | 5.63e-04 | 8.29e-04 | 8.18e-03 | 6.40e-02 | 1.11e-02 | 6.01e-02 | 5.26e-04 | 7.45e-04 | 7.31e-03 | 4.88e-02 | 9.30e-03 | 4.51e-02 |
| | | Occ | 5.16e-04 | 6.70e-04 | 9.02e-04 | 1.48e-02 | 4.34e-03 | 3.24e-02 | 4.54e-04 | 5.93e-04 | 7.61e-04 | 9.12e-03 | 3.09e-03 | 2.17e-02 |
| | Watershed area | CaOC | 1.65e-03 | 2.54e-03 | 2.27e-03 | 2.97e-03 | 2.07e-03 | 2.84e-03 | 1.47e-03 | 2.11e-03 | 2.01e-03 | 2.43e-03 | 1.70e-03 | 2.15e-03 |
| | | Occ | 1.52e-03 | 2.35e-03 | 2.22e-03 | 2.75e-03 | 1.98e-03 | 2.62e-03 | 1.39e-03 | 1.99e-03 | 1.99e-03 | 2.24e-03 | 1.64e-03 | 2.01e-03 |
| | Watershed volume | CaOC | 1.52e-03 | 2.38e-03 | 2.10e-03 | 2.90e-03 | 1.92e-03 | 2.71e-03 | 1.38e-03 | 2.02e-03 | 1.90e-03 | 2.37e-03 | 1.58e-03 | 2.12e-03 |
| | | Occ | 1.35e-03 | 2.16e-03 | 2.14e-03 | 2.68e-03 | 2.02e-03 | 2.58e-03 | 1.25e-03 | 1.85e-03 | 1.92e-03 | 2.17e-03 | 1.65e-03 | 1.95e-03 |
| 400 | BPT | CaOC | 4.57e-04 | 6.86e-04 | 6.42e-03 | 5.48e-02 | 9.36e-03 | 5.30e-02 | 4.21e-04 | 6.06e-04 | 5.98e-03 | 4.32e-02 | 7.55e-03 | 4.08e-02 |
| | | Occ | 4.67e-04 | 5.82e-04 | 5.00e-04 | 1.03e-02 | 4.14e-03 | 3.23e-02 | 4.08e-04 | 5.05e-04 | 4.55e-04 | 6.85e-03 | 2.75e-03 | 2.03e-02 |
| | Watershed area | CaOC | 1.35e-03 | 1.96e-03 | 1.81e-03 | 2.25e-03 | 1.68e-03 | 2.18e-03 | 1.26e-03 | 1.74e-03 | 1.70e-03 | 1.99e-03 | 1.45e-03 | 1.78e-03 |
| | | Occ | 1.27e-03 | 1.84e-03 | 1.78e-03 | 2.12e-03 | 1.61e-03 | 2.02e-03 | 1.20e-03 | 1.62e-03 | 1.69e-03 | 1.88e-03 | 1.39e-03 | 1.64e-03 |
| | Watershed volume | CaOC | 1.25e-03 | 1.85e-03 | 1.69e-03 | 2.19e-03 | 1.56e-03 | 2.12e-03 | 1.18e-03 | 1.66e-03 | 1.62e-03 | 1.97e-03 | 1.35e-03 | 1.76e-03 |
| | | Occ | 1.14e-03 | 1.69e-03 | 1.71e-03 | 2.04e-03 | 1.61e-03 | 1.98e-03 | 1.08e-03 | 1.52e-03 | 1.63e-03 | 1.81e-03 | 1.39e-03 | 1.61e-03 |
| 500 | BPT | CaOC | 3.75e-04 | 5.76e-04 | 3.13e-04 | 8.14e-03 | 7.97e-03 | 4.78e-02 | 3.44e-04 | 5.07e-04 | 3.07e-04 | 6.29e-03 | 3.67e-03 | 3.14e-02 |
| | | Occ | 4.25e-04 | 5.20e-04 | 5.34e-03 | 4.88e-02 | 7.91e-03 | 4.75e-02 | 3.69e-04 | 4.42e-04 | 5.12e-03 | 3.96e-02 | 6.23e-03 | 3.67e-02 |
| | Watershed area | CaOC | 1.16e-03 | 1.60e-03 | 1.54e-03 | 1.83e-03 | 1.41e-03 | 1.77e-03 | 1.10e-03 | 1.47e-03 | 1.46e-03 | 1.66e-03 | 1.25e-03 | 1.51e-03 |
| | | Occ | 1.10e-03 | 1.50e-03 | 1.52e-03 | 1.72e-03 | 1.37e-03 | 1.65e-03 | 1.05e-03 | 1.39e-03 | 1.46e-03 | 1.58e-03 | 1.22e-03 | 1.38e-03 |
| | Watershed volume | CaOC | 1.07e-03 | 1.51e-03 | 1.44e-03 | 1.78e-03 | 1.31e-03 | 1.72e-03 | 1.02e-03 | 1.39e-03 | 1.40e-03 | 1.64e-03 | 1.17e-03 | 1.15e-03 |
| | | Occ | 9.84e-04 | 1.38e-03 | 1.46e-03 | 1.66e-03 | 1.34e-03 | 1.58e-03 | 9.48e-04 | 1.30e-03 | 1.42e-03 | 1.54e-03 | 1.20e-03 | 1.36e-03 |

Table 12: Percentage of images with the original class changed after the **inclusion** (exclusively) of selected explanation regions for Cat vs. Dog dataset. Highlighted in blue are the configurations presented in the main paper. We tested hierarchies constructed by filtering out smaller regions than 200, 300, 400 and 500 pixels, segmentation based on **Edges**, Integrated Gradients (**IG**), Guided-Backpropagation (**BP**), Input X Gradients (**I X G**) and **Saliency**. We tested three different strategies to for the first hierarchical segmentation: BPT, watershed with area attribute, and watershed with volume attribute. We expect smaller rate values of class change.

| % of images | | | Cat vs. Dog | | | | | | | | | |
| --- | --- | --- | --- | --- | --- | --- | --- | --- | --- | --- | --- | --- |
| | | | VGG | | | | | ResNet | | | | |
| | | | Edges | IG | BP | I X G | Saliency | Edges | IG | BP | I X G | Saliency |
| 200 | BPT | CaOC | 0.16 | 0.49 | 0.38 | 0.49 | 0.37 | 0.22 | 0.45 | 0.28 | 0.47 | 0.35 |
| | | Occ | 0.11 | 0.18 | 0.04 | 0.24 | 0.11 | 0.19 | 0.40 | 0.18 | 0.43 | 0.31 |
| | Watershed area | CaOC | 0.30 | 0.45 | 0.39 | 0.46 | 0.42 | 0.34 | 0.47 | 0.43 | 0.45 | 0.49 |
| | | Occ | 0.22 | 0.22 | 0.26 | 0.24 | 0.31 | 0.30 | 0.49 | 0.41 | 0.49 | 0.52 |
| | Watershed volume | CaOC | 0.27 | 0.44 | 0.43 | 0.50 | 0.45 | 0.34 | 0.47 | 0.49 | 0.46 | 0.50 |
| | | Occ | 0.24 | 0.30 | 0.36 | 0.28 | 0.29 | 0.30 | 0.50 | 0.46 | 0.47 | 0.50 |
| 300 | BPT | CaOC | 0.19 | 0.49 | 0.37 | 0.49 | 0.36 | 0.22 | 0.49 | 0.28 | 0.49 | 0.30 |
| | | Occ | 0.10 | 0.27 | 0.05 | 0.35 | 0.10 | 0.17 | 0.46 | 0.22 | 0.48 | 0.25 |
| | Watershed area | CaOC | 0.30 | 0.43 | 0.39 | 0.44 | 0.43 | 0.33 | 0.46 | 0.40 | 0.44 | 0.47 |
| | | Occ | 0.18 | 0.20 | 0.25 | 0.24 | 0.29 | 0.24 | 0.46 | 0.40 | 0.43 | 0.47 |
| | Watershed volume | CaOC | 0.28 | 0.45 | 0.43 | 0.45 | 0.45 | 0.31 | 0.46 | 0.46 | 0.45 | 0.46 |
| | | Occ | 0.21 | 0.29 | 0.34 | 0.27 | 0.29 | 0.25 | 0.49 | 0.45 | 0.45 | 0.45 |
| 400 | BPT | CaOC | 0.21 | 0.49 | 0.39 | 0.49 | 0.36 | 0.22 | 0.50 | 0.29 | 0.50 | 0.29 |
| | | Occ | 0.10 | 0.34 | 0.07 | 0.40 | 0.11 | 0.18 | 0.48 | 0.25 | 0.49 | 0.27 |
| | Watershed area | CaOC | 0.26 | 0.41 | 0.38 | 0.44 | 0.42 | 0.31 | 0.47 | 0.38 | 0.44 | 0.46 |
| | | Occ | 0.17 | 0.19 | 0.21 | 0.22 | 0.29 | 0.24 | 0.43 | 0.36 | 0.40 | 0.45 |
| | Watershed volume | CaOC | 0.27 | 0.44 | 0.43 | 0.46 | 0.45 | 0.33 | 0.47 | 0.43 | 0.47 | 0.46 |
| | | Occ | 0.19 | 0.26 | 0.30 | 0.28 | 0.28 | 0.22 | 0.47 | 0.41 | 0.44 | 0.43 |
| 500 | BPT | CaOC | 0.20 | 0.49 | 0.41 | 0.49 | 0.35 | 0.22 | 0.50 | 0.31 | 0.50 | 0.29 |
| | | Occ | 0.08 | 0.38 | 0.11 | 0.43 | 0.12 | 0.16 | 0.49 | 0.29 | 0.50 | 0.26 |
| | Watershed area | CaOC | 0.26 | 0.41 | 0.37 | 0.44 | 0.41 | 0.32 | 0.43 | 0.35 | 0.41 | 0.46 |
| | | Occ | 0.17 | 0.19 | 0.22 | 0.22 | 0.28 | 0.23 | 0.42 | 0.32 | 0.40 | 0.42 |
| | Watershed volume | CaOC | 0.25 | 0.45 | 0.42 | 0.44 | 0.42 | 0.32 | 0.44 | 0.42 | 0.45 | 0.45 |
| | | Occ | 0.19 | 0.28 | 0.31 | 0.26 | 0.29 | 0.22 | 0.45 | 0.37 | 0.42 | 0.41 |

Table 13: Percentage of images with the original class changed after the **inclusion** (exclusively) of selected explanation regions for CIFAR10 dataset. Highlighted in blue are the configurations presented in the main paper. We tested hierarchies constructed by filtering out smaller regions than 4, 16, 32 and 64 pixels, segmentation based on **Edges**, Integrated Gradients (**IG**), Guided-Backpropagation (**BP**), Input X Gradients (**I X G**) and **Saliency**. We tested three different strategies to for the first hierarchical segmentation: BPT, watershed with area attribute, and watershed with volume attribute. We expect smaller rate values of class change.

| % of images | | | CIFAR10 | | | | | | | | | |
|---|---|---|---|---|---|---|---|---|---|---|---|---|
| | | | VGG | | | | | ResNet | | | | |
| | | | Edges | IG | BP | I X G | Saliency | Edges | IG | BP | I X G | Saliency |
| **4** | BPT | CaOC | 0.79 | 0.86 | 0.87 | 0.87 | 0.86 | 0.81 | 0.86 | 0.86 | 0.87 | 0.86 |
| | | Occ | 0.44 | 0.38 | 0.45 | 0.30 | 0.66 | 0.51 | 0.46 | 0.53 | 0.41 | 0.70 |
| | Watershed area | CaOC | 0.80 | 0.89 | 0.88 | 0.89 | 0.88 | 0.83 | 0.88 | 0.87 | 0.88 | 0.88 |
| | | Occ | 0.55 | 0.81 | 0.81 | 0.81 | 0.81 | 0.59 | 0.81 | 0.79 | 0.80 | 0.80 |
| | Watershed volume | CaOC | 0.80 | 0.89 | 0.88 | 0.89 | 0.88 | 0.83 | 0.88 | 0.88 | 0.88 | 0.88 |
| | | Occ | 0.53 | 0.82 | 0.82 | 0.81 | 0.80 | 0.58 | 0.81 | 0.80 | 0.81 | 0.79 |
| **16** | BPT | CaOC | 0.83 | 0.85 | 0.86 | 0.87 | 0.83 | 0.86 | 0.86 | 0.85 | 0.87 | 0.83 |
| | | Occ | 0.54 | 0.41 | 0.43 | 0.44 | 0.60 | 0.61 | 0.51 | 0.54 | 0.54 | 0.63 |
| | Watershed area | CaOC | 0.80 | 0.87 | 0.87 | 0.87 | 0.87 | 0.83 | 0.87 | 0.86 | 0.87 | 0.87 |
| | | Occ | 0.54 | 0.77 | 0.76 | 0.77 | 0.77 | 0.59 | 0.78 | 0.76 | 0.77 | 0.77 |
| | Watershed volume | CaOC | 0.80 | 0.87 | 0.87 | 0.87 | 0.86 | 0.83 | 0.87 | 0.87 | 0.87 | 0.86 |
| | | Occ | 0.53 | 0.78 | 0.78 | 0.77 | 0.76 | 0.58 | 0.78 | 0.77 | 0.77 | 0.76 |
| **32** | BPT | CaOC | 0.87 | 0.87 | 0.87 | 0.88 | 0.83 | 0.89 | 0.88 | 0.85 | 0.89 | 0.81 |
| | | Occ | 0.62 | 0.59 | 0.52 | 0.62 | 0.61 | 0.68 | 0.64 | 0.60 | 0.68 | 0.62 |
| | Watershed area | CaOC | 0.80 | 0.85 | 0.85 | 0.86 | 0.85 | 0.83 | 0.85 | 0.85 | 0.86 | 0.86 |
| | | Occ | 0.53 | 0.73 | 0.73 | 0.73 | 0.74 | 0.59 | 0.74 | 0.72 | 0.74 | 0.74 |
| | Watershed volume | CaOC | 0.79 | 0.86 | 0.85 | 0.86 | 0.85 | 0.83 | 0.85 | 0.85 | 0.86 | 0.85 |
| | | Occ | 0.52 | 0.75 | 0.75 | 0.74 | 0.72 | 0.58 | 0.75 | 0.75 | 0.75 | 0.73 |
| **64** | BPT | CaOC | 0.89 | 0.89 | 0.88 | 0.89 | 0.86 | 0.90 | 0.89 | 0.87 | 0.90 | 0.86 |
| | | Occ | 0.71 | 0.74 | 0.66 | 0.77 | 0.67 | 0.75 | 0.77 | 0.72 | 0.79 | 0.69 |
| | Watershed area | CaOC | 0.80 | 0.84 | 0.83 | 0.85 | 0.83 | 0.85 | 0.83 | 0.82 | 0.84 | 0.83 |
| | | Occ | 0.54 | 0.68 | 0.67 | 0.68 | 0.71 | 0.61 | 0.69 | 0.67 | 0.68 | 0.71 |
| | Watershed volume | CaOC | 0.80 | 0.84 | 0.84 | 0.85 | 0.83 | 0.84 | 0.83 | 0.83 | 0.83 | 0.83 |
| | | Occ | 0.54 | 0.71 | 0.73 | 0.70 | 0.69 | 0.61 | 0.71 | 0.72 | 0.70 | 0.69 |

Table 14: Percentage of images with the original class changed after the **inclusion** (exclusively) of selected explanation regions for Imagenet dataset. Highlighted in blue are the configurations presented in the main paper. We tested hierarchies constructed by filtering out smaller regions than 200, 300, 400 and 500 pixels, segmentation based on **Edges**, Integrated Gradients (**IG**), and Guided-Backpropagation (**BP**). We tested three different strategies to for the first hierarchical segmentation: BPT, watershed with area attribute, and watershed with volume attribute. We expect smaller rate values of class change.

| % of images | | | Imagenet | | | | | |
|---|---|---|---|---|---|---|---|---|
| | | | VGG | | | ResNet | | |
| | | | Edges | IG | BP | Edges | IG | BP |
| **200** | **BPT** | **CaOC** | 0.95 | 1.00 | 0.97 | 0.96 | 1.00 | 0.98 |
| | | **Occ** | 0.72 | 0.69 | 0.51 | 0.74 | 0.57 | 0.50 |
| | **Watershed area** | **CaOC** | 0.98 | 0.99 | 0.98 | 0.98 | 0.99 | 0.99 |
| | | **Occ** | 0.92 | 0.93 | 0.91 | 0.93 | 0.94 | 0.90 |
| | **Watershed volume** | **CaOC** | 0.98 | 0.99 | 0.99 | 0.98 | 0.99 | 0.99 |
| | | **Occ** | 0.91 | 0.97 | 0.97 | 0.92 | 0.96 | 0.96 |
| **300** | **BPT** | **CaOC** | 0.94 | 1.00 | 0.97 | 0.95 | 1.00 | 0.98 |
| | | **Occ** | 0.72 | 0.82 | 0.55 | 0.74 | 0.72 | 0.55 |
| | **Watershed area** | **CaOC** | 0.97 | 0.98 | 0.98 | 0.98 | 0.99 | 0.98 |
| | | **Occ** | 0.92 | 0.93 | 0.90 | 0.92 | 0.94 | 0.90 |
| | **Watershed volume** | **CaOC** | 0.97 | 0.99 | 0.99 | 0.97 | 0.99 | 0.99 |
| | | **Occ** | 0.91 | 0.96 | 0.96 | 0.91 | 0.96 | 0.96 |
| **400** | **BPT** | **CaOC** | 0.94 | 1.00 | 0.97 | 0.95 | 1.00 | 0.98 |
| | | **Occ** | 0.73 | 0.88 | 0.59 | 0.75 | 0.81 | 0.60 |
| | **Watershed area** | **CaOC** | 0.97 | 0.98 | 0.97 | 0.97 | 0.99 | 0.98 |
| | | **Occ** | 0.91 | 0.93 | 0.90 | 0.92 | 0.93 | 0.89 |
| | **Watershed volume** | **CaOC** | 0.97 | 0.99 | 0.98 | 0.97 | 0.99 | 0.99 |
| | | **Occ** | 0.90 | 0.96 | 0.96 | 0.91 | 0.96 | 0.96 |
| **500** | **BPT** | **CaOC** | 0.94 | 0.92 | 0.98 | 0.94 | 0.86 | 0.98 |
| | | **Occ** | 0.74 | 1.00 | 0.63 | 0.75 | 1.00 | 0.63 |
| | **Watershed area** | **CaOC** | 0.96 | 0.98 | 0.97 | 0.97 | 0.99 | 0.98 |
| | | **Occ** | 0.90 | 0.92 | 0.89 | 0.91 | 0.93 | 0.89 |
| | **Watershed volume** | **CaOC** | 0.96 | 0.98 | 0.98 | 0.97 | 0.99 | 0.98 |
| | | **Occ** | 0.89 | 0.96 | 0.96 | 0.90 | 0.96 | 0.96 |

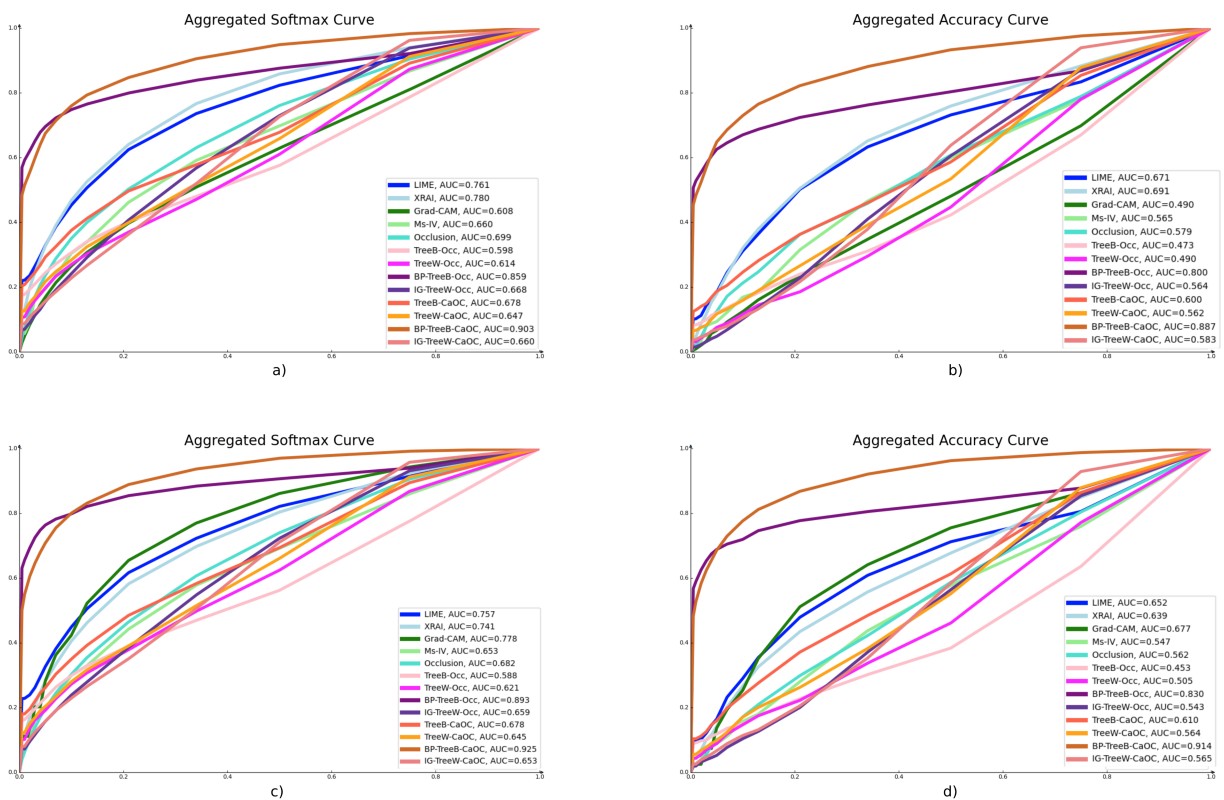

Figure 10: Softmax (a,c) and Accuracy (b,d) when including regions filtered by different percentage thresholds of most important scores, for VGG16 (a,b) and ResNet-18 (c,d) models. We evaluate each threshold as a hierarchy level in eight configurations of *xAiTrees* (**C1** and **C2**), in a bottom-up approach (from smaller highly important regions to the bigger structures). We compare these configurations to the baselines: LIME, XRAI, Grad-CAM, Ms-IV, and Occlusion, by filtering the maps using the same threshold. The curves are an aggregation by the average of 1,000 randomly selected images from Imagenet dataset. AUC values are included in the graphs. BP-TreeB-Occ and BP-TreeB-CaOC considerably surpassed the other curves. However, we notice a good early behavior of the methods except for Grad-CAM, Ms-IV and Occlusion.

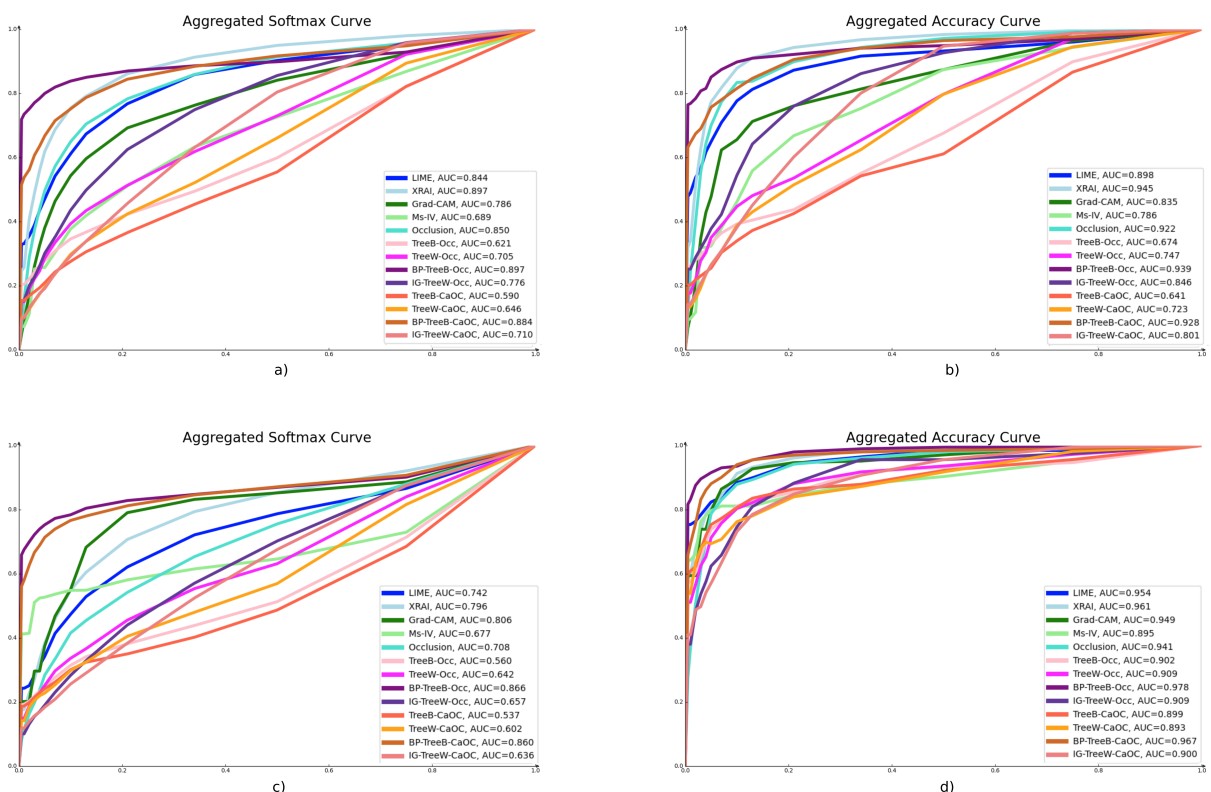

Figure 11: Softmax (a,c) and Accuracy (b,d) when including regions filtered by different percentage thresholds of most important scores, for VGG16 (a,b) and ResNet-18 (c,d) models. We evaluate each threshold as a hierarchy level in eight configurations of *xAiTrees* (**C1** and **C2**), in a bottom-up approach (from smaller highly important regions to the bigger structures). We compare these configurations to the baselines: LIME, XRAI, Grad-CAM, Ms-IV, and Occlusion, by filtering the maps using the same threshold. The curves are an aggregation by the average of 512 images from Cat vs. Dog dataset. AUC values are included in the graphs.

**Time analysis:** Table 15 shows the execution times to explain an image (averaged from 100 randomly selected cat vs. dog images). Tree-Occ is a much faster option for providing explanations than SOTA methods such as XRAI. However, the execution time of the xAiTrees framework is related to the choice of scoring method used to assign importance to regions (CaOC, Occ, or LIME), therefore methods such as LIME that take more time to generate explanations will increase the time of xAiTrees, as shown in the Tree-LIME example.

Table 15: Execution times to explain an image (averaged from 100 randomly selected cat vs. dog images).

| Mean time 100 images | | LIME | XRAI | Grad-CAM | Occ. | Ms-IV | TreeB Occ | BP-TreeB Occ | TreeW Occ | IG-TreeW Occ |
|---|---|---|---|---|---|---|---|---|---|---|
| VGG | Mean | 7.87 | 11.97 | 0.009 | 6.65 | 0.27 | 0.60 | 0.65 | 0.76 | 1.31 |
| | Avg. | 0.82 | 1.25 | 0.0002 | 2.47 | 0.03 | 0.08 | 0.09 | 0.10 | 0.08 |
| ResNet | Mean | 5.64 | 10.19 | 0.007 | 3.41 | 0.17 | 0.50 | 0.57 | 0.58 | 0.84 |
| | Avg. | 0.40 | 1.36 | 0.0002 | 0.03 | 0.07 | 0.06 | 0.05 | 0.06 | 0.07 |
| | | TreeB CaOC | BP-TreeB CaOC | TreeW CaOC | IG-TreeW CaOC | TreeB LIME | BP-TreeB LIME | TreeW LIME | IG-TreeW LIME | |
| VGG | Mean | 0.71 | 0.81 | 0.87 | 1.53 | 31.91 | 41.07 | 49.96 | 54.98 | - |
| | Avg. | 0.11 | 0.11 | 0.13 | 0.10 | 7.56 | 8.24 | 11.11 | 3.25 | - |
| ResNet | Mean | 0.60 | 0.68 | 0.73 | 1.04 | 19.44 | 27.01 | 31.58 | 44.46 | - |
| | Avg. | 0.08 | 0.08 | 0.10 | 0.07 | 4.75 | 3.99 | 6.99 | 2.90 | - |

### A.6 Qualitative analysis

**Models' description:** Table 16 shows the number of images in the train set for three Cat vs. Dog ResNet18 biased models. For **Bias 1** the biased class is composed of only cats on top of cushions. For **Bias 2** the biased class is composed of only dogs next to grades. For **Bias 3** the biased class is composed of only cats with humans. We also include the accuracy percentage per class when predicting a non-biased validation set composed by 5,109 images.

The biased models were trained with initial weights from Imagenet, learning rate $5e-7$, cross-entropy loss, the Adam optimizer, and early stop in 20 epochs of non-improving validation loss.

Table 16: Number of images (for a normal and an induced biased class) for training three biased ResNet18 models. We also present the accuracy of the models when predicting each class image from a non-biased validation dataset (5,109 images).

|        | Normal class | Acc. orig. val normal (%) | Bias class | Acc. orig. val bias (%) |
|--------|--------------|---------------------------|------------|-------------------------|
| Bias 1 | 138          | 86.91                     | 69         | 84.82                   |
| Bias 2 | 85           | 97.97                     | 56         | 37.81                   |
| Bias 3 | 161          | 86.28                     | 46         | 80.96                   |

**Comparison of misclassified images:** Considering the hierarchical characteristic of our methodology, we can perform a deeper analysis of the explanations by selecting regions by the percentage of importance to be visualized, as shown in Figure 12. In the last level of the dishwasher example, the model seems to focus on the cat's dish after having focused on the sink (in the previous level).

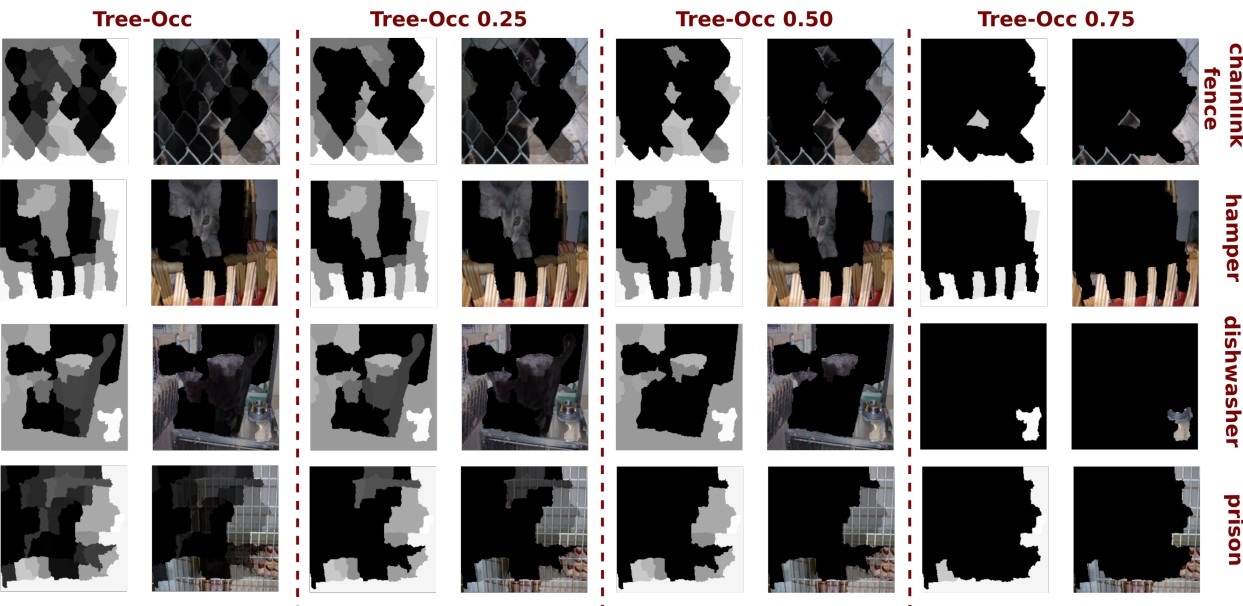

Figure 12: Different visualization levels on the explanation hierarchy. We illustrate a deeper analysis of the explanations of four images from Figure 2 using Tree-Occ (minimal region size of 500 pixels). We can note the evolution of the importance in the images' shapes, for examples: in the hamper image, although the hamper is the most important, the cat has also an important that disappears at the more selective level (Tree-Occ 0.75); in the dishwasher the initial explanations show the sink as important but at the most selective level, the cat's dish is the only one remaining. This analysis can be helpful to understand the reasoning behind predictions.

**Human evaluation in bias analysis:**  As mentioned on the paper, we used the configuration **C1** for human-interpretation evaluation compared to baseline techniques: IG, Grad-CAM, OCC, LIME, Ms-IV, ACE. We presented the same five image visualizations (from corrected classified images by the biased class) for the baseline methods and the methods from **C1**.

The idea was to analyze the impact of the visualizations on people from different backgrounds. We limited ourselves to people over the age of 18 and recorded their self-identification as expert, non-expert, field of expertise, and country. We show some statistics of each group of 41 people that participated in our evaluation.

Expertise areas:

- 48.8% of people from computer science;

- 17.1% of people from human sciences;

- 19.5% of people from life sciences;

- 9.8% of people from exact sciences (not in Computer Science);

- 4.9% of people from none of the above.

Graduate degree:

- 20.0% of PhDs;

- 32.5% of Masters;

- 30.0% of Bachelor's degree;

- 17.5% of High school diploma.

AI expertise:

- 22.0% None;

- 36.6% Basic;

- 9.8% Intermediate;

- 7.3% Advanced;

- 24.4% Expert (working with AI).

We intended to verify if: (i) humans can detect the wrong focus given based on a class prediction (**Detection**); and (ii) humans can recognize which was the cause of the bias (**Identification**).

To test (i) and (ii), for each **Bias** type we produce for each of the xAI methods an explanation image. By presenting five image explanations (the same images) for each of the xAI methodologies, we asked volunteers, based on the explanations provided, what did they think the highlighted regions referred to. The five image explanations are presented in Figure 13 for each **Bias** type (1-(a), 2-(b), and 3-(c)).

Here, we display the text provided to the volunteers for this experiment:

---

**[FORM] Part I - Determining the focus of the images:** For each question, we provide two rows of images:

- The first row displays the original images, each representing a specific class.

- The second row showcases an image for each image from the first row, highlighting the important parts for the class.

[IMPORTANT] What is a class?

A class refers to a category or type of object, animal, or characteristic depicted in the images. For instance, a class of cat images would include images featuring cats, while a class of dog images would comprise images featuring dogs. Similarly, a class of cartoon images would include images characterized by cartoon-like features. In essence, a class represents a distinct category used to classify and organize images based on their content or characteristics.

Throughout the questions, our objective is to identify the common important parts present in the images of the first row, as indicated by the corresponding images in the second row.

If no common important parts are identified for most of the images, the answer should be Not identified.

---

And for each method visualization:

---

For the following three questions, the second row of images displays significant image components to the class of the animal.

What are the significant components of the images highlighted, as depicted in the second row of images?

---

To test ACE similarly as we did with the other methods, we highlight the top five concepts found (described as sufficient in the original ACE paper (Ghorbani et al., 2019)) in the same five selected images. However, we also show the visualizations of the ten most activated images for the top five found concepts in Figure 14.

In our final qualitative experiment, using the same methods as the previous human evaluation, we presented four image explanation visualizations for non-biased models to determine xAI model preferences. The images are presented in Figure 15. We presented the following explanation and question:

---

**[FORM] Part II: Choosing the best representation:**

For the next questions, you will be asked to answer which image number do you prefer to describe the class we indicate.

You should choose the image that seems to highlight class features in an easier way to understand.

Which image do you think better shows representative parts of the animal?

---

For the two first images (Figures 15 (a) and (b)), over 70% preferred Tree-Occ and Tree-CaOC over others. For the third image (Figure 15 (c)), IG was preferred by 26.5%, followed by Tree-OCC and Occlusion with 20.6%. Grad-CAM was preferred in the fourth image (Figure 15 (d)), with 60.6%, followed by Tree-CaOC with 18.2%. The visualizations suggest a preference for explanations that highlight the complete concept (cat or dog) rather than focusing on specific small animals' regions.

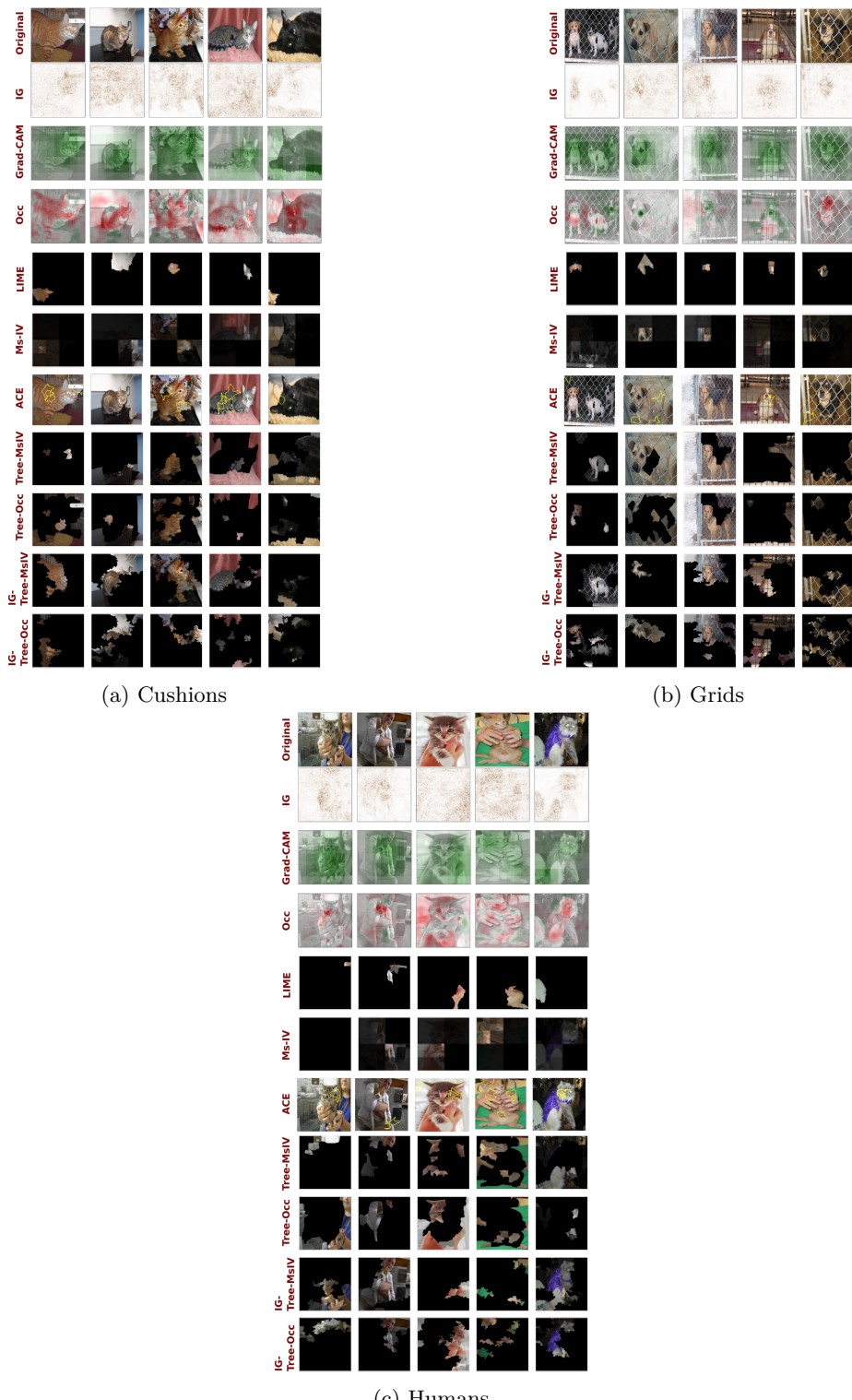

(a) Cushions

(b) Grids

(c) Humans

Figure 13: Explanations of visualizations used on our human-based evaluations for bias detection and identification, of all the ten compared methods: IG, Grad-CAM, OCC, LIME, Ms-IV, ACE, Tree-MsIV, Tree-Occ, IG-Tree-Msiv, and IG-Tree-Occ. We showed the same five image explanations for all the methods.

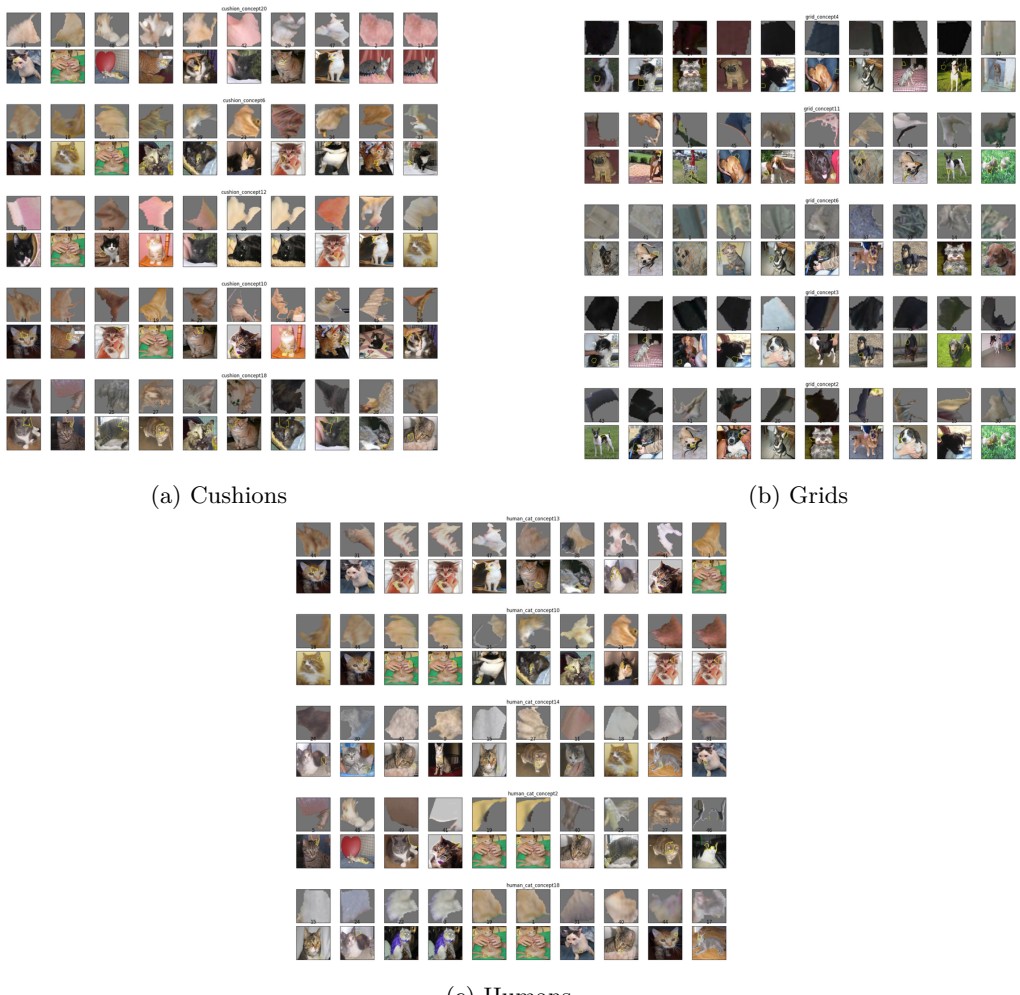

(a) Cushions

(b) Grids

(c) Humans

Figure 14: Original explanations of the top 5 concepts generated by ACE for the three biased models. Instead of showing the 10 most concepts' activated images we draw these five top concepts on the 5 selected images from Figure 13 to have a fairer comparison with the other methods.

## A.7   Possible adaptations of the framework

Regarding the adaptability of the framework for using other xAI methods to score regions, we present in Table 17 preliminary results of the exclusion of important regions experiment (Section 5 and A.5) by using LIME in the place of Occlusion and CaOC. The results demonstrate Tree-LIME improves the class change under important regions' occlusion for the Cat vs. Dog dataset (compared to Table 1). We also show two image explanations generated by Tree-LIME in Figure 16. We do not include this variation in all the experiments due to the time consumption (Table 15). However, these preliminary results demonstrate Tree-LIME can increase the class change (**Ch.**) reaching higher percentages than the best configuration presented in the main paper.

Regarding the adaptability of the framework to other tasks such as learning representations, one suggestion would be calculating the distance between the two learned representations (original and after occlusion) to attribute regions' scores instead of verifying the *logits* different as in classification tasks. The type of distance applied should be tested. A more sophisticated evaluation of impact (scoring the hierarchy regions) would be to include a network to quantify the quality of the representation for the task at hand.

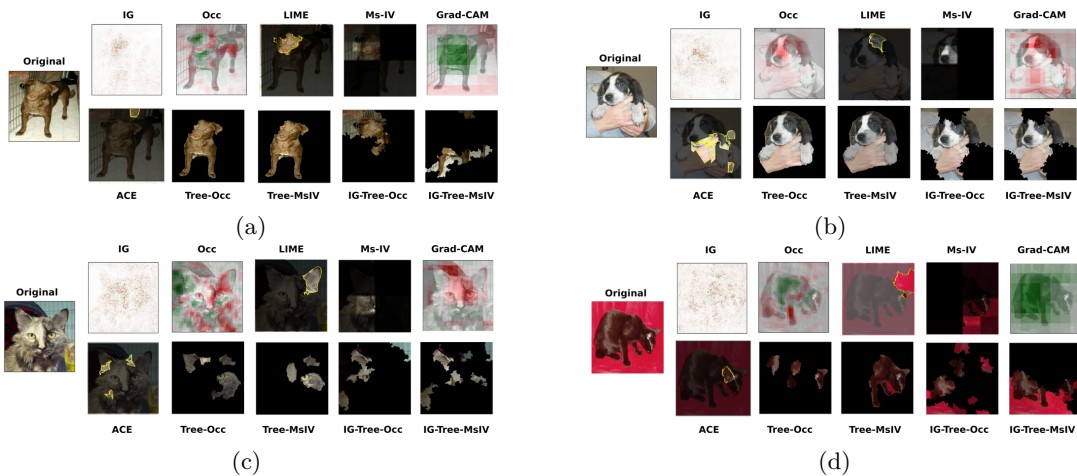

Figure 15: Explanations of visualizations used on our human-based evaluations for preference analysis, of all the ten compared methods: IG, Grad-CAM, OCC, LIME, Ms-IV, ACE, Tree-MsIV, Tree-Occ, IG-Tree-Msiv, and IG-Tree-Occ. We showed the same five image explanations for all the methods.

Table 17: Percentage of images with the original class changed after the **exclusion** of selected explanation regions. We tested TreeW, IG-TreeW, TreeB, and BP-TreeB combined to LIME (instead of using occlusion to score regions) in two architectures, VGG-16 and ResNet18, for the Cat vs. Dog dataset. We compare the results of the four configurations to the best configuration using Occlusion (BP-TreeB-Occ) showing the p-score in brackets (Mcnemar test). We expect a higher percentage of class change (**Ch.**) when the region is excluded. **Same** column shows images maintaining the original class when the output was reduced, and **Total** is the sum of class change (**Ch.**) and class reduction (**Same**).

| % of images | Cat vs. Dog | | | | | |
|---|---|---|---|---|---|---|
| | VGG | | | ResNet | | |
| | **Ch.** | **Same** | **Total** | **Ch.** | **Same** | **Total** |
| **TreeW-LIME** | **0.34 (0.0)** | 0.64 (0.0) | **0.98** | **0.43 (1.2-4)** | 0.56 (7.2-10) | **0.98** |
| **IG-TreeW-LIME** | **0.45 (2.2-9)** | 0.54 (2.7-10) | **0.99** | **0.43 (2.0-4)** | 0.55 (2.5-9) | **0.98** |
| **TreeB -LIME** | **0.57 (0.05)** | 0.42 (0.01) | **0.98** | **0.64 (2.3-3)** | 0.35 (0.34) | **0.99** |
| **BP-TreeB-LIME** | **0.70 (2.2-9)** | 0.29 (2.7-10) | **0.99** | **0.77 (2.0-4)** | 0.21 (2.5-9) | **0.98** |
| **BP-TreeB-Occ** | 0.63 | 0.35 | **0.98** | 0.55 | 0.37 | **0.92** |

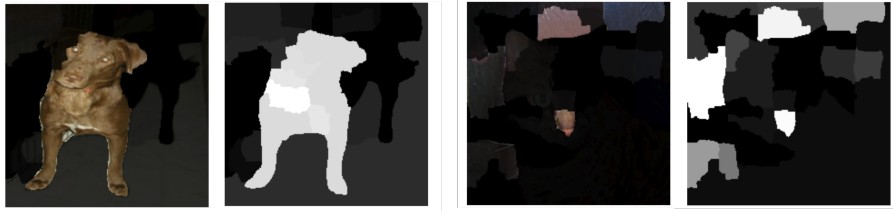

Figure 16: Examples of Tree-LIME demonstrating the adaptability of xAiTrees framework.

Concerning other modalities, such as text as input, we could consider a tokenization process as the segmentation. A first idea would be to define a tokenization algorithm that can learn merge rules and use them to construct the segmentation tree. Each node in this tree can be considered a segment. Another idea would be to use grammar rules to define parts of a sentence, such as nouns, verbs, and declinations. However, this approach would be focused on a specific language.

