# OpenReview forum: "Explaining with trees: interpreting CNNs using hierarchies"
_TMLR — Accepted by TMLR_

### Review · Reviewer_PDyw · 2025-04-29

**Summary Of Contributions:**

This paper introduces xAiTrees, a framework designed to generate more faithful and interpretable explanations for CNNs by utilizing hierarchical segmentation techniques. Addressing the trade-off between model fidelity and human understanding seen in existing XAI methods, xAiTrees constructs multiscale explanations by combining hierarchical structures (derived from either traditional edge detection or novel model-based attribution maps) with region importance scoring (like Occlusion). The authors claim this approach, particularly the model-based segmentation variant, significantly improves upon methods like LIME and Grad-CAM, offering clearer, more precise identification of impactful regions, better elucidation of misclassifications, and enhanced effectiveness in detecting model biases, as supported by quantitative evaluations and human studies.

**Audience:**

Yes

**Claims And Evidence:**

Yes

**Requested Changes:**

Please try to discuss more about the weaknesses.

**Strengths And Weaknesses:**

Strengths:
- Novelty in Model-Based Segmentation: The introduction of model-based hierarchical segmentation, where the segmentation hierarchy is derived from pixel-wise XAI attributions (like IG, BP), is a novel and interesting contribution. This aims to align the segmentation structure more closely with the model's internal reasoning compared to standard, human-centric segmentation.

- Comprehensive Framework (xAiTrees): The paper proposes a well-structured, four-step framework that integrates hierarchical segmentation, attribute computation (importance scoring), tree shaping (persistence analysis), and hierarchical visualization. This provides a systematic approach to generating multiscale explanations.

- Framework Flexibility: The paper demonstrates that xAiTrees can be adapted by plugging in different segmentation criteria, hierarchical segmentation algorithms, and region scoring metrics.

Weaknesses:
- Complexity of the Framework: The multi-step process involving choices of segmentation type, pixel-wise method (for model-based), occlusion metric, tree shaping, and aggregation heuristic adds complexity. Understanding the contribution of each component and tuning the framework might require significant expertise.
- Interpretation of Model-Based Segmentation: While novel, the claim that segmentation based on IG/BP truly reflects the model's "vision" warrants further discussion. These methods capture gradient information, but whether the resulting segments align with meaningful semantic concepts used by the model (or just group pixels with similar attribution values) is not fully explored. There's a potential risk of the explanation being an artifact of the specific pixel-wise method chosen.
- Dependence on Base Methods: As acknowledged, the quality of the final explanation heavily depends on the quality of the underlying segmentation algorithm and the chosen region importance scoring method (Occlusion, CaOC, etc.). xAiTrees inherits the limitations of these components.

---

> ### Author Response · Authors · 2025-05-06
>
> We would like to thank the reviewer for taking the time to read our work and for the helpful comments. We discuss the identified weaknesses in the following paragraphs.
>
> ----
>
> Complexity of the Framework:  The complexity of our methodology comes mainly from its flexibility. To modify the components of our framework, it will be necessary to understand the framework and the implications of changes. However, it was a project choice to make these modifications available to the users. We believe that, despite its complexity, this flexibility offers a significant advantage, allowing users to explore the model's knowledge from multiple perspectives.
>
> Knowing this, we also believe it is important to have standard configurations, for general use, that could reflect the two main aspects we want to emphasize in explainability: the model’s “vision” and the human “vision”. Based on our experiments, we propose two general configurations to the users: one that reflects the model’s perspective, IG-Tree-Occ, and another that incorporates human bias during segmentation, Tree-Occ. These configurations are available in our GitHub repository to facilitate their use.
>
> ----
> Interpretation of Model-Based Segmentation: When we refer to the model-based segmentation as the model’s "vision," we aim to distinguish it from the conventional segmentation approach, which reflects human visual biases. By using attribution methods to guide segmentation, we aim to group pixels that most strongly activate the model, an approach analogous to how edges in images attract human attention. However, both methods can be misleading due to the inherent limitations of attribution techniques and edge detectors, among other factors.
>
> That said, we agree with the reviewer’s concern regarding the potential for artifacts in the explanations. Since there is no ground truth for the concepts learned by the model, we evaluated the alignment of these explanations with the model’s internal knowledge through quantitative experiments, specifically by measuring performance when key regions were included or excluded.
>
> ----
>
> Dependence on Base Methods: We acknowledge the limitations of the method, which is why we invested considerable effort in testing across diverse datasets, models, and techniques to build our framework. Through these experiments, we found that some of our framework configurations, in particular IG-Tree-Occ, consistently outperform baseline methods in terms of fidelity to the model, based on both quantitative and qualitative metrics.
>
> However, this performance gain is not our primary focus. What we emphasize most is the advantage of using a hierarchical structure, which allows for a flexible and interpretable balance between fidelity and interpretability. This flexibility is a core of our approach: as individual components, such as attribution methods or segmentation techniques, are improved, they can be integrated into the framework, enabling continuous enhancement without redesigning the entire system.

---

### Review · Reviewer_3YHd · 2025-06-02

**Summary Of Contributions:**

First, let me apologize for the delay. I had done the review once but unfortunately lost my notes and could not recover them. I thus did this review a second time.
The authors developed a hierarchical segmentation based XAI framework to improve CNN’s model explanations, applied to OCC, LIME, … etc. They experimentally show that their methods identify important zones of the images for the decisions, and that it augments bias detection and identification scores.

Most crucial claims seem to be supported by the experimental evidence, but they are not clearly outlined.

With a greatly improved presentation, this paper should be of interest for PMLR's readers.

**Audience:**

Yes

**Broader Impact Concerns:**

Impact of better explanations methods, and bias identification could be discussed in a dedicated section.

**Claims And Evidence:**

Yes

**Requested Changes:**

**Improve clarity of your contributions in the introduction.**

Many sentences bring no new/relevant information, due to their imprecise aspect. E.g. “*This need arises from biases present at various stages of model development and deployment*.”

Time and space are precious values, use it to precisely explain the problem you are addressing, the method you developed, the evaluation you conducted.  \
“*To meet the demand for explanation,*” can be omitted.

Please verify that your claims are supported by published available work, or by your experimental evaluation. e.g.:

* “*This needs to be clarified: “using a segmentation framework introduces human bias, which [...] may reduce faithfulness to the model’s actual behavior.*” Why ?
* “*However, these methods are approximations of model behavior. Different techniques prioritize either faithfulness to the model’s behavior or human interpretability, posing a challenge in balancing the two.*”

If existing published work supports this, just point to it, otherwise, you need to develop your argumentation. Anyways, not sure if these help the reader understand which precise scientific problem you are trying to tackle.

**Use figures as the concise version of your paper.** Many readers, after reading the abstract, skim through the figures and tables. Thus, I advise incorporating these points within the figures/tables:



* What is the (unsolved) problem at hand that your method tackles or improves?
* How does your method work (nicely done in Figure 3)
* Different figures that each answer an evaluation criteria. Ideally, your experiment section is composed of precise scientific questions aimed to be answered. For example: \
5.1 Does xAITrees provide more accurate important regions ?   \
5.2 Does xAITrees help with bias detection and identification ?  \

Each of your figures then answer these questions. Please convert difficult to read tables into bar diagram, and place the table in appendix.

**Clarify your figures, and their captions**. E.g. Figure 1. This figure is supposed to highlight the main scientific problem addressed in this paper. Thus, the first sentence should describe it (in bold). E.g.:

>**Current explanation methods are difficult to interpret, our Tree-Occ method improves interpretations**, as shown by the fact that it highlights the fence…

(I am personally not convinced by the “better” explanation of Tree-Occ here). A user study might be useful to determine if the Tree-occ’s segmentation provides better interpretations.

On this figure, the y label is “Imagenet classes”, that should correspond to the label associated with the depicted images, within the Imagenet dataset. However, the caption claims that these are classes predicted by VGG and Resnet. I doubt that, seeing that some have one, and others many classes. Do VGG and Resnet agree on the classes to output for all of these classes?

It seems that for each explanation model, the first column is the explanation map, and the second is the aggregated mask and image. There is thus only one mask per image, does it come from VGG or ResNet ?

Finally, your example of the cat image is quite interesting. There are indeed several objects corresponding to classes present in imagenet (the cat and the dishwasher). You are writing: “* Tree-Occ avoids the mistake of highlighting the cat when the models predict classes such as dishwasher*”. Does the used model *really* predict “*dishwasher*” ? I would expect it to predict “*cat*”, particularly when looking at the other explanation methods. (this also applies to the other depicted images containing cats).

Figure 2 seems to be an experimental result, supporting a point, why is it placed so early in the paper? Again here, no clear message can be extracted. Figures should be standalone, and understandable even if the reader has not yet read the full text.

For Figure 4, you want to show the superiority of Tree-method over method, then use e.g. dotted line for the original method and full line for your adapted ones, keep the same color to compare them. Further, it seems that TreeB-Occ and TreeW-Occ really underperform there. While you can comment on this for precision and fairness of your method, you can drop them from this figure to augment the visual clarity (btw, you could use smaller linewidth, and a smaller alpha of e.g. 0.8).


**While the reported results of table 4 are very relevant, the metrics are really unintuitive.**

“Animal” and “Not identified” are imprecise labels that give the wrong intuition. Reading “Identification” and “Not identified”, I get the impression that the first one is 100 - the other. This table also lack second order measures (like std)

**No *Limitation* section**. The paper should include a limitation section. Is your method applicable to more modern architectures ? Only if these explanation methods are ? What is the time needed to generate an explanation compared to the baselines ?

**Strengths And Weaknesses:**

The method seems really interesting, even if the paper does not really convey why the hierarchical structure is needed, and the provided illustrations are not really convincing.

The main issue addressed in this paper is not precisely defined: Use a bold sentence to tell the reader what the main addressed problem is.
“We propose to explore the trade-off between explaining Convolutional Neural Networks (CNNs) with model faithfulness and human interpretability.” This is not a clear definition of the problem, thus a loss of space (and time). Then “we introduce a framework that combines hierarchical segmentation with region-based explanation methods.” Does that mean that no other framework already does this ? If so, then what difference does it have to yours? Why would PMLR readers use this framework rather than any other one (such as Multiscale Interpretable Visualization)?

Your contribution seems to be something like “we provide an explanation-model agnostic framework to transform pixel-wise attribution methods to region-based ones”. (correct me if I am wrong, I have not really been able to identify the exact problem+contribution.)
“We can transform any pixel-wise attribution into region-based explanations.”

Confusing vocabulary: “Reasoning” is usually used to refer to logic based (or at least symbolic) agents. CNNs do not reason per se, they infer.

Bold unsupported claims: see in the next section for examples.

Unclear figures: This is a crucial point for me. see next section for requested changes.

No limitation figure

---

> ### Author Response · Authors · 2025-06-20
>
> We thank you for the time you dedicated to reading our work and for providing valuable suggestions to improve it. We have done our best to address your concerns. Below, we detail the modifications made to the paper:
>
> **Improve clarity of your contributions in the introduction**
>
> We revised the introduction to enhance clarity and incorporated additional literature references to better support our claims.
>
> “Explainable Artificial Intelligence (xAI) provides methods that clarify models’ decision-making processes with different levels of interpretability, which can be described as the measure of how easy it is to understand an explanation (Gilpin et al., 2018). Providing interpretable explanations is especially important in the previously mentioned contexts (e.g., health), where humans need to understand models’ decisions.
>
> For this purpose, some xAI methods use object-structure-based visualizations to enhance human interpretation. They decompose images in ways that mimic human perception, grouping objects by attributes such as color, texture, and edges (Hubel & Wiesel, 1959). Techniques such as LIME (Ribeiro et al., 2016) and KernelSHAP (Lundberg & Lee, 2017) have used this approach effectively, segmenting images into meaningful parts to improve interpretability. However, the size of the segmented regions affects the information extracted: small regions can be difficult to interpret, while large regions may overlook fine details. Additionally, using a segmentation framework may introduce a form of human bias, encouraging the model to attribute importance to structures that a human might consider relevant. This can aid comprehension but may reduce fidelity
> to the model’s actual behavior (Miró-Nicolau et al., 2024), which does not necessarily align with human reasoning.
>
> On the other hand, methods like Deconvolution (Zeiler & Fergus, 2014), Integrated Gradients
> (IG)(Sundararajan et al., 2017), and LRP(Bach et al., 2015), which attribute importance to features (pixels), aid in understanding deep learning models across various applications (Borys et al., 2023; Dharshini et al., 2023; Chaddad et al., 2023), serving as better approximations of model behavior (Borys et al., 2023), and locally explaining decisions. However, they often lack interpretability due to their pixel-level explanations, which can be difficult for humans to understand (Kim et al., 2018). Different techniques tend to prioritize either faithfulness to the model’s behavior or human interpretability, making it challenging to balance the two.”
>
> We also highlighted the main propositions in bold, as suggested, to improve clarity.
>
> **Use figures as the concise version of your paper.**
>
> We agree with the reviewer on the importance of using figures to improve the paper’s clarity and facilitate comprehension. Accordingly, we have added three additional figures as suggested: to illustrate the motivation (Figure 1), the quantitative experiments (Figure 5), and the qualitative experiments (Figure 7).
>
> We also revised Figure 6 by reducing the line width and representing the baselines with dashed lines, as suggested, to improve readability.
>
> We considered converting the tables into bar plots. However, given the complexity of the data—with 14 methods, 2 models, and 3 datasets—such plots would become dense and difficult to interpret unless split into multiple graphs. To maintain readability and conserve space in the main paper, we chose to keep the tables, which present the data more compactly. We will also consider using a different color scheme to further improve clarity.
>
> **Clarify your figures, and their captions**
>
> We acknowledge the potential confusion regarding Figure 1 and the models used. In this example, we selected misclassifications from either ResNet or VGG—not both simultaneously. The images shown are not from the ImageNet dataset, although the models were trained on ImageNet. We defined misclassification based on our initial expectations, as the central object in each image was either a dog or a cat. We have now clarified in the figure which model was used to generate each explanation.
>
> We initially included the Tree-LIME figure in Figure 2a to illustrate the flexibility of the framework and to optimize page layout. However, we agree that the figure was misplaced there, so we have moved it to the supplementary material. Additionally, we updated Figure 2b to better illustrate the idea of using the hierarchy.

---

> ### Author Response · Authors · 2025-06-20
>
> **While the reported results of table 4 are very relevant, the metrics are really unintuitive**
>
> The total of all responses (for each bias and each method) must sum to 100%, as each explanation is assigned to a single category. To improve clarity, we changed the category "Animal" by "No Bias," which indicates that the evaluators believed there was no bias in the explanation. Additionally, the category "Not Identified" was renamed to "Unknown," meaning that the participants were unsure about where the model's focus was.
>
> We did not calculate the standard deviation from this user study  because the data consisted of categorical responses from 41 participants. Since standard deviation is a measure of dispersion for numerical data, we presented only the percentages of each category chosen.
>
> **No *Limitation* section**
>
> The segmentation section was originally placed in the supplementary materials, but we agree with the reviewer on its importance and have now included it in the main paper:
>
> “The computational time is a limitation when using time-expensive methods to attribute region scores, such as LIME. We show the time comparison including the baseline methods in A.5 Table 15. That is why we limited our analysis to Tree-Occ and Tree-CaOC.
>
> The method needs adaptations to be used in different tasks such as learning representations, or to be applied to other modalities such as texts. These adaptations are discussed in A.7.
>
> The proposed version of xAiTrees framework is dependent on the base methods used. Therefore, by using an edge-based segmentation method, we will not obtain a semantic-based explanation, i.e., the final technique will inherit the limitations of the base methods. Future works will be focused on semantic segmentation.”

---

### Review · Reviewer_9sNp · 2025-06-13

**Summary Of Contributions:**

The paper proposes a methodology for explaining images classified by neural networks followed by a number of experimental evaluations.

**Audience:**

Yes

**Claims And Evidence:**

No

**Requested Changes:**

see above

**Strengths And Weaknesses:**

Although explainability (xAI) is a very relevant and timely area of research, there are a number of problems with the paper. The main problem has to do with the limited novelty: the proposed methodology puts together a number of existing methods and approaches with some justification for the choices made, but the paper does not introduce a new xAI approach. The main motivation is rather flawed: reasoning! Despite appearing in the title of the paper, there is very little about reasoning that is discussed or formally defined in the paper. As a result, there is no relationship to be identified between the proposed methodology and reasoning as understood in the context of knowledge representation in AI. Another main problem is the limited related work.

The paper focuses on local explainability of images and uses a well-known image xAI method (LIME) in its pipeline. It is well-known now that LIME can be very limited in what it explains (https://arxiv.org/abs/1908.03020) and how that explanation is measured. A reference is made in the paper to the "fidelity" of the explanation. This can be measure (see e.g. https://arxiv.org/abs/2106.14556), but the experimental evaluations are mostly concerned with specific cases and subjective evaluations. The limitations of occlusion based approaches are also well-studied. Although the tree structure is used in the proposed methodology to reduce the problem of explainability based on occlusion, the main evaluation is based on the direct comparison of occlusion inside and outside the hierarchy. It is difficult to see how such a measure can be reliable. This kind of evaluation seems to be highly context dependent. The rest of the evaluation is based on human inspection. Here, a lot more detail would need to be provided on the evaluation method, how many participants, questionnaires, etc. The pointed out limitations of Grad-CAM are also well known. In summary, with limited novelty, one would expect extensive and detailed critical comparative evaluations of results to have been carried out, but the empirical analysis is also limited, as already noted.

The metric "original class changing after the exclusion of selected explanations" seems to tell us very little. The process that generates the segmentation hierarchy is crucial and is not presented in any level of detail. What motivated the choice of the specific shaping approach?
Visualization by itself does not really provide explanations and I can't see at all how it might help with reasoning. The "occlusion scores" are not defined. The choice of certain approaches to be part of the proposed pipeline is said to be based on the fact that the methods are state of the art at pixel-wise importance attribution. This claim is not backed by references. State of the art wrt which metric?

The generation of the initial tree and the process to refine the tree are crucial but are discussed in little detail. I'd refer to this paper which extracts a tree from a CNN layer: https://ceur-ws.org/Vol-3432/paper3.pdf. It could offer a way of comparing results, even though the method used there isn't for local xAI.

How would the results be affected by the performance of other approaches such as Segment Anything? Various other approaches exist that seek to benefit from some concept hierarchy, typically in the learning of the part-of relation (see e.g. https://arxiv.org/abs/1705.08968). Some comparison or at least a discussion of pros and cons, and limitations of the proposed method, should have been discussed.

---

> ### Author Response · Authors · 2025-06-20
>
> **Limited novelty**
>
> Thank you for your feedback on our submission. We appreciate your comment regarding the perceived limited novelty of our work. While we understand the importance of novelty in research, we would like to highlight that, according to the TMLR guidelines, the journal "emphasizes technical correctness over subjective significance." Our paper focuses on providing a technically sound contribution, with rigorous methodology and validated results, which we believe aligns with TMLR’s stated priorities.
>
> While we acknowledge that our approach draws upon existing segmentation and explainability techniques, we would like to emphasize that our primary contribution lies in the **design and integration of a novel framework—xAiTrees—for hierarchical segmentation-based explanations.**
>
> Specifically, our work introduces three key innovations:
>
> 1. **xAiTrees**, a novel hierarchical explanation framework, explicitly incorporates **multiscale region importance** into model interpretation. While prior works provide flat or pixel-level attributions, xAiTrees enables a **structured decomposition** of the model explanation.
> 2. We introduce a **model-based segmentation strategy** within xAiTrees that allows flexible adaptation of **pixel-wise attribution methods into region-level explanations**, thereby bridging the gap between low-level saliency and high-level interpretation. This is a **technical integration** that improves both usability and interpretability.
> 3. Beyond proposing the framework, we provide a **quantitative and qualitative evaluation**, including **bias identification via human analysis**, which demonstrates practical utility and adds empirical value to the contribution.
>
> While it is true that xAiTrees builds on existing components (e.g., attribution and segmentation techniques), the **framework itself is novel**, and the way in which these components are **re-purposed and structured hierarchically for xAI** offers new explanatory capabilities.
>
> **Reasoning as main motivation**
>
> We agree that the term *reasoning* may not be the most accurate descriptor for our approach, as it could suggest that the method itself performs full model interpretation or autonomous inference. Our intention, however, is different. We use *reasoning* in the sense of **supporting human reasoning**—that is, enabling users to better interpret and navigate model explanations. The goal of our framework is to present explanations in a structured, hierarchical form that helps users reason more effectively about model behavior. In doing so, we aim to **facilitate the trade-off between fidelity and interpretability**, allowing explanations to remain faithful to the model while being more accessible and meaningful to human users.
>
> However, as we agree that this can generate confusion in the main motivation, we changed the title of the paper from **Reasoning with trees: interpreting CNNs using hierarchies** to **Explaining with trees: interpreting CNNs using hierarchies.**
>
> **Limited related work.**
>
> We increased our related work section by including a paragraph about neural-symbolic explanations:
>
> “Neural-symbolic explanations: Techniques such as DCR [1], X-NeSyL [2], and the approach proposed by [3] adopt neural-symbolic strategies to explain the decisions of neural networks within specific contexts. These methods enhance transparency by translating low-level neural activations into high-level concepts, logical rules, or hierarchical structures, making explanations more aligned with human reasoning. However, this often comes at the cost of increased complexity, as such approaches may require manually crafted symbolic rules or domain-specific annotations to generate meaningful explanations.”
>
> **The paper focuses on local explainability of images and uses a  well-known image xAI method (LIME) in its pipeline**
>
> We would like to clarify that LIME is **not used as a core component** of our pipeline. The example included in the paper is intended purely as an **illustrative case**, demonstrating that our framework is compatible with a wide range of region-based scoring methods. This flexibility is a key part of our contribution.
>
> We fully agree with the reviewer regarding the limitations of methods like LIME—particularly in terms of fidelity to the original model. However, these methods **remain widely used** in practice, which is why we aimed to design a framework that can **partially mitigate their shortcomings**. Our goal is to provide a **flexible, modular pipeline** that supports both established and emerging xAI techniques, while offering improved interpretability without sacrificing too much fidelity.

---

> ### Author Response · Authors · 2025-06-20
>
> **"Fidelity" of the explanation**
>
> By *fidelity*, we refer to how well an explanation reflects the true behavior of the underlying model. Common metrics for evaluating fidelity include occlusion and inclusion tests on important regions, as well as SIC and AIC curves [5]. For this reason, we included these experiments in the paper.
>
> Due to space constraints, we limited the number of comparisons in the main text. However, the full set of results—including comparisons with **five baseline methods and 120 configurations  of our proposed framework**—are provided in the supplementary materials. These evaluations were conducted across **two model architectures** and **three datasets** to ensure robustness.
>
> Finally, the subjective (human-centered) evaluations are intended to **complement the quantitative results** by assessing how interpretable the explanations are from a human perspective—highlighting not just fidelity to the model, but also usefulness and clarity for end users.
>
> **The main evaluation is based on the direct comparison of  occlusion inside and outside**
>
> Besides the comparison with occlusion inside the framework and outside, we also included comparisons with other types of baseline methods (5 different methods) that do not rely directly on simple occlusions such as XRAI and LIME . The selection of baselines was guided by a review of the most commonly used methods in the xAI literature.
>
> **The rest of the evaluation is based on human  inspection:** Here, a lot more detail would need to be provided on the  evaluation method, how many participants, questionnaires, etc.
>
> Due to space limitations, we included the full details of the user study in the supplementary material (Section A.6). This includes descriptions of the questions, the images used in the forms, and statistics of the 41 evaluators who participated in the study.
>
> **Metric "original class changing after the exclusion of selected  explanations"**
>
> The first idea for the metric was simply to calculate the impact on the logits after occlusion. However, any kind of perturbation can affect the logits and not necessarily the classification. In this particular case, since we are dealing with a classification problem, we consider the class change as the main evidence that an important image region has been occluded. Therefore, Ch is the % of class change, Same is the % of same class but with reduced logits, and Total is the sum of both, showing the number of images where the region occlusion had a negative impact on the current class. This evaluation is inspired by the region deletion from [4].
>
> In the Inclusion [4] metric, we evaluate the opposite when we include the important region, how many images are not able to keep the correct classification. In this metric, we expect smaller percentages for better methods.
>
> However, we understand that this way of using insertion and deletion metrics is not so conventional, therefore we present some additional results to evaluate the differences between Softmax and Accuracy by applying different threshold importance to the explanations. We used the methodology proposed by [5] by inserting important regions in a basic blurred image (obtained from the original). We used the thresholds of 0.005, 0.01, 0.02, 0.03, 0.04, 0.05, 0.07, 0.10, 0.13, 0.21, 0.34, 0.5, 0.75 percent of the most important region values.
>
> **Process that generates the segmentation hierarchy and motivation of shaping approach**
>
> The segmentation strategies, including the concepts of segmentation trees, Binary Partition Tree (BPT), and watershed methods used in this paper, are detailed in Appendix A.1 for further reference. We also included a new diagram in Appendix A. 1, exemplifying the segmentation tree construction, to enhance the explanations of the method.
>
> The tree shaping approach proposed in [6, 7] aims to modify the structure of the original image segmentation tree based on a new attribute—in our case, the evaluation of each region’s impact on the model’s decision. Initially, the hierarchical segmentation provides a set of candidate regions, and subsequently, the shaping process adjusts the hierarchy to reflect the importance of these regions in the model’s predictions. This refinement ensures that the final tree better represents the model-relevant structures.
>
> **The "occlusion scores"  and chosen  pixel-wise importance attribution**
>
> We described the metrics such as occlusion and CaOC on supplementary material A.2.
>
> We chose some of the pixel-wise importance approaches for the model-based segmentation part according to the findings of [8] about gradient-based methods to have higher fidelity.  We included the reference in the main text.

---

> ### Author Response · Authors · 2025-06-20
>
> **Generation of the initial tree**
>
> The generation of the original tree is described in A.1 and used a framework named Higra [9, 10]. To increase clarity, we included a diagram exemplifying a segmentation tree construction in A.1. It follows a classical hierarchical image segmentation approach. We found the work in [3], particularly interesting and have included it in the related work section. However, we chose not to use it as a baseline because it requires training on an additional dataset with annotated parts to enable symbolic rule extraction. This requirement does not meet our goal of providing **flexible, context-adaptable explanations** without the need for task-specific annotations.
>
> **How would the results be affected by the performance of other  approaches, such as Segment Anything and concept hierarchy?**
>
> Although Segment Anything is a powerful segmentation method, it does not explicitly model the segmentation as a hierarchy, it treats segments as mostly **flat**, not nested or hierarchical, which invibilizes the use in our framework.
>
> Our idea of hierarchy is image-based, which provides the flexibility to construct it through different approaches. Specifically, we explored both human-guided segmentation—using image edges to inform the hierarchy—and model-based segmentation—leveraging attribution maps. Employing a concept hierarchy instead would have prevented a direct comparison between these methods.
>
> We tested two classic hierarchical segmentation strategies: watershed and Binary Partition Trees (BPT). The complete results are provided in the supplementary materials. These methods were selected for their simplicity and ease of integration, as our goal was to demonstrate that the framework can effectively operate with straightforward, well-established segmentation techniques.
>
> ---
>
> [1] Pietro Barbiero, Gabriele Ciravegna, Francesco Giannini, Mateo Espinosa Zarlenga, Lucie Charlotte Magister, Alberto Tonda, Pietro Lió, Frederic Precioso, Mateja Jamnik, and Giuseppe Marra. Interpretable neural-symbolic concept reasoning. In Proceedings of the 40th International Conference on Machine Learning, ICML’23. [JMLR.org](http://jmlr.org/), 2023.
>
> [2] Natalia Díaz-Rodríguez, Alberto Lamas, Jules Sanchez, Gianni Franchi, Ivan Donadello, Siham Tabik, David Filliat, Policarpo Cruz, Rosana Montes, and Francisco Herrera. Explainable neural-symbolic learning (x-nesyl) methodology to fuse deep learning representations with expert knowledge graphs: The monumai cultural heritage use case. Information Fusion, 79:58–83, 2022. ISSN 1566-2535.
>
> [3]  K. H. Ngan, J. Phelan, E. Mansouri-Benssassi, J. Townsend, and A. S. d’Avila Garcez. Closing the neural-symbolic cycle: Knowledge extraction, user intervention and distillation from convolutional neural networks. In 17th International Workshop on Neural-Symbolic Learning and Reasoning, volume 3432, pp. 19–43.
>
> [4] Covert, I., Lundberg, S., & Lee, S. I. (2021). Explaining by removing: A unified framework for model explanation. Journal of Machine Learning Research, 22(209), 1-90.
>
> [5] Kapishnikov A., Bolukbasi T., Viegas F., Terry M. XRAI: Better Attributions Through Regions. ICCV. 2019.
>
> [6] Yongchao Xu, Thierry Géraud, and Laurent Najman. Connected filtering on tree-based shape-spaces. IEEE transactions on pattern analysis and machine intelligence, 38(6):1126–1140, 2015.
>
> [7] Yongchao Xu, Edwin Carlinet, Thierry Géraud, and Laurent Najman. Hierarchical segmentation using tree-based shape space. IEEE Transactions on Pattern Analysis and Machine Intelligence, 39:1–14, 04 2016. doi: 10.1109/TPAMI.2016.2554550
>
> [8] Miquel Miró-Nicolau, Antoni Jaume i Capó, and Gabriel Moyà-Alcover. Assessing fidelity in xai post-hoc techniques: A comparative study with ground truth explanations datasets. Artificial Intelligence,335:104179, 2024. ISSN 0004-3702.
>
> [9] Benjamin Perret, Giovanni Chierchia, Jean Cousty, Silvio Jamil Ferzoli Guimaraes, Yukiko Kenmochi, and Laurent Najman. Higra (hierarchical graph analyisis) documentation. https://higra.readthedocs.io/,2018.
>
> [10] Benjamin Perret, Giovanni Chierchia, Jean Cousty, Silvio Jamil Ferzoli Guimaraes, Yukiko Kenmochi, and Laurent Najman. Higra: Hierarchical graph analysis. SoftwareX, 10:100335, 2019.

---

### Author Response · Authors · 2025-06-20

We thank all the reviewers for their time and valuable suggestions. We have made our best effort to address all concerns raised and have implemented several revisions to the paper. These modifications are highlighted in red for clarity. Below, we provide a point-by-point response to each reviewer, along with a summary of the main revisions.

**Reviewer 3YHd:** We thank the reviewer for finding our paper interesting and for the valuable suggestions to improve the presentation of our motivation and results. In response, we revised the introduction and added new figures to more clearly convey our main objective: to propose a flexible framework that enables users to prioritize either fidelity or interpretability in model explanations. We also expanded on the explanation behind our use of hierarchies. As suggested, we clarified the existing figures and graphs and included visual summaries of the experimental sections to enhance overall clarity and reader understanding.

**Reviewer PDyw:**  We thank the reviewer for recognizing the novelty of our work in the context of model-based segmentation, as well as for highlighting the value of our systematic approach to multiscale explanations and the flexibility of our framework. We acknowledge the reviewer’s concern regarding the complexity of our approach and would like to emphasize that this complexity primarily arises from the flexibility we intentionally designed to support diverse use cases. While modifying components does require understanding the framework and its implications, this design choice was intentional, allowing users to adapt and extend the framework as needed.  To clarify this, we revised the manuscript to more clearly present flexibility as a core motivation and design principle. Additionally, as also suggested by Reviewer 3YHd, we included a discussion of the framework’s limitations in the main paper.

**Limitations:** Based on feedback from Reviewers PDyw and 3YHd, we recognized the importance of including a discussion of our framework’s limitations in the main paper. As noted by Reviewer PDyw, one limitation is the framework’s dependency on the underlying methods it integrates. To mitigate this, we highlight the conducted extensive testing across a variety of datasets, models, and attribution techniques to ensure generalizability and robustness.

**Reviewer 9sNp**: We thank Reviewer 9sNp for the detailed feedback. Regarding the concern about limited novelty, we acknowledge the importance of innovation in research. However, we respectfully emphasize that, in line with TMLR’s guidelines prioritizing *technical correctness over subjective significance*, our contribution lies in providing a rigorous, modular, and extensible framework with validated results. We agree on the need for extensive experimentation and have conducted over 120 configurations of our framework against five baselines, using two model architectures and three datasets—including ImageNet—evaluated with standard xAI metrics. Due to space constraints, only selected configurations appear in the main paper, with full experiments, qualitative analyses, and user study materials included in the supplementary materials.

In response to comments on related work and comparisons, we have added some of the suggested references, from the neuro-symbolic domain, on the related work discussion. However, our work is situated within the xAI domain, where different assumptions and evaluation standards apply. We have nonetheless ensured that our comparisons use widely accepted baselines and metrics in this field. We also included new references to support our choice of methods.

We acknowledge that the term *reasoning* may not be the most accurate, as it could imply that our method performs full model interpretation or autonomous inference. Our intention was to use *reasoning* in the sense of supporting human reasoning—helping users better understand and navigate model explanations. However, to avoid potential confusion, we decided to change the title of our paper from *Reasoning with trees* to *Explaining with trees*.

Finally, we addressed the concerns about segmentation tree and shaping explanations by adding a diagram and extending our description. Due to space limits, these additions are also available in the supplementary materials.

---

### Decision · Action_Editor_w2bS · 2025-12-07

**Recommendation:** Accept with minor revision

**Additional Comments:**

I have asked for some rewriting of the contribution section to indicate the main results from exclusion, inclusion, PIR metrics and the human study. That would be minimally important given the reviewer consensus on novelty.

**Audience:**

Yes

**Audience Explanation:**

I think this should be of interest to folks who use explainable AI methods and those that do research on it. Authors argue for a complimentary framework with a code accompanied that can integrate with many methods. I think this paper and the resources would be valuable.

**Claims And Evidence:**

Yes

**Claims Explanation:**

I apologize to the authors for my delay in this decision. Given the concerns around novelty and authors response to consider the merits of the work not solely based on subjective notions of importance, I had to take my own time reading the paper (I read the paper and went through the principal results in the supplement).

 Main issues are:

1) Reviews state that there is not much novelty in this paper except proposal of a segmentation hierarchy to view explanations at multiple scales. Sometimes even the segmentation hierarchy is based on existing explanations like IG and BP.

2) Reviews also question the metrics used (inclusion and exclusion based) and  some aspects to the human study.

3) Reviews also state that authors prioritize certain figures to highlight certain aspects of the paper and this priority is not really reflecting the clear contribution of the work.

My own assessment based on reviews:

a) In short, given my reading of the paper, authors at several parts state that looking at any known explanation method through xAI trees trades off fidelity versus interpretability and this is their main contribution. So I am willing to overlook this issue since TMLR is a venue to verify correctness and if there are novelty claims in the paper, only those need to be assessed (as per indications from the EiCs as well). The claim of the authors is *not about a new manner in which we come up with explanation* but rather to view explanations from within a hierarchical segmentation viewpoint. Only this framework is being evaluated.

b) Authors primary evaluation in the paper compares all methods (with their xAI trees and standalone XAI methods) and states 1) If you use exclusion of the region pointed out by a method, there should be a high rate of change from the given class  2) If you include those regions exclusively, there should be a low rate of change from the given class 3) PIR measures the specificity of the region highlighted to avoid huge regions that include the main signal being highlighted to make sure the the first two metrics are not hacked.

Overall, I came away impressed by a reasonably comprehensive user study, detailing exactly what the task was, what was asked, what was desirable and how all methods performed.  I also appreciate authors highlighting other methods like LIME, XRAI's notable performance across these tables.

In my experience reviewing (I state this with all humility) for explainable AI area, one of the constant reminders reviewers give authors is  human corroboration. I think *authors did a commendable job* where they manipulated the fine tuning data (applied on ResNets trained on image net) to introduce bias (cats always with cushions, dogs with grids, etc..) and asked if humans can detect bias and identify cause of bias from various explanations and scored them.


c) (3)  is the issue *where I side with reviewers even after rebuttal*. One suggestion is to avoid too many bold face highlights. The first two pages can crisply say the inclusion, exclusion and PIR metrics and how their xAI tree based methods are more or less competitive uniformly. Further, they can summarize the gist of the human study.  Instead **only** the claims about xAI trees are stated as being desirable etc..

Human study is quoted very briefly in my opinion in the main paper at the end. It is good to bring in the details from the supplement and then summarize it in the first two pages. The three points that detail contribution are not precise enough to know what the contributions (some important details) are.

d) Another issue (which reviewers also hint on) is the lack of dataset and task variety. Most of the experiments are on cats and dogs. I understand the need to study a lot of xAI techniques deeply (here with respect to hierarchical segmentation regions used by different methods). However, the diversity in terms of tasks and datasets could have been a bit more. Although I don't mean for authors to act on this given their work is comprehensive on the baselines benchmarking. Would be nice for the authors to note this in limitations.